

# Conformal geometry from entanglement

Isaac H. Kim[1], Xiang Li[2], Ting-Chun Lin[2], John McGreevy[2] and Bowen Shi[2]

**1** Department of Computer Science, University of California, Davis, CA 95616, USA
**2** Department of Physics, University of California San Diego, La Jolla, CA 92093, USA

## Abstract

In a physical system with conformal symmetry, observables depend on cross-ratios, measures of distance invariant under global conformal transformations (conformal geometry for short). We identify a quantum information-theoretic mechanism by which the conformal geometry emerges at the gapless edge of a 2+1D quantum many-body system with a bulk energy gap. We introduce a novel pair of information-theoretic quantities $(\mathfrak{c}_{\text{tot}}, \eta)$ that can be defined locally on the edge from the wavefunction of the many-body system, without prior knowledge of any distance measure. We posit that, for a topological groundstate, the quantity $\mathfrak{c}_{\text{tot}}$ is stationary under arbitrary variations of the quantum state, and study the logical consequences. We show that stationarity, modulo an entanglement-based assumption about the bulk, implies (i) $\mathfrak{c}_{\text{tot}}$ is a non-negative constant that can be interpreted as the total central charge of the edge theory. (ii) $\eta$ is a cross-ratio, obeying the full set of mathematical consistency rules, which further indicates the existence of a distance measure of the edge with global conformal invariance. Thus, the conformal geometry emerges from a simple assumption on groundstate entanglement. We show that stationarity of $\mathfrak{c}_{\text{tot}}$ is equivalent to a vector fixed-point equation involving $\eta$, making our assumption locally checkable. We also derive similar results for 1+1D systems under a suitable set of assumptions.

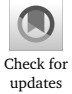

# 1 Introduction

## 1.1 Philosophy of entanglement bootstrap and motivations of this work

In a many-body system consisting of a large number of microscopic degrees of freedom, a new emergent phenomenon may arise at a macroscopic scale [1]. These phenomena form a basis by which one can define a phase of matter, a central concept in condensed matter physics. Intuitively, a phase of matter can be viewed as an equivalence class of renormalization group (RG) flows with the same fixed point [2]. Under the RG flow, different theories at the ultraviolet (UV) may flow to the same infrared (IR) fixed point. These IR fixed points serve as the common route through which one can study universal properties of different phases.

At zero temperature, the theories at the IR fixed points may exhibit exotic emergent phenomena, such as the emergence of anyons in two-dimensional gapped spin liquid systems [3–6]. An important discovery is the fact that many universal properties of the fixed-point are encoded in the entanglement structure of the underlying groundstates. The work of extracting such universal properties from groundstates are numerous: In 2+1D gapped systems, examples of such work include the extraction of quantum dimensions of anyons [7,8], anyon types and fusion rules [9], chiral central charge [10]. In critical systems, the central charge can be extracted [11]. Since their discovery, these signatures have become useful tools to characterize phases of matter and transitions between them in numerical studies (for a small subset of examples, see *e.g.* [12–21]).

Thus, we are invited to explore the possibility that *all* the universal properties of the phase are encoded in the groundstate. This is surprising because the universal properties of the phase can include data that is *a priori* independent of the groundstate. For instance, important data that defines an anyon theory is the braiding and fusion rules of the anyons, which pertains to the low-energy point-like excitations, and the chiral central charge, which pertains to the heat transport at low but finite temperature.

A surprising aspect of these recent developments is that the universal properties follow from some local conditions on the groundstate entanglement. The success of this approach raises a fundamental question: how do the universal properties of the phase (or critical point) emerge from groundstate entanglement? This question can be unpacked as the following series of questions: Given a state, how can we tell, *from some local conditions*, whether it's a representative state of some phase of matter? If so, what phase does it represent? If not, does it represent a phase boundary, i.e., a critical theory? In addressing these questions, we shall not start from the IR theory in the first place but rather assume several locally-checkable conditions on a given quantum state (we shall call it "a reference state") and examine whether some universal properties or even the whole IR theory is the logical consequence of these local conditions.

Much progress has been recently made towards answering this question. A program called "entanglement bootstrap" (EB) demonstrates that universal properties of 2+1D and 3+1D topologically-ordered phases follow logically from locally-checkable assumptions about the many-body entanglement of local regions on a reference quantum state [9, 22–24]. A similar approach has been advocated for 1+1D conformal field theory (CFT) [25].

In this work, we focus on the emergent phenomenon associated with gapless edges from some 2+1D gapped states. Systems with an energy gap in the bulk[1] can have gapless edges. In some cases, the gapless edge is robust from being gapped out by local perturbations.[2] Examples include edges of chiral gapped states [6] as well as non-chiral states with some non-zero higher central charges [26,27] or non-zero minimal total central charge [28]. In many ex-

---

[1] We will call them gapped systems for short.
[2] Such edge states are commonly called ungappable.

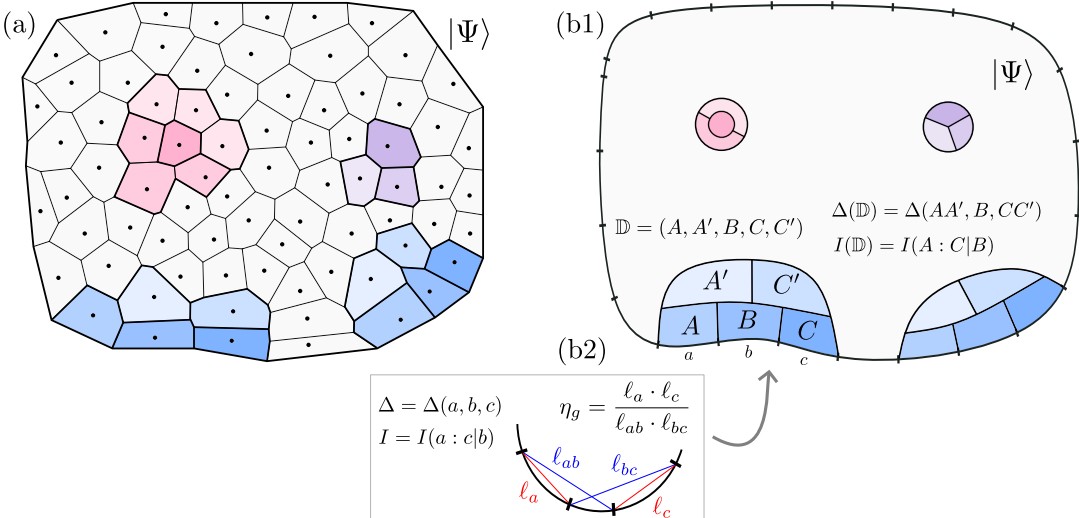

Figure 1: Basic setups. (a) $|\Psi\rangle$ on a "coarse-grained" lattice. Each gray block stands for a coarse-grained site, which is a group of many microscopic degrees of freedom [Section 2]. The regions colored are combinations of regions involved in the three assumptions, which will be explicitly introduced in Section 3 and Section 4. (b1) A "smooth" version. $\mathbb{D} = (A, A', B, C, C')$ is a conformal ruler, where one can compute $\Delta(\mathbb{D}), I(\mathbb{D})$ defined in Eq. (16). (b2) 1+1D CFT groundstate stacked on the edge of a round disk, with the bulk being a trivial product state. In this case, $\Delta(\mathbb{D}) = \Delta(a, b, c) = -\frac{c_{\text{tot}}}{6} \ln(\eta_g)$, $I(\mathbb{D}) = I(a : c|b) = -\frac{c_{\text{tot}}}{6} \ln(1 - \eta_g)$. $\eta_g$ is the geometric cross-ratio, computed using the chord distance (represented by the red and blue lines) of these edge intervals.

amples, such as fractional quantum Hall systems, people have conjectured or verified that the gapless edges are described by CFT at the IR limit [4,29,30]. The key question that motivates this work is: What's the mechanism that results in the emergence of conformal symmetry in these gapless edges? Is the mechanism rooted from quantum entanglement properties of local regions near the edge of a reference state? This question can be explicitly phrased as: *Is the emergence of conformal gapless edge a logical consequence of some quantum entanglement properties of local regions near the edge of a reference state?* Moreover, can we understand the robustness of the ungappable edges in terms of this mechanism? To answer these questions, we must first forgo the CFT assumption and try to identify several locally-checkable conditions on a reference state, from which one can prove the emergence of conformal symmetry. Furthermore, from the robustness of these local conditions, one can tell the robustness of the emergent conformal symmetry. This paper makes the first step towards this goal.

## 1.2 A summary of this work

In this work, we shall study a quantum state $|\Psi\rangle$ on a two-dimensional lattice on a disk; see Fig. 1 for an illustration. We assume that on the coarse-grained lattice in the bulk regions [Fig. 1], the state satisfies one of the entanglement bootstrap axiom called **A1** and has a non-zero chiral central charge computed from modular commutators [Section 3]. These two assumptions are borrowed from previous work [9, 10], and they effectively enforce the bulk wavefunction to be a fixed-point wavefunction of some chiral gapped systems.

Our edge assumptions are defined on a collection of local edge regions $\mathbb{D}$ [Fig. 1(b1)], called as *conformal ruler* for the reason that will be clear later. For each of those regions, we introduce quantum information-theoretic quantities called (total) *central charge* candidate

$\mathfrak{c}_{\text{tot}}(\mathbb{D})_{|\Psi\rangle}$ and *quantum cross-ratio* candidate $\eta(\mathbb{D})_{|\Psi\rangle}$, defined as

$$e^{-6\Delta(\mathbb{D})/\mathfrak{c}_{\text{tot}}} + e^{-6I(\mathbb{D})/\mathfrak{c}_{\text{tot}}} = 1, \qquad \eta \equiv e^{-6\Delta(\mathbb{D})/\mathfrak{c}_{\text{tot}}}, \tag{1}$$

where $\Delta(\mathbb{D})$ and $I(\mathbb{D})$ are two certain linear combination of entanglement entropies in $\mathbb{D}$ [Fig. 1(b1)]:

$$\begin{aligned}
\Delta(\mathbb{D}) &= \Delta(AA', B, CC') = S_{AA'B} + S_{BCC'} - S_{AA'} - S_{CC'}, \\
I(\mathbb{D}) &= I(A : C|B) = S_{AB} + S_{BC} - S_B - S_{ABC}.
\end{aligned} \tag{2}$$

Here $S_X \equiv S(\rho_X) \equiv -\text{Tr}(\rho_X \ln \rho_X)$ is the entanglement entropy of a reduced density matrix on region $X$, computed from the reference state $|\Psi\rangle$. These two particular entropy combinations are designed in a way that UV contributions from these entanglement entropies are canceled. The definition of $\mathfrak{c}_{\text{tot}}(\mathbb{D})$ and $\eta(\mathbb{D})$ are motivated from 1+1D CFT. One can consider a 1+1D CFT groundstate stacking on the edge of a regular disk, with the bulk being a trivial product state. $\Delta(\mathbb{D})$ and $I(\mathbb{D})$ becomes a simple linear combination of entanglement entropies over three contiguous intervals [Fig. 1(b2)]. Utilizing the formula of the entanglement entropy of an interval of chord length $\ell$ on the CFT groundstate [11], $S(\ell) = \frac{c_{\text{tot}}}{6} \ln(\ell/\epsilon)$, where $c_{\text{tot}}$ is the total central charge of the CFT and $\epsilon$ is a UV cutoff, one can explicitly obtain $\Delta(\mathbb{D}) = -\frac{c_{\text{tot}}}{6} \ln(\eta_g)$ and $I(\mathbb{D}) = -\frac{c_{\text{tot}}}{6} \ln(1 - \eta_g)$. Therefore, the solution of Eq. (1) in this case is exactly the total central charge $c_{\text{tot}}$ and the geometric cross-ratio $\eta_g$ of the three contiguous intervals [Fig. 1(b2)]. Our approach is to turn this table around and make $\mathfrak{c}_{\text{tot}}$ and $\eta$ defined in Eq. (1) as a candidate of central charge and cross-ratio over any quantum state without assuming any symmetry beforehand. Remarkably, under the edge assumption we posit on $\mathfrak{c}_{\text{tot}}$, $\eta$ indeed can be interpreted as a cross-ratio, which further indicates the existence of edge conformal geometry. Explicitly, the assumption [Stationarity condition] states: For every $\mathbb{D}$, under any infinitesimal (norm-preserving) variation of the state $|\Psi\rangle \rightarrow |\Psi\rangle + \epsilon |\Psi'\rangle$, $\mathfrak{c}_{\text{tot}}(\mathbb{D})$ is stationary, i.e.

$$\delta \mathfrak{c}_{\text{tot}}(\mathbb{D}) = 0, \tag{3}$$

where $\delta \mathfrak{c}_{\text{tot}}(\mathbb{D})$ denotes the resulting variation of $\mathfrak{c}_{\text{tot}}$ in linear order of $\epsilon$.

We remark that our assumption is motivated by the speculation that $\mathfrak{c}_{\text{tot}}$ defined this way behaves as a $c$-function near the critical RG fixed point, analogous to the one defined by Zamolodchikov [31] or Casini-Huerta [32] in the context of 1+1D relativistic quantum field theory. One evidence for this speculation is that $\mathfrak{c}_{\text{tot}}$ is indeed stationary if the edge physics is described by a 1+1D CFT. This comes from the fact that the stationarity condition is equivalent to a *vector fixed-point equation* in terms of $\eta$ defined in Eq. (1):

$$\left( \eta \hat{\Delta}(\mathbb{D}) + (1 - \eta)\hat{I}(\mathbb{D}) \right) |\Psi\rangle \propto |\Psi\rangle, \quad \forall \mathbb{D}, \tag{4}$$

where $\hat{\Delta}(\mathbb{D}), \hat{I}(\mathbb{D})$ are linear combinations of modular Hamiltonians,

$$\begin{aligned}
\hat{\Delta}(\mathbb{D}) &= \hat{\Delta}(AA', B, CC') = K_{AA'B} + K_{BCC'} - K_{AA'} - K_{CC'}, \\
\hat{I}(\mathbb{D}) &= \hat{I}(A : C|B) = K_{AB} + K_{BC} - K_B - K_{ABC},
\end{aligned} \tag{5}$$

with $K_X \equiv -\ln(\rho_X)$ denoting the modular Hamiltonian of a reduced density matrix on region $X$. Eq. (4) generalizes the vector fixed-point equation derived in 1+1D CFT [25]. In particular, since this equation is satisfied by 1+1D CFT [25], the stationarity condition holds true. We will discuss the equivalence of the two conditions in more detail in Section 4.

Our approach to demonstrating the emergence of conformal geometry is to derive the defining relations of cross-ratios. More precisely, on the physical setup described in [Section 2], based on these three assumptions [Section 3 and 4] — (1) bulk **A1**, (2) non-zero

chiral central charge from the bulk modular commutator, and (3) stationarity condition of $(\mathfrak{c}_{\text{tot}})_{|\Psi\rangle}$ — we prove that $\eta$ satisfies certain consistency relations [Section 5]. These consistency relations also appear in the mathematics literature, which axiomatically define a set of cross-ratios [33,34]. Moreover, these relations enable us to map all the (coarse-grained) edge intervals to a set of intervals on a circle, such that the quantum cross-ratios determined from our method are precisely equal to the geometric cross-ratios computed on the circle. Therefore, these consistency relations are enough to justify viewing $\eta$s as legitimate cross-ratios. We will explain this fact in detail in [Section 5.3]. In addition, under these three assumptions, utilizing the cross-ratio relations, we show that $\mathfrak{c}_{\text{tot}}$ is the same for every region along the edge [Section 5.4]. In [Section 6], replacing the non-zero chiral central charge assumption with another assumption [genericity condition], we derive a similar set of results for non-chiral states.

Let us remark on the significance of our main result, the emergence of cross-ratios on the chiral edge. Firstly, cross-ratios provide a distance measure modulo global conformal transformation. The emergence of cross-ratios indicates the emergence of conformal geometry, whose origin is purely quantum information-theoretic; the proper notion of distance for the cross-ratios emerged from our approach, even without making any further assumptions! Secondly, the emergence of conformal geometry is robust. Note that we did not assume any symmetry or geometric property of the edge. Even if the actual edge can be irregular, in the sense that the system does not have any translational symmetry around the edge, our approach continues to work. We discuss a result of a simple numerical example that demonstrates this point in Appendix G. This phenomenon is likely tied to the robustness of the gapless edge under local perturbations.

What is more, the quantum cross-ratios enable us to construct approximate Virasoro generators in the purely chiral state, generalizing the ideas in [35]. Proving the full algebraic relation is tangent to the future work. This work is the root of many future research directions, which will be discussed in Section 7.

## 2 Preliminaries

Throughout this paper, we shall study a many-body quantum state on a two dimensional disk, referred to as the *reference state*. Physically, we can view the reference state as a groundstate of some 2+1D local Hamiltonian with a bulk gap, though we do not make use of this fact. Below we introduce our notations [Section 2.1] and the physical setup [Section 2.2] to describe this state.

### 2.1 Notations

Unless specified otherwise, we shall refer to the reference state as $|\Psi\rangle$ throughout this paper. We shall reserve the uppercase letters to denote subsystems (or equivalently, regions). The complement of a subsystem will be denoted by placing a bar on top. For instance, for a subsystem $A$, $\bar{A}$ is the complement of that region. For denoting the Hilbert space, the symbols representing the subsystem will be placed in the subscript, e.g., $\mathcal{H}_A$.

We shall use the standard notation for the density matrix, using Greek letters such as $\rho, \sigma, \lambda, \dots$ For the reduced density matrix of a subsystem, we define $\rho_A = \text{Tr}_{\bar{A}}(\rho)$. Entanglement entropy of a subsystem is defined as $S(\rho_A) \equiv -\text{Tr}(\rho_A \ln \rho_A)$. We will often deal with various linear combinations of entanglement entropies over different subsystems. When the underlying global state is the same, we shall use the following short-hand notation. Without loss of generality, suppose we are given an expression of the form $\left(\sum_i \alpha_i S_{A_i}\right)_\rho$, where $\alpha_i \in \mathbb{R}$

and $\{A_i\}$ is a collection of subsystems. This expression is defined as

$$\left(\sum_i \alpha_i S_{A_i}\right)_\rho = \sum_i \alpha_i S(\rho_{A_i}). \tag{6}$$

There are two linear combinations of entanglement entropies which shall be used frequently in this paper. The first is the conditional mutual information, defined as

$$I(A:C|B)_\rho \equiv (S_{AB} + S_{BC} - S_B - S_{ABC})_\rho\,. \tag{7}$$

The strong subadditivity (SSA) of the von Neumann entropy [36] can be expressed as $I(A:C|B) \geq 0$. The other is a linear combination that appears in the weak monotonicity inequality,[3] which is

$$\Delta(A,B,C)_\rho \equiv (S_{AB} + S_{BC} - S_A - S_C)_\rho\,. \tag{8}$$

We remark that both quantities are non-negative.

We shall also consider operator analogs of Eq. (7) and Eq. (8). Let $K_A \equiv -\ln\rho_A$ be the modular Hamiltonian. These are defined as

$$\begin{aligned}
\hat{I}(A:C|B)_\rho &\equiv K_{AB} + K_{BC} - K_B - K_{ABC}\,, \\
\hat{\Delta}(A,B,C)_\rho &\equiv K_{AB} + K_{BC} - K_A - K_C\,.
\end{aligned} \tag{9}$$

We remark that $\hat{I}(A:C|B)$ is not necessarily positive semi-definite. However, $\hat{\Delta}(A,B,C)$ is indeed positive semi-definite [37].

## 2.2 Physical setup

The reference state $|\Psi\rangle$ can be defined on any two-dimensional manifold with boundaries, although we focus on a disk-like geometry for concreteness. More precisely, we envision a two-dimensional disk consisting of microscopic degrees of freedom, each locally interacting with each other. The reference state is a vector in a many-body Hilbert space that has a tensor product structure (or a $\mathbb{Z}_2$-graded tensor product structure for fermions) over these microscopic degrees of freedom.

Although the reference state is formally defined over these microscopic degrees of freedom, we shall study the same state from a more coarse-grained point of view. By partitioning the system into large disks and viewing each disk as a "supersite," we obtain a state defined over a coarse-grained lattice [Fig. 2]. Each site of the lattice now contains a large enough number of degrees of freedom so as to satisfy the assumptions elucidated in Section 3 and 4. Although there are more fine-grained spatial structures within each supersite, we will remain agnostic about this internal structure, simply viewing each supersite as an indecomposable object.

At this point, a natural question is how large the supersite should be. From a RG point of view, the disks ought to be large enough so that the physics at the scale of the supersites can be accurately described by an effective theory in the IR. More specifically, in the IR we expect the local reduced density matrices over a few supersites to satisfy certain nontrivial conditions. (These conditions are the bulk assumptions and edge assumptions we describe in Section 3 and 4.) Furthermore, we expect the effective theory in the IR to emerge from these conditions.

We remark that such a coarse-graining isn't part of the assumptions in the logical framework in our work. As long as the local conditions describe in Section 3 and 4 are satisfied, we can say the microscopic degrees of freedom has been coarse-grained enough.

---

[3]Weak monotonicity, $\Delta(A,B,C) \geq 0$ is equivalent to the strong subadditivity of entropy by the trick of purifying a quantum state, as is well known.

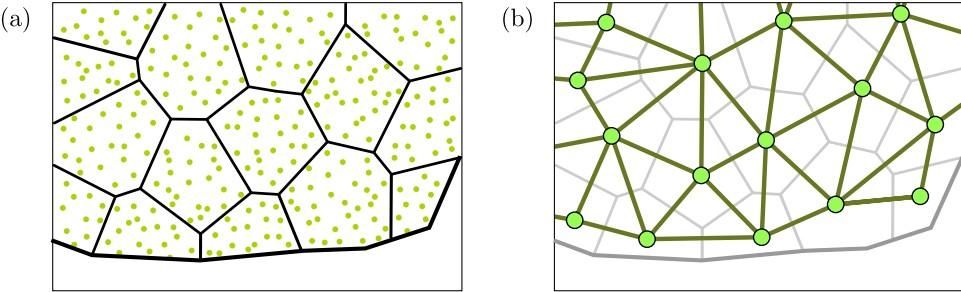

Figure 2: A coarse-grained lattice with an edge. (a) Each coarse-grained site (super-site) is a face that contains a set of sites contained within the face. (b) Coarse-grained sites are connected to each other by an edge if their corresponding faces are adjacent to each other.

## 3 Bulk assumptions

We now introduce the main assumptions about the reference state in the bulk, referred to as the *bulk assumptions*. These are assumptions imposed on regions away from the edge, borrowed from the recent developments in entanglement bootstrap [9,10]. The entanglement bootstrap program rests on two basic axioms, referred to as **A0** and **A1**. In this paper, we will only impose **A1** in the bulk, discussed in more detail in Section 3.1. (The reason for dropping **A0** as well as its potential usage in future works are discussed in Section 3.3.) In addition, we assume that the bulk is chiral in the sense we make precise in Section 3.2.

We remark that we do not anticipate the assumptions presented below to hold on every physical state. In fact, there are states that break our assumption in a robust manner, such as highly-excited states, or even groundstates in the presence of defects [38,39] and domain walls [22,40,41]. Understanding the origin of such violations is of independent interest.

### 3.1 Bulk A1

The first bulk assumption, which we refer to as "bulk **A1**," is one of the entanglement bootstrap axioms [9]:

**Assumption 3.1** (Bulk **A1**). *We assume the reference state $|\Psi\rangle$ satisfies **A1**: for any disk-like region of linear size $O(1)^4$ in the bulk with partition $BCD$ topologically equivalent to the one in Fig. 3,*

$$\Delta(B, C, D)_{|\Psi\rangle} \equiv (S_{BC} + S_{CD} - S_B - S_D)_{|\Psi\rangle} = 0. \tag{10}$$

One way to understand this assumption is to use the area law of entanglement entropy. For any region $A$ in the bulk, the entanglement entropy satisfies

$$S(A) = \alpha|\partial A| - \gamma + \dots, \tag{11}$$

where $\alpha$ is a UV-dependent quantity, $\gamma$ is the topological entanglement entropy [7,8], and the ellipsis is the subleading term that vanishes in the limit of $|\partial A| \to \infty$. Importantly, one can verify that this form of the area law implies the bulk **A1** assumption.

**A1** is useful because one can deduce from it that certain density matrices are quantum Markov chains. Generally speaking, a tripartite quantum state $\rho_{ABC}$ is a quantum Markov chain if it satisfies $I(A : C|B)_\rho = 0$. Assuming **A1** holds for the subsystem $BCD$, for any $A$ in the complement of $BCD$, the strong subadditivity (SSA) of entropy implies the following:

$$I(A : C|B)_\rho \le \Delta(B, C, D)_\rho = 0. \tag{12}$$

---

[4]i.e. order 1 of the coarse-grained lattice sites.

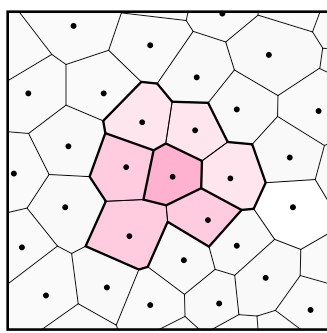 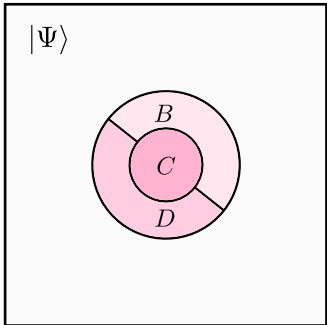

Figure 3: Axiom **A1**: $\Delta(B,C,D)_{|\Psi\rangle} \equiv (S_{BC} + S_{CD} - S_B - S_D)_{|\Psi\rangle} = 0$ for partition of a bulk disk into $B, C, D$ in a way topologically equivalent in the figure.

Because $I(A:C|B) \geq 0$ again by SSA, we conclude that $I(A:C|B)_\rho = 0$ and therefore $\rho_{ABC}$ is a quantum Markov chain. Note that this argument is agnostic about the choice of $A$ as long as it is in the complement of $BCD$. In particular, $A$ can include regions on the edge, even though we made no assumption about the edge so far!

Throughout this paper, we will utilize the quantum Markov chain structure in two ways. Firstly, although we only assumed **A1** in every $O(1)$-sized regions, this assumption implies that **A1** holds at a larger scale. (This is referred to as "extension of axioms" in [9].) This allows one to deform the regions used in certain linear combinations of entropies; see Appendix B for details. Secondly, a quantum Markov chain enables us to decompose the modular Hamiltonian of larger regions in terms of those on smaller regions [42]: for any state $|\psi\rangle$,

$$I(A:C|B) = 0 \quad \Longleftrightarrow \quad K_{ABC}|\psi\rangle = (K_{AB} + K_{BC} - K_B)|\psi\rangle . \tag{13}$$

We will apply this identity throughout this paper.

We note that in various lattice models, such as those with non-zero chiral central charge, **A1** can be satisfied only approximately, i.e., $\Delta(B,C,D)_{|\Psi\rangle} \approx 0$, if the dimension of the local Hilbert space is finite (e.g. [43,44]). Thus our results would not directly apply to such lattice models. Nevertheless, it has been observed in numerical studies that the violation decreases as the subsystem size increases [35]. Therefore, we anticipate our results to be applicable in the limit where the size of every considered subsystem becomes large. More generally, we anticipate that if the quantum state is "close enough" to a zero-correlation length RG fixed-point, then the violation of bulk **A1** will decrease to zero as the state further approaches the fixed-point. Proving such a statement on a rigorous footing is a subject for future work.

## 3.2 Non-zero chiral central charge

The second bulk assumption states that the bulk is chiral. Our assumption can be precisely stated in terms of the modular commutator [10,45]. This is a quantity defined for any tripartite quantum state $\rho_{ABC}$, denoted as $J(A,B,C)_{\rho_{ABC}}$:

$$J(A,B,C)_{\rho_{ABC}} \equiv i\mathrm{Tr}([K_{AB},K_{BC}]\rho_{ABC}). \tag{14}$$

Throughout this paper, we will define the chiral central charge as a constant $c_-$ appearing in the following formula [10,45]:

$$J(A,B,C)_{|\Psi\rangle} = \frac{\pi}{3}c_-, \tag{15}$$

where $ABC$ is a local disk with partition shown in Fig. 4. Due to **A1**, the value of $c_-$ obtained from Eq. (15) is a constant everywhere in the bulk [10,45]. Now we can state the second bulk assumption:

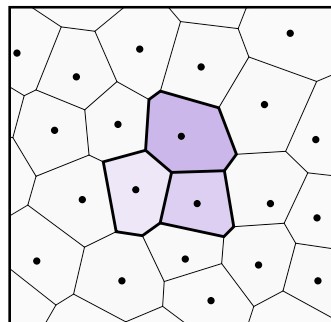
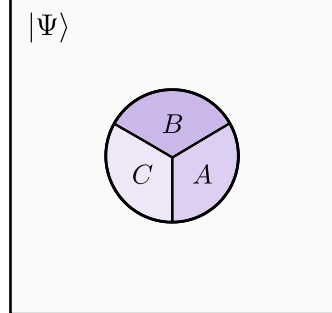

Figure 4: Partition of a disk-shaped region $ABC$ in the bulk. Each subsystem is assumed to be sufficiently large compared to the correlation length.

**Assumption 3.2** (Non-zero chiral central charge). *We assume the state $|\Psi\rangle$ is chiral in the sense that the chiral central charge $c_-$ computed from Eq. (15) is non-zero.*

Throughout the rest of the paper, when we refer to the chiral central charge $c_-$, we mean the one computed from Eq. (15). There has been evidence [45, 46] on why this definition of chiral central charge should match the traditional definition [6, 47, 48] on physical states. As such, we shall call the state with $c_- \neq 0$ a *chiral state*.

### 3.3 A comment on the role of A0

There is another axiom of entanglement bootstrap, known as **A0**. It states that for any $BCD$ partition of a bulk disk of the topology shown in Fig. 3, $S_C + S_{BCD} - S_{BD} = 0$ for the reference state of interest. Because our results simply do not make use of this assumption, the conclusions drawn in this work should apply regardless of **A0**. Nonetheless, **A0** may play a nontrivial role in the future. We briefly comment on that prospect below.

Assuming **A1**, the fact that $c_-$ attains a constant value everywhere in the bulk was proved in Ref. [10]. However, without **A0**, it is not possible to conclude that $c_-$ is quantized. If we do not demand **A0**, we can consider a reference state of the form of $|\Psi\rangle = \sqrt{p}|\Psi_1\rangle + \sqrt{1-p}|\Psi_2\rangle$, where $p \in (0, 1)$ and $|\Psi_1\rangle$ and $|\Psi_2\rangle$ are two topologically ordered groundstates. Let us further suppose that the two states are supported on orthogonal subspaces on each lattice site. The chiral central charge (computed from the modular commutator) would be $c_- = pc_-^1 + (1-p)c_-^2$, which can be continuously tuned between $c_-^1$ and $c_-^2$. For the chiral central charge appearing in the anyon theory, it is well-known that it must attain a quantized value related to the universal properties of the anyons; see [6, Appendix E]. In order to rule out examples like this, we would need to assume **A0**.

## 4 Edge assumption

The two bulk assumptions reviewed in Section 3 are the assumptions already used in the existing literature [9, 10]. In this Section, we introduce a new assumption on the edge from which certain features of the conformal symmetry emerge.

In order to state our assumption, we shall first define a pair of information-theoretic quantities from a region adjacent to the edge [Fig. 5], denoted as $\mathfrak{c}_{\text{tot}}$ and $\eta$ [Section 4.1]. These quantities shall ultimately correspond to the central charge and the cross-ratio of a CFT under our assumption, though at this point they are merely some information-theoretic quantities definable over any quantum state. In Section 4.2, we put forward our main assumption — formulated in terms of $\mathfrak{c}_{\text{tot}}$ and $\eta$ — from which aspects of the conformal symmetry emerge.

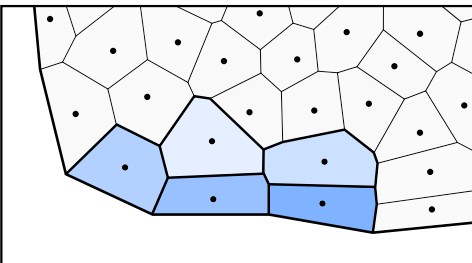 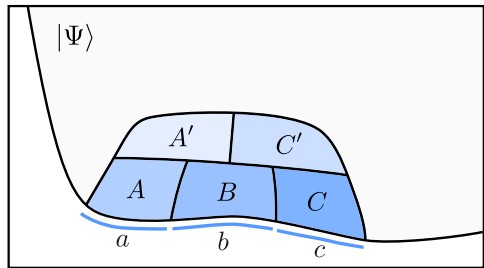

Figure 5: A conformal ruler near the edge. It is a disk-like region with a partition $\mathbb{D} = (A, A', B, C, C')$ with a topology as shown. The three contiguous edge intervals $a, b, c$ are obtained by the intersection of $A, B, C$ with the edge.

## 4.1 Key concepts: Central charge and quantum cross-ratio candidates

In this Section, we introduce two information-theoretic quantities that will play a central role in this paper. These quantities will ultimately correspond to the central charge and the cross-ratio. However, without making any further assumptions (such as the ones in Section 4.2) such an interpretation cannot be justified. Therefore, for now we simply refer to them as *central charge candidate* and *quantum cross-ratio candidate,* denoted as $\mathfrak{c}_{\text{tot}}$ and $\eta$, respectively. Later in Section 5 and 6, we will provide conditions under which these can be viewed as the central charge and the cross-ratio. In that context, we will refer to them simply as central charge and quantum cross-ratio.

Here are the main motivations behind our definitions. Our definitions of $\mathfrak{c}_{\text{tot}}$ and $\eta$ are aimed at identifying the central charge and the cross ratio near the edge of a 2+1D groundstate with a gapped bulk, using entanglement entropies. In order to isolate the contribution to the entanglement entropy from the edge, we need to ensure that the contributions from the bulk cancel each other out in a judicious way. Furthermore, we want the definitions to be applicable even without any prior knowledge of the distance metric (used in defining the cross-ratio) near the edge.

Both of these challenges can be solved by using a 5-partite block $\mathbb{D} = (A, A', B, C, C')$ [Fig. 5]. Each such block allows us to compute a pair $(\mathfrak{c}_{\text{tot}}, \eta)$ [Definition 4.1]. Such pairs can be used to (i) determine whether the underlying state allows a conformal distance measure on its edge [Section 4.2], and (ii) assign such a conformal distance measure (i.e. a distance measure modulo global conformal transformation) to the coarse-grained edge interval [Section 5.3]. Because such a block $\mathbb{D}$ allows us to recover the conformal distance measure, we refer to it as a *conformal ruler*.

For each such $\mathbb{D} = (A, A', B, C, C')$, we put forward two linear combinations of entropies, which carefully cancel out area law contribution from the bulk, namely

$$
\begin{aligned}
\Delta(\mathbb{D}) &\equiv \Delta(AA', B, CC') = S_{AA'B} + S_{CC'B} - S_{AA'} - S_{CC'}, \\
I(\mathbb{D}) &\equiv I(A : C|B) = S_{AB} + S_{BC} - S_B - S_{ABC}.
\end{aligned}
\tag{16}
$$

These quantities retain nontrivial information about the edge. For the physically interesting case of gapless edges, we expect $I, \Delta > 0$. For gapped edges, both $I$ or $\Delta$ become vanishingly small, in the limit the size of each subsystem becomes large. Unless stated otherwise, for each $\mathbb{D}$, we will demand the following topological requirement on the underlying regions. Firstly, $AA'BCC'$ should be topologically a disk, and $A, B, C$ should be anchored at three contiguous coarse-grained intervals on the edge. Secondly, $AA'CC'$ should completely cover $B$ shielding it from the complement of $AA'BCC'$ [Fig. 5]. Lastly, $A, C$ should not be adjacent to each other.

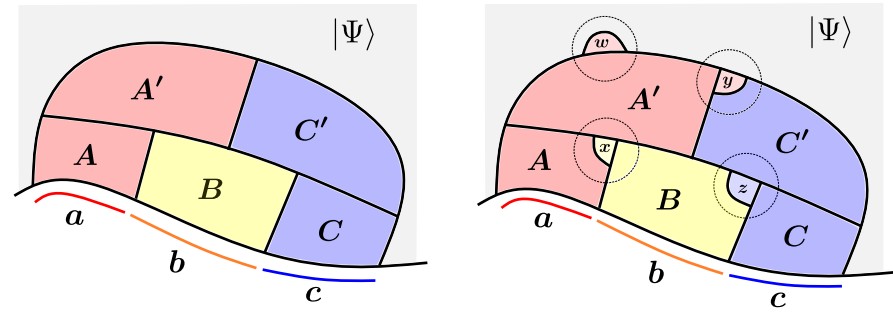

Figure 6: Examples of bulk deformation of a $\mathbb{D}$: (i) $A \rightarrow A \setminus x, B \rightarrow Bx$; (ii) $A' \rightarrow A'w$; (iii) $A' \rightarrow A'y, C' \rightarrow C' \setminus y$; (iv) $C' \rightarrow C'z, B \rightarrow B \setminus z$.

We remark that $\Delta(\mathbb{D}), I(\mathbb{D})$ defined in Eq. (16) is the "canonical" expression once a conformal ruler $\mathbb{D}$ is specified. For example, in the same 5-partite region in Fig. 5, $\mathbb{D}' = (AA', \emptyset, B, C, C')$ is also a conformal ruler, in which $\Delta(\mathbb{D}') = \Delta(AA', B, CC')$, $I(\mathbb{D}') = I(AA' : C|B)$.

We now introduce the definition of $\mathfrak{c}_{\text{tot}}$ and $\eta$.

**Definition 4.1** ($\mathfrak{c}_{\text{tot}}(\mathbb{D})$ and $\eta(\mathbb{D})$). Let $|\Psi\rangle$ be a state on a disk that satisfies bulk **A1**. Consider a conformal ruler $\mathbb{D} = (A, A', B, C, C')$. We define $\mathfrak{c}_{\text{tot}}(\mathbb{D})_{|\Psi\rangle}$ and $\eta(\mathbb{D})_{|\Psi\rangle}$ as the solution to the following equations:

$$e^{-6\Delta(\mathbb{D})/\mathfrak{c}_{\text{tot}}(\mathbb{D})} + e^{-6I(\mathbb{D})/\mathfrak{c}_{\text{tot}}(\mathbb{D})} = 1,$$

$$\eta(\mathbb{D})_{|\Psi\rangle} \equiv e^{-6\Delta(\mathbb{D})/\mathfrak{c}_{\text{tot}}(\mathbb{D})}. \tag{17}$$

A few remarks are in order. First, when $I(\mathbb{D}), \Delta(\mathbb{D}) > 0$, Eq. (17) has a unique solution, where

$$\mathfrak{c}_{\text{tot}}(\mathbb{D})_{|\Psi\rangle} > 0, \qquad \eta(\mathbb{D})_{|\Psi\rangle} \in (0, 1). \tag{18}$$

When $I = 0$ or $\Delta = 0$, we can set $\mathfrak{c}_{\text{tot}}(\mathbb{D})_{|\Psi\rangle} = 0$, which is the limit of $\mathfrak{c}_{\text{tot}}$ defined in Eq. (17) as $\Delta \rightarrow 0$ or $I \rightarrow 0$ [see Appendix C]. Secondly, while $\mathfrak{c}_{\text{tot}}$ and $\eta$ become the central charge and the cross ratio *under some assumptions*, more generally, one cannot interpret them in such a way. We shall discuss the relevant examples in the latter part of this Section. Thirdly, $\mathfrak{c}_{\text{tot}}(\mathbb{D})_{|\Psi\rangle}$ and $\eta(\mathbb{D})_{|\Psi\rangle}$ are invariant under the deformations of the conformal ruler in the bulk, such as the one shown in Fig. 6. This is because both $\Delta$ and $I$ are invariant under such deformation, a fact that follows straightforwardly from bulk **A1** [Appendix B].

Continuing the last remark, due to the invariance under the deformation in the bulk, it is sometimes more informative to specify the conformal ruler in terms of the edge intervals. Without loss of generality, let $a, b$, and $c$ be the edge intervals on which the subsystems $A, B$, and $C$ are anchored. We will sometimes refer to a conformal ruler of those regions as $\mathbb{D}(a, b, c)$. While this notation hides the explicit choice of $A, B, C, A'$, and $C'$, these details are irrelevant in calculations involving $I(\mathbb{D})$ and $\Delta(\mathbb{D})$. Later in Section 5 and 6, we will use even more succinct notations, such as $\Delta_{a,b,c}$ and $I_{a,b,c}$ for $\Delta(\mathbb{D}(a, b, c))$ and $I(\mathbb{D}(a, b, c))$.

We now provide examples for which $\mathfrak{c}_{\text{tot}}$ and $\eta$ take a clear physical meaning.

**Example 4.2** (1+1D CFT groundstate). The simplest example is a 1+1D CFT groundstate on a circle. We identify the circle with the edge, and the bulk is left empty. The entanglement entropy of an interval in the groundstate of a 1+1D CFT on a circle is $S(\ell) = \frac{c_{\text{tot}}}{6} \ln(\ell/\epsilon)$ [11], where $\ell$ is the chord length of the interval, $\epsilon$ is a cutoff, and $c_{\text{tot}}$ is the total central charge. One can calculate

$$\Delta = \Delta(a, b, c), \qquad I = I(a : c|b). \tag{19}$$

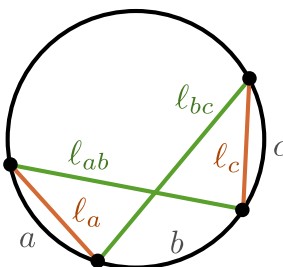

Figure 7: The chord lengths used in defining the geometric cross-ratio on a circle.

Plugging them into Definition 4.1, one can see that indeed $\mathfrak{c}_{\text{tot}}$ equals the total central charge of the CFT and $\eta$ equals the geometric cross-ratio ($\eta_g$) for the interval $(a, b, c)$ on a circle:

$$\mathfrak{c}_{\text{tot}} = c_{\text{tot}}, \qquad \eta = \eta_g \equiv \frac{\ell_a \cdot \ell_c}{\ell_{ab} \cdot \ell_{bc}}, \tag{20}$$

where $\ell_a$ is the chord length associated with interval (or arc) $a$. See Fig. 7 for an illustration.

**Example 4.3.** There is a limit of our quantity $\mathfrak{c}_{\text{tot}}$ in which, when applied to the groundstate of a relativistic 1+1D QFT, it is related to Casini and Huerta's c-function $c_{\text{CH}} \equiv 6r\partial_r S(r)$, where $S(r)$ is the entanglement entropy of an interval of length $r$ [49]. Using strong subadditivity and Lorentz symmetry, Casini and Huerta showed that this quantity is monotonic under RG flows of couplings in relativistic QFT.

Consider a translation-invariant state and suppose that $\eta$ is equal to the geometrical cross-ratio,[5] so

$$\mathfrak{c}_{\text{tot}} = 6\frac{\Delta}{\ln 1/\eta}. \tag{21}$$

Take the regions to be as in the argument for constraints on derivatives of $S(r)$ from SSA in [50], so $|ab| = |bc| = r, |a| = |c| = r - \delta r$, and take $\delta r$ infinitesimal. Then $\eta = (r - \delta r)^2/r^2 = 1 - 2\delta r, \ln 1/\eta = 2\delta r$ and

$$\mathfrak{c}_{\text{tot}} = 6\frac{\Delta}{\ln 1/\eta} = 6\frac{2S(r) - 2S(r - \delta r)}{2\delta r} = 6r\partial_r S(r) = c_{\text{CH}}, \tag{22}$$

the RG monotone of Casini and Huerta. The relation (22) between $c_{\text{CH}}$ and $\Delta$ is not entirely a surprise since the fact that $c_{\text{CH}}$ can be related to the quantity appearing in the weak monotonicity inequality is the crucial step of their proof of RG monotonicity [49].

**Example 4.4** (Chiral gapped system on a disk). Another class of examples is chiral gapped systems on a disk. Here, we assume that the bulk is gapped and satisfies the area law and the edge is one that obeys Hypothesis 1 of [35], concerning the CFT behavior of the chiral edge; as explained in the reference, one can conclude that $\mathfrak{c}_{\text{tot}}$ is the central charge and $\eta$ is the geometric cross-ratio. (For an alternative physical argument, see the "cylinder argument" in the same reference.)

**Example 4.5** (Chiral gapped system with an irregular edge). Here is an example from numerical observation. On an irregular edge of $p + ip$ superconductor (detailed in Appendix G), we computed $\mathfrak{c}_{\text{tot}}$ and $\eta$. The value $\mathfrak{c}_{\text{tot}} \approx 1/2$ matches the anticipated central charge of the edge. The values of $\eta$s computed from different choices of edge intervals satisfy the consistency relations one would expect for a set of cross-ratios with high precision. (See Section 5.2 for those rules). Unlike the previous examples, in which the distance measure is given to us from the beginning, this example does not begin with any preferred choice of distance measure.

---

[5]The reader will wonder how strong this assumption is. Indeed we should not expect it to hold for QFTs far from a fixed point. However, it should hold for small deviations away from a CFT.

## 4.2  Assumption: Stationarity condition

We now introduce our rather minimalistic edge assumption. The assumption we put forward is the *stationarity condition* of $\mathfrak{c}_{\text{tot}}$, which posits that $\mathfrak{c}_{\text{tot}}$ is invariant under infinitesimal norm-preserving perturbations.

Without loss of generality, consider any norm-preserving perturbation of the state $|\Psi\rangle$ of the form $|\Psi\rangle \to |\Psi\rangle + \epsilon|\Psi'\rangle$, where $\epsilon \in \mathbb{R}$ is infinitesimal and $|\Psi'\rangle$ is a state orthogonal to $|\Psi\rangle$. We use $\delta\mathfrak{c}_{\text{tot}}$ to denote the resulting variation of $\mathfrak{c}_{\text{tot}}$ in linear order of $\epsilon$. The stationarity condition states:

**Assumption 4.6** (Stationarity condition)**.** *We assume the state $|\Psi\rangle$ satisfies the following stationarity condition: for every conformal ruler $\mathbb{D}$,*

$$\delta\mathfrak{c}_{tot}(\mathbb{D})_{|\Psi\rangle} = 0\,, \tag{23}$$

*for any norm-preserving perturbation of $|\Psi\rangle$.*

Interestingly, the stationarity condition turns out to be equivalent to a seemingly unrelated condition. This is the *vector fixed-point equation* involving $\eta$ and the modular Hamiltonians, similar to the one introduced in Ref. [25]. In order to describe this assumption, it will be helpful to introduce the following notation. Given a conformal ruler $\mathbb{D} = (A, A', B, C, C')$, we can consider operator analogs of $\Delta(AA', B, CC')$ and $I(A : C|B)$:

$$\begin{aligned}
\hat{\Delta}(\mathbb{D}) &\equiv \hat{\Delta}(AA', B, CC') = K_{AA'B} + K_{CC'B} - K_{AA'} - K_{CC'}\,, \\
\hat{I}(\mathbb{D}) &\equiv \hat{I}(A : C|B) = K_{AB} + K_{BC} - K_B - K_{ABC}\,.
\end{aligned} \tag{24}$$

As we show in Appendix B, when those operators act on $|\Psi\rangle$, one can deform their supports in the bulk region. Therefore, for a $\mathbb{D}$ that anchors on the edge intervals $a, b, c$, we sometimes denote $\hat{\Delta}(\mathbb{D}) = \hat{\Delta}_{a,b,c}$, and $\hat{I}(\mathbb{D}) = \hat{I}_{a,b,c}$, when they act on the reference state.

Now we can state our stationarity condition equivalently using the vector fixed-point equation:

**Definition 4.7** (Vector fixed-point equation)**.** *A state $|\Psi\rangle$ satisfies the vector fixed-point equations if the following is true: For any conformal ruler $\mathbb{D}$,*

$$\mathcal{K}_{\mathbb{D}}(\eta(\mathbb{D}))|\Psi\rangle \propto |\Psi\rangle\,, \tag{25}$$

*where*

$$\mathcal{K}_{\mathbb{D}}(x) \equiv x\hat{\Delta}(\mathbb{D}) + (1-x)\hat{I}(\mathbb{D})\,. \tag{26}$$

The equivalency relation between the stationarity condition and the vector fixed-point equation [Definition 4.7] follows from the theorem stated below:

**Theorem 4.8.** *Consider a conformal ruler $\mathbb{D} = (A, A', B, C, C')$, with $\mathfrak{c}_{tot}(\mathbb{D})_{|\Psi\rangle}$ and $\eta(\mathbb{D})_{|\Psi\rangle}$ defined as in Definition 4.1. $\mathfrak{c}_{tot}(\mathbb{D})_{|\Psi\rangle}$ is stationary if and only if*

$$\mathcal{K}_{\mathbb{D}}(\eta)|\Psi\rangle \propto |\Psi\rangle\,, \tag{27}$$

*where $\eta$ is the solution of Eq. (17).*

The proof of this theorem is given in Appendix D.
Following this theorem, one can conclude that the stationarity condition, namely $\mathfrak{c}_{\text{tot}}(\mathbb{D})_{|\Psi\rangle}$ is stationary for every conformal ruler $\mathbb{D}$, is equivalent to the vector fixed point equation condition [Definition 4.7].

While the stationarity condition [Assumption 4.6] and the vector fixed-point equation condition [Definition 4.7] are equivalent thanks to Theorem 4.8, the motivations behind them are different. A motivation behind Assumption 4.6 is to define a $c$-function $\mathfrak{c}_{\text{tot}}$ that generalizes the central charge of the 1+1D CFT, even for non-relativistic systems. For relativistic systems, there are $c$-functions which monotonically decrease under RG flow [31, 32, 49]. In particular, at an RG fixed point, the $c$-function ought to be stationary against perturbations of the Lagrangian. Assumption 4.6 posits a condition in this spirit.[6] A motivation behind the vector fixed-point equation [Definition 4.7] is to define a proper notion of cross ratio $\eta$ even without knowing the distance measure. For a 2+1D gapped system on a disk whose edge is described by a CFT, if the edge is known to have a translational symmetric geometry, then as argued in [35], it is expected that the edge shall hold the vector fixed-point equation [Definition 4.7] with geometric cross-ratio, which is a generalization of the vector fixed-point equation for 1+1D CFT groundstate [25]. However, in systems in which the translational symmetric is not manifest or even absent, e.g., the edge of a disordered quantum Hall system, it is less clear how to define the cross ratio. Demanding the vector fixed-point equation is one viable approach. It is remarkable that the two assumptions motivated from seemingly unrelated reasons are in fact equivalent to each other.

Although the stationarity condition may appear to be a global condition, it is in fact locally checkable due to its equivalence to the vector fixed-point equation. That is, one only needs to work with the reduced density matrix ($\rho_{\mathbb{D}}$) on the conformal ruler $\mathbb{D}$:

$$\delta\mathfrak{c}_{\text{tot}}(\mathbb{D})_{|\Psi\rangle} = 0 \quad \Longleftrightarrow \quad \mathcal{K}_{\mathbb{D}}(\eta)|\Psi\rangle \propto |\Psi\rangle \quad \Longleftrightarrow \quad \mathcal{K}_{\mathbb{D}}(\eta)\rho_{\mathbb{D}} \propto \rho_{\mathbb{D}}. \tag{28}$$

The first $\Longleftrightarrow$ is the content of the Theorem 4.8. Now we explain the second $\Longleftrightarrow$. The $\Rightarrow$ direction is simple as one can simply trace out the complement of $AA'BCC'$ on both hand side of $\mathcal{K}_{\mathbb{D}}(\eta)|\Psi\rangle\langle\Psi| \propto |\Psi\rangle\langle\Psi|$. To see the $\Leftarrow$, one can first purify $\rho_{\mathbb{D}} \to |\Psi'\rangle$ and obtain $\mathcal{K}_{\mathbb{D}}(\eta)|\Psi'\rangle \propto |\Psi'\rangle$. Then by Uhlmann's theorem [51], one can obtain $|\Psi\rangle$ from $|\Psi'\rangle$ by a unitary $I_{\mathbb{D}} \otimes U_{\overline{\mathbb{D}}}$ whose support is only within $\overline{\mathbb{D}}$. As the unitary commutes with $\mathcal{K}_{\mathbb{D}}(\eta)$, one obtains $\mathcal{K}_{\mathbb{D}}(\eta)|\Psi\rangle \propto |\Psi\rangle$.

One particular usage of the edge assumption is to decompose the modular Hamiltonians on a larger region in terms of the linear combinations of the modular Hamiltonians on smaller regions, when they act on $|\Psi\rangle$. The reverse process also works, allowing us to glue the local modular Hamiltonians to obtain modular Hamiltonian on larger regions. This resembles the situation of quantum Markov chains, where if some state $|\psi\rangle$ satisfies $I(A : C|B)_{|\psi\rangle} = 0$, one can decompose $K_{ABC}|\psi\rangle = (K_{AB} + K_{BC} - K_B)|\psi\rangle$, and vice versa. In the context of 1+1D CFT, this same decomposition idea is observed in [25].

One might wonder why we advocated using the stationarity condition instead of the vector fixed-point equation, in spite of the fact that they are equivalent [Theorem 4.8]. While this is of course a matter of taste, we have reasons to believe that the stationary condition has a potential to be applicable in broader contexts. For one thing, the stationarity condition in its formulation explicitly includes a set of states in a small neighborhood of the underlying state. We can thus speculate that, near the RG-fixed point, $\mathfrak{c}_{\text{tot}}$ monotonically decreases under the RG flow. It will be interesting to understand if our definition of $\mathfrak{c}_{\text{tot}}$ measures the "number of degrees of freedom" in general quantum many-body systems, just like Zamolodchikov's $c$-function does for relativistic systems [31].

There is also numerical evidence for our perspective that these edge conditions hold at fixed points of the RG, and that their violation decreases under coarse-graining. See, for example, Fig. 19 of [35] where the violation of the vector fixed-point equation decreases as the subsystem size increases.

---

[6]A difference is that we are directly perturbing a quantum state, whereas in Ref. [31, 32, 49], it is the Lagrangian that is being perturbed.

The relation between the stationarity condition and the vector fixed-point equation is evocative of the relation between the action principle and the equation of motion. At a classical level, they are equivalent formulations of the same physical laws. Often times, the equation of motion is more practical. However, the action principle played a key role in going beyond classical mechanics in terms of the path integral, thereby explaining the origin of the action principle. This anecdote invites us to wonder if the stationarity condition has more to tell us about the physics of quantum many-body systems in the future.

### 4.3 Examples of stationary states

In this section, we provide several examples of states that satisfy the stationarity condition. This is done by verifying the vector fixed-point equation [Definition 4.7]. Then the claim follows immediately from Theorem 4.8.

**Example 4.9** (Gapped state with gapped boundaries)**.** For a topological order with a gapped boundary, the stationarity condition holds trivially. In the language of entanglement bootstrap, for a gapped boundary, in addition to the bulk **A1** described above, a boundary version of **A1** is also satisfied; see [22]. This boundary axiom is precisely $\Delta = 0$ (defined in (16)) for each choice of $\mathbb{D}$. By strong subadditivity, $I = 0$. Thus

$$\mathfrak{c}_{\text{tot}} = 0, \qquad \hat{I}|\Psi\rangle = \hat{\Delta}|\Psi\rangle = 0. \tag{29}$$

The vector fixed-point equation holds trivially, and this also implies $\delta\mathfrak{c}_{\text{tot}} = 0$. One may also derive $\delta\mathfrak{c}_{\text{tot}} = 0$ bypassing the vector fixed-point equation. Note that $\mathfrak{c}_{\text{tot}} \geq 0$ by definition. Therefore, $\mathfrak{c}_{\text{tot}} = 0$ is the absolute minimum, which implies the stationarity.

**Example 4.10** (1+1D CFT on a circle)**.** To fit our definition of $\mathfrak{c}_{\text{tot}}$ and $\eta$, one can regard the 1+1D CFT groundstate on a circle as the edge state of a 2+1D system on a disk with empty bulk. As we showed before

$$\Delta(AA', B, CC') = \Delta(a, b, c), \qquad I(A : C|B) = I(a : c|b), \tag{30}$$

where $a, b, c$ are the edge intervals associated with region $A, B, C$ as in Fig. 5. Therefore, $\eta = \eta_g$, the geometric cross-ratio computed from the circle (20), and

$$\begin{aligned} \mathcal{K}_{\mathbb{D}}(\eta)|\Psi\rangle &= \left(\eta_g\hat{\Delta}(a, b, c) + (1 - \eta_g)\hat{I}(a : c|b)\right)|\Psi\rangle \\ &\propto |\Psi\rangle, \end{aligned} \tag{31}$$

where the "$\propto$" in the second line is by the vector fixed-point equation derived in [25]. By Theorem 4.8, 1+1D CFT groundstates satisfy the stationarity condition $\delta\mathfrak{c}_{\text{tot}} = 0$.

**Example 4.11** (Chiral states with a bulk energy gap)**.** Consider a chiral state with a bulk energy gap on a two-dimensional manifold with edges. As derived in [35] under some mild hypothesis, for a local region near an edge, the vector fixed-point equation is satisfied:

$$\mathcal{K}_{\mathbb{D}}(\eta)|\Psi\rangle \propto |\Psi\rangle. \tag{32}$$

Here $\eta$ computed from $I$ and $\Delta$ is identical to the geometric cross-ratio. Hence, the stationarity condition is satisfied on the edges of such chiral states.

**Example 4.12** (Chiral gapped system with an irregular edge)**.** We numerically tested the irregular edges of a $p + ip$ superconductor groundstate [Appendix G]. In this setup, a preferred distance measure for the cross-ratio is unclear due to the irregularity. By computing the quantum cross-ratio $\eta$ from the groundstate (Definition 4.1) and checking the validity of the vector

fixed-point equation for this $\eta$, we found a sharp verification of our assumption. More precisely, we have computed the error of vector equation $\mathcal{K}_{\mathbb{D}}(x)|\Psi\rangle \propto |\Psi\rangle$ for $x \in [0,1]$, where $\mathcal{K}_{\mathbb{D}}(x)$ is that in Eq. (26). Here the error is defined as $\sigma(\mathcal{K}_{\mathbb{D}}(x)) = \sqrt{\langle \mathcal{K}_{\mathbb{D}}(x)^2 \rangle - \langle \mathcal{K}_{\mathbb{D}}(x)\rangle^2}$, expectation taken from $|\Psi\rangle$. The error reaches its minimum for $x \approx \eta$. The smallness of the error, $\sigma(\mathcal{K}_{\mathbb{D}}(\eta)) \approx 10^{-3}$ suggests the validity of the vector fixed-point equation $\mathcal{K}_{\mathbb{D}}(\eta)|\Psi\rangle \propto |\Psi\rangle$. It follows that stationarity holds approximately on the edge.

**Non-example 4.13** (Exotic non-CFT states that match CFT entropy)**.** There are states that match CFT entropy on each interval choice, which, nonetheless, violate the vector fixed-point equation. Such states do not satisfy the stationarity condition, and they are not true CFT groundstates. See Appendix H.1 for the construction of such wavefunctions. Note that these exotic examples cannot be distinguished from CFT groundstates by the presence of a constant $\mathfrak{c}_{\text{tot}}$. (A constant $\mathfrak{c}_{\text{tot}}$ always implies that $\eta$ is a cross ratio, as explained in Prop. 5.6 below.)

We remark on the importance of this non-example. During our search for a suitable local edge condition, one failed attempt was to demand $\mathfrak{c}_{\text{tot}}(\mathbb{D})$ to be a constant, independent of the choice of $\mathbb{D}$. At first, this appears to be a reasonable candidate because $\mathfrak{c}_{\text{tot}}(\mathbb{D})$ being constant is equivalent to the condition that $\eta(\mathbb{D})$ is a cross-ratio [Section 5.4]. However, this non-example shows that even a non-CFT state can satisfy these conditions, at least if we only consider a finite subset of coarse-grained intervals. This example is ruled out by the vector fixed-point equation, or equivalently, by the stationarity condition. While these conditions do imply the constant $\mathfrak{c}_{\text{tot}}(\mathbb{D})$, the converse is not necessarily true.

# 5 Emergence of conformal geometry: Chiral edge

In this section, we will prove the emergence of conformal geometry based on the setups [Section 2] and three assumptions [Section 3 and 4]. Namely, we shall assume bulk **A1** [Assumption 3.1], nonzero chiral central charge [Assumption 3.2], and stationarity [Assumption 4.6]. By conformal geometry, we mean the existence of a map from the chiral edge to a round circle, which allows us to define a distance modulo global conformal transformation on the chiral edge [Prop. 5.4]. The existence of such a map follows from the relations between the quantum cross ratio candidates [Prop. 5.2 and 5.3], which follow from our three assumptions.

We summarize the route towards the proof of this main result in Fig. 8. Also summarized are the major results we derive along the way. Below is a summary of the content of the subsections, with bracketed remarks pointing to their relation with Fig. 8. In Section 5.1, we first identify two other quantum cross-ratio candidates $\eta_J$ and $\eta_K$ which are a priori unrelated to the quantum cross-ratio candidate $\eta$ [Section 4.1]. We prove that these alternative candidates are equal to $\eta$. Then, in Section 5.2, we prove that $\eta$ satisfies a set of relations that *define* cross-ratios [Gray box]. In Section 5.3, we explain that our quantum cross-ratios provide a distance measure modulo global conformal transformations [pink box]. Finally, we show that $\mathfrak{c}_{\text{tot}}$ being constant is equivalent to $\eta$ being a cross-ratio in Subsection 5.4 [purple box]; importantly, stationarity implies that $\mathfrak{c}_{\text{tot}}$ is constant [Prop. 5.5], though the converse is not necessarily true [Appendix H].

## 5.1 Three candidate cross-ratios

Recall that, given a conformal ruler $\mathbb{D}$ near the edge, we defined a quantum cross-ratio candidate $\eta(\mathbb{D})$ by Eq. (17) from $\Delta$ and $I$ [Section 4.1]. We referred to $\eta(\mathbb{D})$ as a quantum cross-ratio candidate because it is computed from quantum information quantities without the help of any distance measure, and it becomes the geometric cross-ratio when the state is



Figure 8: A summary of the main results and the routes to prove them from the assumptions.

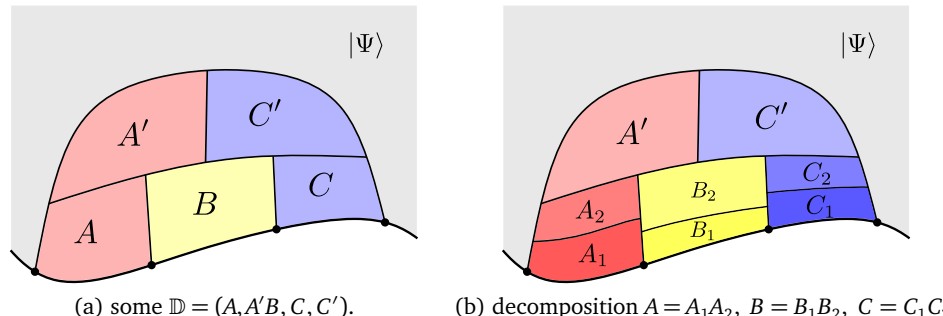

(a) some $\mathbb{D} = (A, A'B, C, C')$.    (b) decomposition $A = A_1 A_2$, $B = B_1 B_2$, $C = C_1 C_2$.

Figure 9: Diagrams for showing $J(AA', B, CC') = \frac{\pi c_-}{3} + J(A, B, C)$.

a 1+1D CFT (Example 4.2 and 4.10). Are there other ways to compute the CFT geometric cross-ratio with quantum information quantities? Can they be similarly promoted to quantum cross-ratio candidates in the broader setup of interest to us? Under what assumptions do they agree? Here we discuss two more such candidates, $\eta_J$ and $\eta_K$.

In [52], it was argued based on CFT assumptions that cross-ratio shows up in the modular commutator computed near a chiral edge. Motivated by this fact, one can formally define a cross-ratio candidate $\eta_J$ according to

$$J(A, B, C) = -\frac{\pi c_-}{3} \eta_J,\tag{33}$$

where $A, B, C$ here belongs to some conformal ruler $\mathbb{D} = (A, A', B, C, C')$ [Fig. 9(a)]. Note that $c_- \neq 0$ is needed to define $\eta_J$ unambiguously, which is true because of Assumption 3.2.

Another setup in which geometric cross-ratios appear is the vector fixed-point equation derived in [25] for 1+1D CFT groundstates. A direct way to generalize this is to formally define $\eta_K$, such that there also exists a vector equation

$$\mathcal{K}_{\mathbb{D}}(\eta_K) |\Psi\rangle \propto |\Psi\rangle.\tag{34}$$

Here $\mathcal{K}_{\mathbb{D}}(x) \equiv x\hat{\Delta}(\mathbb{D}) + (1-x)\hat{I}(\mathbb{D})$. We emphasize that $\eta_K$ is defined as the solution(s) to Eq. (34). If the stationarity condition is satisfied, we already find a solution $\eta(\mathbb{D})$ solved from $\Delta, I$. The question is: Is it the *only* solution to Eq. (34)?

As it turns out, on a state $|\Psi\rangle$ satisfying our assumptions, these quantum cross-ratio candidates are the same as $\eta(\mathbb{D})$:

$$\boxed{\eta(\mathbb{D}) = \eta_J(\mathbb{D}) = \eta_K(\mathbb{D}).}\tag{35}$$

This is true because of the following proposition.

**Proposition 5.1** (Edge modular commutator and uniquness). *Let $|\Psi\rangle$ be a state with bulk **A1** and a non-zero chiral central charge. Suppose the stationarity condition is satisfied on a choice of $\mathbb{D}$ near the edge, then*

$$J(A, B, C)_{|\Psi\rangle} = -\frac{\pi c_-}{3} \eta(\mathbb{D})_{|\Psi\rangle},\tag{36}$$

*and $\eta(\mathbb{D})_{|\Psi\rangle}$ is the only value of $x$ that obeys $\mathcal{K}_{\mathbb{D}}(x)|\Psi\rangle \propto |\Psi\rangle$, leading to a unique vector equation*

$$\mathcal{K}_{\mathbb{D}}(\eta(\mathbb{D})) |\Psi\rangle \propto |\Psi\rangle.\tag{37}$$

**Remark.** In Eq. (36), the chiral central charge $c_-$ is the one defined in terms of the *bulk* modular commutator. If the state has zero bulk modular commutator then it immediately follows that $J(A, B, C)_{|\Psi\rangle} = 0$.

*Proof.* Without loss of generality, consider a conformal ruler $\mathbb{D}$ depicted in Fig. 9(a). We will prove our claim for these regions. Because the conformal ruler we use in this proof shall be always $\mathbb{D}$, we will simplify our notation by writing $\eta(\mathbb{D}), \eta_J(\mathbb{D}), \eta_K(\mathbb{D})$ as $\eta, \eta_J$, and $\eta_K$, respectively.

We now show that $\eta_J = \eta$. To that end, we can first derive a formula for the modular commutator for another choice of regions $(AA', B, CC')$.

$$J(AA', B, CC') = i\langle\Psi|[K_{AA'B}, K_{BCC'}]|\Psi\rangle = \frac{\pi c_-}{3}(1 - \eta_J). \tag{38}$$

The key to deriving this equation is the Markov decomposition. Let $A = A_1 A_2$, $B = B_1 B_2$, $C = C_1 C_2$, as shown in Fig. 9(b). Notice

$$\begin{aligned}
I(A_1 B_1 : A'|A_2 B_2) = 0 &\quad\Rightarrow\quad K_{AA'B}|\Psi\rangle = (K_{A_1 A_2 B_1 B_2} + K_{A' A_2 B_2} - K_{A_2 B_2})|\Psi\rangle, \\
I(C_1 B_1 : C'|C_2 B_2) = 0 &\quad\Rightarrow\quad K_{CC'B}|\Psi\rangle = (K_{C_1 C_2 B_1 B_2} + K_{C' C_2 B_2} - K_{C_2 B_2})|\Psi\rangle.
\end{aligned} \tag{39}$$

Therefore

$$\begin{aligned}
J(AA', B, CC') &= J(A_1 A_2, B_1 B_2, C_1 C_2) + J(A_2 A', B_2, C_2 C') \\
&= -\frac{\pi c_-}{3}\eta_J + \frac{\pi c_-}{3} \\
&= \frac{\pi c_-}{3}(1 - \eta_J).
\end{aligned} \tag{40}$$

In the first equality, we used the fact that $I(A : C|B) = 0 \Rightarrow J(A, B, C) = 0$ to show only two out of nine terms survive. The second line follows from the definition of $\eta_J$ in Eq. (33) and the bulk modular commutator formula.

With both $J(A, B, C)$ and $J(AA', B, CC')$ written in terms of $\eta_J$, we now use the vector fixed-point equation to show $\eta_J = \eta$. Since $\mathcal{K}_{\mathbb{D}}(\eta)|\Psi\rangle \propto |\Psi\rangle$ [Theorem 4.8], we have

$$\begin{aligned}
&\langle\Psi|[\mathcal{K}_{\mathbb{D}}(\eta), K_{BCC'}]|\Psi\rangle = 0 \\
\Rightarrow\quad &\eta\langle\Psi|[K_{AA'B}, K_{BCC'}]|\Psi\rangle + (1 - \eta)\langle\Psi|[K_{AB}, K_{BCC'}]|\Psi\rangle = 0 \\
\Rightarrow\quad &\eta(1 - \eta_J) - \eta_J(1 - \eta) = 0 \\
\Rightarrow\quad &\eta = \eta_J.
\end{aligned} \tag{41}$$

Therefore, we proved that $\eta = \eta_J$.

The derivation above indicates the uniqueness of the solution to

$$\mathcal{K}_{\mathbb{D}}(\eta_K)|\Psi\rangle \propto |\Psi\rangle. \tag{42}$$

One can simply repeat the steps in Eq. (41) with $\eta_K$ replacing $\eta$, then

$$\eta_K = \eta_J. \tag{43}$$

Therefore,

$$\boxed{\eta(\mathbb{D}) = \eta_J(\mathbb{D}) = \eta_K(\mathbb{D}).} \tag{44}$$

Note that we only used a vector equation (alternatively stationarity) at a single $\mathbb{D}$. $\qquad\square$

This result implies that, for a chiral state satisfying the stationarity condition near the edge, the "correct" $\eta$ that goes into the vector fixed-point condition

$$\mathcal{K}_{\mathbb{D}}(\eta)|\Psi\rangle \propto |\Psi\rangle, \tag{45}$$

can be computed not only using $\Delta, I$ [Eq. (17)], but also using the edge modular commutator formula [Eq. (36)]. This fact plays a key role in the proof of the consistency relations of cross-ratios in the next subsection.

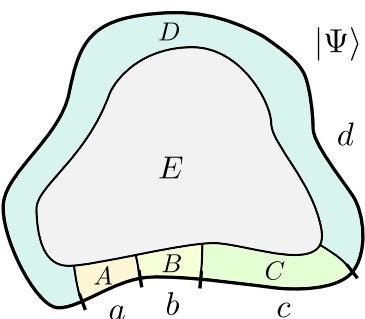

Figure 10: A disk with an edge partitioned into four intervals $a, b, c$ and $d$. The partition of the disk into $A, B, C, D, E$ is good enough for computing $\mathfrak{c}_{\text{tot}}$ and $\eta$ for any three successive intervals. Note that $\mathbb{D}(a, b, c) = (A, E, B, C, \emptyset)$ makes a legitimate conformal ruler. In the same way, we can identify conformal rulers $\mathbb{D}(b, c, d), \mathbb{D}(c, d, a), \mathbb{D}(d, a, b)$ in this figure.

## 5.2 Consistency relations of cross-ratios

In this Section, we will derive a set of consistency relations for $\eta$s for different conformal rulers, taking advantage of the results in Section 5.1. These relations turn out to be the defining properties of cross-ratios.

Throughout this derivation, we shall use the following simplified notation. Because objects such as $\Delta(AA', B, CC')$ and $I(A : C|B)$, and their operator analogs acting on the global state $|\Psi\rangle$ ($\hat{\Delta}(AA', B, CC')$ and $\hat{I}(A : C|B)$), are invariant under the deformation of the bulk region, it makes sense to specify the conformal ruler only in terms of the edge intervals. More precisely, consider a contiguous set of intervals $a, b$, and $c$ on the edge, on which the subsystems $A, B$, and $C$ are anchored. We shall denote such a conformal ruler as $\mathbb{D}(a, b, c)$. We can further define the following short-hand notations:

$$\hat{\Delta}_{a,b,c} \equiv \hat{\Delta}(\mathbb{D}(a, b, c)), \qquad \hat{I}_{a,b,c} \equiv \hat{I}(\mathbb{D}(a, b, c)), \tag{46}$$

$$\Delta_{a,b,c} \equiv \Delta(\mathbb{D}(a, b, c)), \qquad I_{a,b,c} \equiv I(\mathbb{D}(a, b, c)), \tag{47}$$

$$\mathfrak{c}_{\text{tot}}(a, b, c) \equiv \mathfrak{c}_{\text{tot}}(\mathbb{D}), \qquad \eta(a, b, c) = \eta(\mathbb{D}). \tag{48}$$

These are the conventions that we will use in this Section.

We identify two groups of consistency relations. The first group relates $\eta(a, b, c)$ to $\eta$s that involve the complement of $abc$ on the physical edge. We call it *complement relations* (Prop. 5.2). The second group of relations enables us to decompose $\eta$s on larger intervals in terms of those on smaller intervals. We refer to these as *decomposition relations* (Prop. 5.3).

**Proposition 5.2** (Complement relation)**.** *Consider a conformal ruler $\mathbb{D}$ that intersects with the physical edge at $(a, b, c)$. Let $d$ be the complement of $abc$ on the edge (see Fig. 10). If $|\Psi\rangle$ satisfies bulk **A1**, then*

$$\eta(a, b, c) = 1 - \eta(b, c, d) = \eta(a, d, c) = 1 - \eta(d, a, b), \tag{49}$$

*and*

$$\mathfrak{c}_{tot}(a, b, c) = \mathfrak{c}_{tot}(b, c, d) = \mathfrak{c}_{tot}(a, d, c) = \mathfrak{c}_{tot}(d, a, b). \tag{50}$$

*Proof.* The proof only requires bulk **A1** and the pure state condition. Due to the bulk **A1** condition, $(\mathfrak{c}_{\text{tot}}, \eta)$ depends only on the edge intervals. Pure state condition enables us to identify the entanglement entropy of a region $X$ with that of its complement, i.e., $\overline{X}: S_X = S_{\overline{X}}$. This lets us relate $\mathbb{D}(a, b, c)$ to the other conformal rulers containing the complement of $abc$, namely $\mathbb{D}(b, c, d), \mathbb{D}(c, d, a), \mathbb{D}(d, a, b), d = \overline{abc}$.

Consider the partition of the system shown in Fig. 10. We take $\mathbb{D}(a,b,c) = (A,E,B,C,\emptyset)$ and $\mathbb{D}(b,c,d) = (B,\emptyset,C,D,E)$ for an example:

$$
\begin{aligned}
I_{a,b,c} &\equiv I(A:C|B) & \Delta_{a,b,c} &\equiv \Delta(AE,B,C) \\
&= S_{AB} + S_{BC} - S_B - S_{ABC} & &= S_{ABE} + S_{BC} - S_{AE} - S_C \\
&= S_{CDE} + S_{BC} - S_B - S_{DE} & &= S_{CD} + S_{BC} - S_{BCD} - S_C \\
&= \Delta(B,C,DE) & &= I(B:D|C) \\
&\equiv \Delta_{b,c,d}, & &\equiv I_{b,c,d},
\end{aligned}
\tag{51}
$$

where the third line in both columns follows from the pure state condition. To relate $(\mathfrak{c}_{\text{tot}},\eta)$ between these two conformal rulers: Recalling $\Delta_{a,b,c} = -\frac{\mathfrak{c}_{\text{tot}}(a,b,c)}{6}\ln(\eta(a,b,c))$ and $I_{a,b,c} = -\frac{\mathfrak{c}_{\text{tot}}(a,b,c)}{6}\ln(1-\eta(a,b,c))$, we can obtain

$$
\begin{aligned}
\Delta_{a,b,c} = I_{b,c,d} &\Rightarrow \mathfrak{c}_{\text{tot}}(a,b,c)\ln(\eta(a,b,c)) = \mathfrak{c}_{\text{tot}}(b,c,d)\ln(1-\eta(b,c,d)), \\
I_{a,b,c} = \Delta_{b,c,d} &\Rightarrow \mathfrak{c}_{\text{tot}}(a,b,c)\ln(1-\eta(a,b,c)) = \mathfrak{c}_{\text{tot}}(b,c,d)\ln(\eta(b,c,d)) \\
\Rightarrow \begin{cases} \mathfrak{c}_{\text{tot}}(a,b,c) = \mathfrak{c}_{\text{tot}}(b,c,d), \\ \eta(a,b,c) = 1 - \eta(b,c,d). \end{cases}
\end{aligned}
\tag{52}
$$

Similarly, we can obtain

$$
\left.\begin{aligned} \Delta_{a,b,c} &= \Delta_{c,d,a}, \\ I_{a,d,c} &= I_{c,b,a} \end{aligned}\right\} \Rightarrow \begin{cases} \mathfrak{c}_{\text{tot}}(a,b,c) = \mathfrak{c}_{\text{tot}}(c,d,a), \\ \eta(a,b,c) = \eta(c,d,a), \end{cases}
\tag{53}
$$

$$
\left.\begin{aligned} \Delta_{a,b,c} &= I_{d,a,b}, \\ I_{a,b,c} &= I_{d,a,b} \end{aligned}\right\} \Rightarrow \begin{cases} \mathfrak{c}_{\text{tot}}(a,b,c) = \mathfrak{c}_{\text{tot}}(d,a,b), \\ \eta(a,b,c) = 1 - \eta(d,a,b). \end{cases}
\tag{54}
$$

$\square$

We remark again that the above derivation only makes use of bulk **A1** and the pure state condition. The stationarity condition and the chiral state condition are not required. On the other hand, the following part does rely on these extra assumptions.

Now we aim to derive relations that let us decompose $\eta$ on an interval to the $\eta$s on smaller sub-intervals. Any such decomposition can be broken down into a set of more elementary decompositions involving at most four intervals. More precisely, given four successive intervals $a,b,c$, and $d$, we will be able to decompose $\eta(ab,c,d), \eta(a,bc,d)$, and $\eta(a,b,cd)$ into $\eta(a,b,c)$ and $\eta(b,c,d)$ [Prop. 5.3]. This decomposition can be applied iteratively.

To compute the five aforementioned $\eta$s defined over $(a,b,c,d)$, we consider the regions shown in Fig. 11. From these regions we can make five conformal rulers

$$
\mathbb{D}(a,b,c), \quad \mathbb{D}(b,c,d), \quad \mathbb{D}(ab,c,d), \quad \mathbb{D}(a,b,cd), \quad \mathbb{D}(a,bc,d),
\tag{55}
$$

from which we can compute five cross-ratio candidates

$$
\eta(a,b,c), \quad \eta(b,c,d), \quad \eta(ab,c,d), \quad \eta(a,b,cd), \quad \eta(a,bc,d),
\tag{56}
$$

respectively.

**Proposition 5.3** (Decomposition relations). *Suppose $|\Psi\rangle$ satisfies the bulk **A1** and $\mathfrak{c}_{tot}(\mathbb{D})_{|\Psi\rangle} > 0$ is stationary for all five conformal rulers in Eq. (55) associated with Fig. 11. Then, the following*

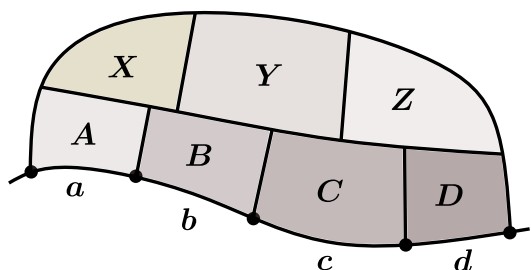

Figure 11: An edge interval partitioned into $a, b, c$ and $d$. Also shown is the associated 7-partite region $(A, X, B, Y, C, Z, D)$ near the edge, which allows us to compute various quantities and derive the decomposition relations [e.g., Prop. (59)].

*relations hold:*

$$\eta(ab, c, d) = \frac{\eta(b, c, d)}{1 - \eta(a, b, c)}, \tag{57}$$

$$\eta(a, b, cd) = \frac{\eta(a, b, c)}{1 - \eta(b, c, d)}, \tag{58}$$

$$\eta(a, bc, d) = \frac{\eta(a, b, c)\,\eta(b, c, d)}{(1 - \eta(a, b, c))(1 - \eta(b, c, d))} = \eta(ab, c, d)\eta(a, b, cd). \tag{59}$$

*Proof.* First of all, $\mathfrak{c}_{\text{tot}} > 0$ indicates $\eta \in (0, 1)$ for all the five conformal rulers. From the stationarity conditions $\delta\mathfrak{c}_{\text{tot}}(a, b, c) = 0$ and $\delta\mathfrak{c}_{\text{tot}}(b, c, d) = 0$, one can obtain the following two vector fixed-point equations [Theorem 4.8]:

$$\mathcal{K}_{\mathbb{D}(a,b,c)}(\eta(a, b, c)) |\Psi\rangle \propto |\Psi\rangle, \tag{60}$$

$$\mathcal{K}_{\mathbb{D}(b,c,d)}(\eta(b, c, d)) |\Psi\rangle \propto |\Psi\rangle. \tag{61}$$

We also know from Prop. 5.1 that[7]

$$J(AB, C, D) = i \langle\Psi| [K_{ABC}, K_{CD}] |\Psi\rangle = -\frac{\pi}{3} c_- \eta(ab, c, d), \tag{62}$$

$$J(A, B, CD) = i \langle\Psi| [K_{AB}, K_{BCD}] |\Psi\rangle = -\frac{\pi}{3} c_- \eta(a, b, cd), \tag{63}$$

$$J(A, BC, D) = i \langle\Psi| [K_{ABC}, K_{BCD}] |\Psi\rangle = -\frac{\pi}{3} c_- \eta(a, bc, d). \tag{64}$$

Using these relations, we can prove our main claim.

The key idea is to use Eq. (60) and Eq. (61) to rewrite $K_{ABC} |\Psi\rangle$ and $K_{BCD} |\Psi\rangle$ in terms of a linear combination of modular Hamiltonians over smaller regions, acting on $|\Psi\rangle$. Plugging in these expressions to Eq. (63), Eq. (62), and Eq. (64), the proof follows immediately. We discuss these in more detail below.

Using Eq. (60) and Eq. (61), the following identities follow:

$$K_{ABC} |\Psi\rangle = \left[ \frac{\eta(a, b, c)}{1 - \eta(a, b, c)} \hat{\Delta}(AX, B, CY) + K_{AB} + K_{BC} - K_B + \alpha \right] |\Psi\rangle, \tag{65}$$

$$K_{BCD} |\Psi\rangle = \left[ \frac{\eta(b, c, d)}{1 - \eta(b, c, d)} \hat{\Delta}(BY, C, DZ) + K_{BC} + K_{CD} - K_D + \beta \right] |\Psi\rangle, \tag{66}$$

where $\alpha, \beta$ are proportionality factors from Eq. (60) and Eq. (61), which are unimportant for this argument. We now use these decompositions to eliminate $K_{ABC} |\Psi\rangle$ and $K_{BCD} |\Psi\rangle$ in

---

[7]Here we utilize the stationarity conditions $\delta\mathfrak{c}_{\text{tot}}(a, b, cd) = 0$, $\delta\mathfrak{c}_{\text{tot}}(ab, c, d) = 0$ and $\delta\mathfrak{c}_{\text{tot}}(a, bc, d) = 0$.

Eq. (62), Eq. (63), Eq. (64). The result is the three relations Eqs. (57), (58), (59). Take Eq. (57) for an example. We obtain

$$i \langle \Psi | [K_{ABC}, K_{CD}] | \Psi \rangle = -\frac{\pi c_-}{3} \eta(ab, c, d)$$
$$= i \frac{\eta(a, b, c)}{1 - \eta(a, b, c)} \langle \Psi | [\hat{\Delta}(AX, B, CY), K_{CD}] | \Psi \rangle + i \langle \Psi | [K_{AB} + K_{BC} - K_B, K_{CD}] | \Psi \rangle$$
$$= \frac{\eta(a, b, c)}{1 - \eta(a, b, c)} i \langle \Psi | [K_{CYB}, K_{CD}] | \Psi \rangle + i \langle \Psi | [K_{BC}, K_{CD}] | \Psi \rangle . \tag{67}$$

Noticing that

$$i \langle \Psi | [K_{CYB}, K_{CD}] | \Psi \rangle = i \langle \Psi | [K_{BC}, K_{CD}] | \Psi \rangle = -\frac{\pi c_-}{3} \eta(b, c, d), \tag{68}$$

one can obtain

$$\eta(ab, c, d) = \frac{\eta(b, c, d)}{1 - \eta(a, b, c)} . \tag{69}$$

The other two relations Eq. (58), Eq. (59) can be obtained similarly by plugging Eq. (65) and Eq. (66) into Eq. (63) and Eq. (64). □

Let us first make some remarks about the relations among Eq. (57), Eq. (58) and Eq. (59). First, Eq. (57) can be derived from Eq. (58) and vice versa. This is due to the simple fact that $\eta(\cdot, \cdot, \cdot)$ is invariant under the exchange of the first and the third argument. Second, in order to obtain Eq. (59), one must make use of the complement relations [Prop. 5.2]. Let $e$ be the complement of $abcd$ on the edge. Because our assumptions are also satisfied near $e$, we obtain

$$\eta(cd, e, a) = \frac{\eta(d, e, a)}{1 - \eta(c, d, e)} \quad \Rightarrow \quad \eta(a, bc, d) = \eta(ab, c, d)\eta(a, b, cd), \tag{70}$$

where the $\Rightarrow$ is due to the complement relations:

$$\eta(cd, e, a) = \eta(a, b, cd), \quad \eta(a, e, d) = \eta(a, bc, d), \quad 1 - \eta(c, d, e) = \eta(ab, c, d). \tag{71}$$

What is the importance of these relations? The fact that they provide a way to relate $\eta$s among different regions is nice. However, more importantly, these relations turn out to be the defining properties of cross-ratios. This statement is a nontrivial observation made in the mathematics literature [33, 34].[8] We shall elucidate this in more detail in Section 5.3.

## 5.3 Emergence of conformal geometry

In Section 5.2, we derived two sets of relations among the quantum cross-ratios $\{\eta(a, b, c)\}$ computed for the coarse-grained intervals along the edge. Here we show that these relations allow us to construct a map $\varphi$ from the coarse-grained edge intervals to a set of intervals of a round circle, such that the quantum cross-ratios are equal to the geometric cross-ratios on the round circle [Prop. 5.4]. With this map $\varphi$, we can use the usual uniform metric on the circle to measure the sizes of the coarse-grained edge intervals.

Let us make some remarks about the setup. Recall that we are starting with a decomposition of the quantum many-body system into coarse-grained regions and intervals along the edge. We envision partitioning the edge into a set of elementary (indecomposable) intervals.

---

[8]In the literature, the cross-ratio is often defined as a function on four points, which we regard as the endpoints defining our three intervals. To make a complete specification of axioms for cross-ratio, beyond the decomposition and complement relations, one has to add further (i) symmetric property $\eta(a, b, c) = \eta(c, b, a)$, and (ii) $\eta(\emptyset, b, c) = 0, \eta(a, b, c) = 1$ if $a, b, c$ is the whole edge, both of which are indeed true for our $\eta(a, b, c)$.

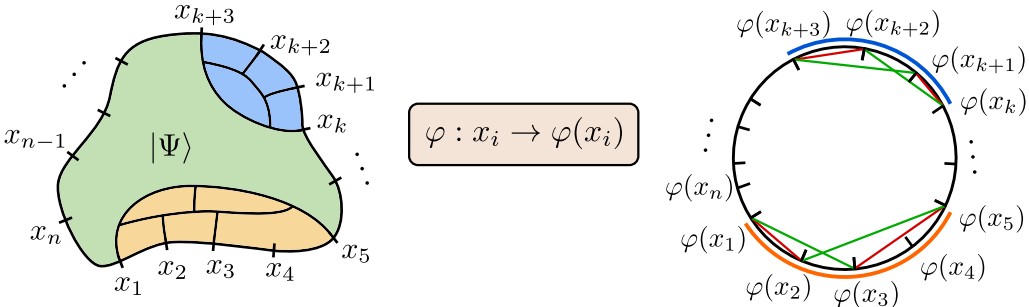

Figure 12: Mapping a chiral edge to a circle according to the cross-ratios. The interval associated with the yellow (blue) conformal ruler on the left figure measures a $[1,1,2]$-type ($[1,1,1]$-type) cross-ratio.

The intervals we consider will be the union of these elementary intervals. Without loss of generality, given three succesive intervals $a$, $b$, and $c$, we can assume that these intervals contain $n_a, n_b$, and $n_c$ elementary intervals. We shall refer to the cross-ratio defined over those intervals as the $[n_a, n_b, n_c]$-type quantum cross-ratio. Similarly, we refer to the associated conformal ruler as $[n_a, n_b, n_c]$-type conformal ruler; see Fig. 12 for an example.

An important point is that the cross-ratios of larger intervals are determined completely by the smaller subintervals. Therefore, the $[1,1,1]$-type quantum cross-ratios can be thought of as the elementary cross-ratios from which all the other quantum cross-ratios are determined. From this point of view, the map $\varphi$ can be constructed by ensuring that $\varphi$ maps the endpoints of the intervals on the edge to a set of points on the circle in such a way that all the $[1,1,1]$-type quantum cross-ratios match the corresponding geometric cross-ratios of the mapped points on the round circle. Ensuring the matching for $[1,1,1]$-type quantum cross-ratios ensures the matching of all possible cross-ratios, because those cross-ratios are determined solely from the $[1,1,1]$-type cross-ratios, using the same equation [Prop. 5.2, Prop. 5.3].

We now formally describe this statement. Without loss of generality, consider a set of elementary intervals $\{a, b, \ldots\}$ on the edge. The endpoints of these intervals are denoted as $x_1, x_2, \ldots$ in the counterclockwise order. Conversely, we may denote an interval by the endpoints. For instance, $(x_i, x_j)$ would be an interval that starts at point $x_i$, goes counterclockwise, and ends at point $x_j$.

**Proposition 5.4.** *If the quantum cross-ratios $\eta$ computed from the elementary intervals satisfy (i) $\eta \in (0,1)$ and (ii) the relations in Prop. 5.2 and Prop. 5.3, then there exists a map $\varphi$ from the set of endpoints $\{x_1, x_2, \ldots\}$ to a set of points on a circle:*

$$\varphi : x_i \to \varphi(x_i) \in S^1, \tag{72}$$

*such that the following properties hold:*

1. *The set of points $\{\varphi(x_1), \varphi(x_2), \ldots\}$ on the circle has the same orientation (e.g., counterclockwise) as $\{x_1, x_2, \ldots\}$ along the physical edge.*

2. *For any three successive intervals $(a, b, c)$, the quantum cross-ratio $\eta(a, b, c)$ is equal to the geometric cross-ratio associated with the successive intervals $(\varphi_a, \varphi_b, \varphi_c)$ on the circle, where $\varphi_a$ denotes $(\varphi(x_i), \varphi(x_j))$ for an interval $a = (x_i, x_j)$ under the map $\varphi$.*

We defer the proof to Appendix E. We briefly remark that the condition $\eta \in (0,1)$ is equivalent to the condition that $\mathfrak{c}_{\text{tot}} > 0$ [Appendix C]. If $\mathfrak{c}_{\text{tot}} = 0$, the mapping to the circle is still valid in a topological sense. This point is also explained in Appendix C.

The map $\varphi$ provides a distance measure for the intervals modulo the orientation-preserving global conformal transformation $PSL(2,\mathbb{R})$. The global conformal symmetry is manifest because $\varphi$ is constructed in terms of the cross-ratios, which are invariant under $PSL(2,\mathbb{R})$.[9]

One may wonder if the more general class of transformations, such as the orientation-preserving diffeomorphisms of $S^1$ (denoted as $\mathrm{Diff}_+(S^1)$) has any physical role. Formally, if we apply such a transformation to a chiral CFT groundstate, we obtain a special type of excited state: a coherent state [54]. The cross-ratios of these excited states will be still defined, but deformed away from their values on the groundstate.

As such, it is natural to ask whether one can apply such a transformation to a microscopic wavefunction. Because such transformations are generated by the generators of the Virasoro algebra, our recent work that constructs such generators from the groundstate wavefunction [35] can be employed for this purpose.

An interesting possibility is to use conformal rulers to measure the cross-ratios of these transformed states. More precisely, we will obtain two maps $\varphi_0$ and $\varphi_1$, each corresponding to the map from the physical edge to $S^1$, with and without the transformation. From these maps, we can construct $\varphi_1 \circ \varphi_0^{-1}$, which must be an element of $\mathrm{Diff}_+(S^1)$. Alternatively, one may compute the expectation values of the commutators of the Virasoro generators constructed via the scheme in Ref. [35], which would be a simple function of quantum cross-ratios.

## 5.4 Constant central charge and cross-ratios

By now, we have shown that a quantum state $|\Psi\rangle$ on a disk satisfying our three assumptions (bulk **A1** [Assumption 3.1], non-zero bulk modular commutator [Assumption 3.2], and stationarity condition on the edge [Assumption 4.6]) yield a set of $\eta(\mathbb{D})_{|\Psi\rangle}$ that defines a cross-ratio. This is due to the consistency relations we proved [Prop. 5.2 and 5.3]. The main purpose of this Section is to prove an interesting consequence of this result: that $\mathfrak{c}_{\mathrm{tot}}(\mathbb{D})_{|\Psi\rangle}$ is a constant.

Here is the main result of this subsection.

**Proposition 5.5** (constant $\mathfrak{c}_{\mathrm{tot}}$)**.** *If a state $|\Psi\rangle$ on a disk satisfies (i) bulk **A1**, (ii) non-zero bulk modular commutator condition, (iii) stationarity condition on the edge, then $\mathfrak{c}_{tot}(\mathbb{D})_{|\Psi\rangle}$ takes the same value for every conformal ruler $\mathbb{D}$ along the edge.*

Prop. 5.5 follows immediately from the fact that $\mathfrak{c}_{\mathrm{tot}}(\mathbb{D})_{|\Psi\rangle}$ is a constant for every $\mathbb{D}$ if and only if the set of cross-ratios over every $\mathbb{D}$ is a valid set of cross-ratios (in the sense of obeying the relations in Prop. 5.2 and 5.3).

**Proposition 5.6.** *For a state $|\Psi\rangle$ with bulk **A1** satisfied and $\mathfrak{c}_{tot}(\mathbb{D}) > 0$, $\forall \mathbb{D}$,*

$$\mathfrak{c}_{tot}(\mathbb{D}) = const. \quad \Longleftrightarrow \quad \{\eta(\mathbb{D})\} \text{ is a valid set of cross-ratios.} \tag{73}$$

*Proof.* We first prove the $\Rightarrow$ direction. First, we prove that the constant $\mathfrak{c}_{\mathrm{tot}}$ implies the complement relations [Prop. 5.2]. Consider a partition of the edge into four intervals, $a, b, c,$ and $d$ [Fig. 10]. Due to the purity of the state, we get

$$\Delta_{a,b,c} = I_{b,c,d} = \Delta_{c,d,a} = I_{d,a,b}. \tag{74}$$

Applying

$$\Delta = -\frac{\mathfrak{c}_{\mathrm{tot}}}{6}\ln(\eta), \qquad I = -\frac{\mathfrak{c}_{\mathrm{tot}}}{6}\ln(1-\eta), \tag{75}$$

for a constant $\mathfrak{c}_{\mathrm{tot}}$, one can obtain the complement relations:

$$\eta(a,b,c) = 1 - \eta(b,c,d) = \eta(c,d,a) = 1 - \eta(d,a,b). \tag{76}$$

---

[9] In general, cross-ratios are invariant under $SL(2,\mathbb{R})$ transformations [53]. In our construction, we explicitly fixed the orientation, therefore our $\varphi$ is $PSL(2,\mathbb{R})$ invariant.

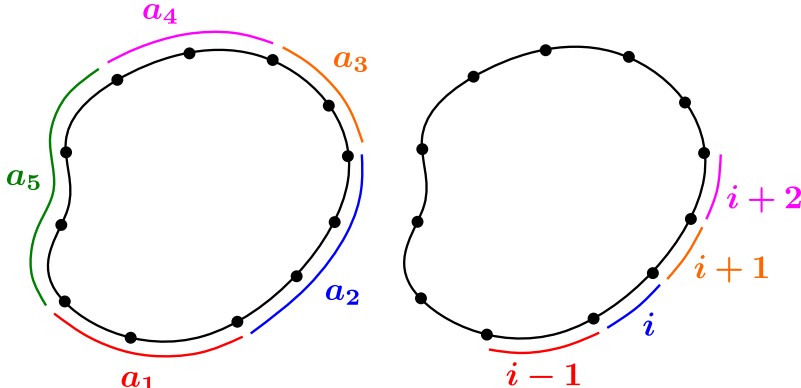

Figure 13: (Left) Five partition of the edge. We omit the conformal rulers anchored on these intervals. (Right) Choose $(a_1, a_2, a_3, a_4) = (i-1, i, i+1, i+2)$ and $a_5$ to be the complement of $(i-1) \cup i \cup (i+1) \cup (i+2)$, to conclude $\mathfrak{c}_{\text{tot}}(i-1, i, i+1) = \mathfrak{c}_{\text{tot}}(i, i+1, i+2), \forall i$.

We now prove that the constant $\mathfrak{c}_{\text{tot}}$ implies the decomposition relations [Prop. 5.3]. Consider a region shown in Fig. 11, which contains five conformal rulers:

$$\mathbb{D}(a, b, c), \quad \mathbb{D}(b, c, d), \quad \mathbb{D}(ab, c, d), \quad \mathbb{D}(a, b, cd), \quad \mathbb{D}(a, bc, d). \tag{77}$$

Using bulk **A1**, by the standard regrouping of entropy terms, we get

$$\begin{aligned}
\Delta_{a,b,cd} &= \Delta_{a,b,c} - I_{b,c,d}, \\
\Delta_{ab,c,d} &= \Delta_{b,c,d} - I_{a,b,c}, \\
\Delta_{a,bc,d} &= \Delta_{a,b,cd} + \Delta_{ab,c,d}.
\end{aligned} \tag{78}$$

Using Eq. (75) and the fact that $\mathfrak{c}_{\text{tot}}$ is a constant, one can immediately obtain the decomposition rules:

$$\eta(ab, c, d) = \frac{\eta(b, c, d)}{1 - \eta(a, b, c)}, \tag{79}$$

$$\eta(a, b, cd) = \frac{\eta(a, b, c)}{1 - \eta(b, c, d)}, \tag{80}$$

$$\eta(a, bc, d) = \frac{\eta(a, b, c)\, \eta(b, c, d)}{(1 - \eta(a, b, c))(1 - \eta(a, b, c))} = \eta(ab, c, d)\eta(a, b, cd). \tag{81}$$

We now prove the $\Leftarrow$ direction. We first partition the whole edge into five intervals, $a_1, a_2, a_3, a_4$, and $a_5$ [Fig. 13 (Left)], each interval might contain more than one coarse-grained interval. We use a short-hand notation

$$((\mathfrak{c}_{\text{tot}})_{a_i}, \eta_{a_i}) \equiv (\mathfrak{c}_{\text{tot}}(a_{i-1}, a_i, a_{i+1}), \eta(a_{i-1}, a_i, a_{i+1})),$$

where the indices are taken over values modulo 5. We first show $(\mathfrak{c}_{\text{tot}})_{a_i} = (\mathfrak{c}_{\text{tot}})_{a_j}$, $i, j = 1, 2, \ldots, 5$. By bulk **A1** and purity of the state, we can first derive

$$\begin{aligned}
&I_{a_{i-2}, a_{i-1}, a_i} + I_{a_i, a_{i+1}, a_{i+2}} = \Delta_{a_{i-1}, a_i, a_{i+1}} \\
\Rightarrow \quad &(\mathfrak{c}_{\text{tot}})_{a_{i-1}} \ln(1 - \eta_{a_{i-1}}) + (\mathfrak{c}_{\text{tot}})_{a_{i+1}} \ln(1 - \eta_{a_{i+1}}) = (\mathfrak{c}_{\text{tot}})_{a_i} \ln \eta_{a_i}.
\end{aligned} \tag{82}$$

Then, applying complement relations and decomposition relations, we can obtain

$$\eta_{a_i} = (1 - \eta_{a_{i-1}})(1 - \eta_{a_{i+1}}). \tag{83}$$

Therefore,

$$(\mathfrak{c}_{\text{tot}})_{a_i} = p_{a_i}(\mathfrak{c}_{\text{tot}})_{a_{i-1}} + (1 - p_{a_i})(\mathfrak{c}_{\text{tot}})_{a_{i+1}}, \tag{84}$$

where

$$p_{a_i} = \frac{\ln(1 - \eta_{a_{i-1}})}{\ln(1 - \eta_{a_{i-1}}) + \ln(1 - \eta_{a_{i+1}})} \in (0, 1), \tag{85}$$

for all $i = 1, 2, 3, 4, 5$. This means

$$\min\{(\mathfrak{c}_{\text{tot}})_{a_{i-1}}, (\mathfrak{c}_{\text{tot}})_{a_{i+1}}\} \le (\mathfrak{c}_{\text{tot}})_{a_i} \le \max\{(\mathfrak{c}_{\text{tot}})_{a_{i-1}}, (\mathfrak{c}_{\text{tot}})_{a_{i+1}}\}, \quad \forall i, \tag{86}$$

which constrains all $(\mathfrak{c}_{\text{tot}})_{a_i}$ to be the same.

We remind reader that the proofs above is applicable to any 5-partition of the edge interval. Now we can apply it to specific cases. Let us consider the edge has $n$ smallest coarse-grained intervals, labeled by $i = 1, 2, 3, \ldots, n$. We can choose the 5-partition to be $a_1 = i - 1, a_2 = i, a_3 = i + 1, a_4 = i + 2$ and $a_5$ be the complement of $(i - 1) \cup i \cup (i + 1) \cup (i + 2)$ on the edge [Fig. 13 (Right)]. Applying the proof above, we can conclude

$$\mathfrak{c}_{\text{tot}}(i - 1, i, i + 1) = \mathfrak{c}_{\text{tot}}(i, i + 1, i + 2), \quad \forall i = 1, \ldots, n. \tag{87}$$

That is, all the $\mathfrak{c}_{\text{tot}}$ computed on the $[1, 1, 1]$-type intervals are the same.

For $\mathfrak{c}_{\text{tot}}$ on general $[n_a, n_b, n_c]$-type intervals, it can be shown that they are equal to the $[1, 1, 1]$-type as follows. Let us consider a $[2, 1, 1]$-type $\mathfrak{c}_{\text{tot}}(ab, c, d)$ for an example.[10] By bulk **A1** and the definition of $(\mathfrak{c}_{\text{tot}}, \eta)$, we can obtain

$$\Delta_{ab,c,d} = \Delta_{b,c,d} - I_{a,b,c} \tag{88}$$
$$\Rightarrow \mathfrak{c}_{\text{tot}}(ab, c, d) \ln(\eta(ab, c, d)) = \mathfrak{c}_{\text{tot}}(b, c, d) \ln(\eta(b, c, d)) - \mathfrak{c}_{\text{tot}}(a, b, c) \ln(1 - \eta(a, b, c)).$$

Since we've shown $\mathfrak{c}_{\text{tot}}(a, b, c) = \mathfrak{c}_{\text{tot}}(b, c, d) = \mathfrak{c}_{\text{tot}}$, then applying the cross-ratio decomposition relation to the $\eta(ab, c, d)$ on the LHS, we can see

$$\mathfrak{c}_{\text{tot}}(ab, c, d) \ln \frac{\eta(b, c, d)}{1 - \eta(a, b, c)} = \mathfrak{c}_{\text{tot}} \ln \frac{\eta(b, c, d)}{1 - \eta(a, b, c)}, \tag{89}$$

which implies $\mathfrak{c}_{\text{tot}}(ab, c, d) = \mathfrak{c}_{\text{tot}}$. Applying this argument repeatedly, one can conclude $\mathfrak{c}_{\text{tot}}$ on any three contiguous intervals must be the same. $\qquad\square$

One may ask: since the cross-ratio properties of $\eta$s follow directly from $\mathfrak{c}_{\text{tot}}$ being constant, why not simply use constant $\mathfrak{c}_{\text{tot}}$ condition as the edge assumption instead of using the stationarity condition? The main reason is that the constant $\mathfrak{c}_{\text{tot}}$ condition is insufficient for the emergence of conformal symmetry. We find a non-CFT example in Appendix H.1 which satisfies the constant $\mathfrak{c}_{\text{tot}}$ condition and has a set of $\eta$s that obey the relations of the geometric cross-ratios. Therefore, a stronger assumption, such as the stationarity condition or the vector fixed-point equation, is needed. Indeed, the non-CFT example is ruled out by such assumptions.

# 6 Emergence of conformal geometry: Non-chiral edge

In this section, we generalize our analysis to non-chiral edges. That is, we drop the assumption that $c_-$ is nonzero and replace it with a different assumption [Assumption 6.1]. It should be noted that a non-chiral state can have a gapless edge described by a non-chiral CFT. A simple

---

[10]$(a, b, c, d)$ are four contiguous "elementary" coarse-grained intervals which can not be further divided into smaller ones.

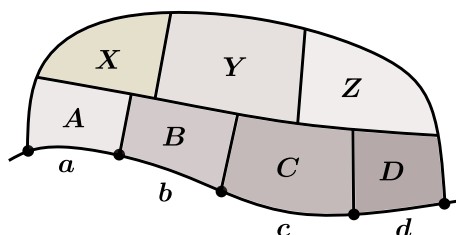

Figure 14: A 7-partite disk-like region $(A, X, B, Y, C, Z, D)$ near the edge.

example can be constructed by stacking a 1+1D non-chiral CFT on the edge of a 2+1D state with gapped boundary. In examples like this, one can add some relevant perturbations on the edge to gap out the CFT. However, there are other cases in which the non-chiral CFT at the edge cannot be gapped out in such a way, such as in the $\nu = 2/3$ fractional quantum Hall state [55]. Besides the chiral edges, the argument we present is expected to be applicable to all these cases with non-chiral gapless edges.

The main operating assumptions of this Section are the bulk **A1** [Assumption 3.1], the stationarity condition [Assumption 4.6], and an additional assumption we introduce below.

**Assumption 6.1** (Genericity condition). *For a 7-partite region $(A, X, B, Y, C, Z, D)$ of any four successive intervals $(a, b, c, d)$, of the topology shown in Fig. 14, we say a reference state $|\Psi\rangle$ satisfies the genericity condition if the following three vectors*

$$\hat{\Delta}_{a,b,c} |\Psi\rangle, \quad \hat{\Delta}_{b,c,d} |\Psi\rangle, \quad |\Psi\rangle, \tag{90}$$

*are linearly independent, where $\hat{\Delta}_{a,b,c}, \hat{\Delta}_{b,c,d}$ denotes $\hat{\Delta}(AX, B, CY)$ and $\hat{\Delta}(BY, C, DZ)$.*

Let us make some remarks on the genericity condition: Firstly, because we are working under the bulk **A1** assumption, $\hat{\Delta}(AX, B, CY)|\Psi\rangle$ and $\hat{\Delta}(BY, C, DZ)|\Psi\rangle$ are invariant under the deformations of the subsystems in the bulk (acting on the global state $|\Psi\rangle$). Therefore, we shall specify these operators in terms of the boundary intervals, i.e., $\hat{\Delta}_{a,b,c}|\Psi\rangle, \hat{\Delta}_{b,c,d}|\Psi\rangle$. Secondly, we note that a state that satisfies bulk **A1** and stationarity condition, non-zero $c_-$ from bulk modular commutators together with $\mathfrak{c}_{\text{tot}}(\mathbb{D}) > 0, \forall \mathbb{D}$ implies that the state satisfies the genericity condition [Assumption 6.1]; see Section 6.2. Because of this fact, the logical machinery we developed in this section is also applicable to the cases of chiral edges we discussed previously.

With these assumptions, we first prove the uniqueness of the cross-ratio satisfying the vector-fixed point equation.

**Proposition 6.2.** *If $\hat{\Delta} |\Psi\rangle$ is not proportional to $|\Psi\rangle$, and the solution to the equation*

$$\mathcal{K}_{\mathbb{D}}(x)|\Psi\rangle \propto |\Psi\rangle \tag{91}$$

*exists, then the solution for $x$ is unique.*

*Proof.* Suppose there are two solutions, $x$ and $y$ and $x \neq y$, s.t.

$$\left[ x\hat{\Delta} + (1-x)\hat{I} \right]|\Psi\rangle \propto |\Psi\rangle, \qquad \left[ y\hat{\Delta} + (1-y)\hat{I} \right]|\Psi\rangle \propto |\Psi\rangle. \tag{92}$$

If $x = 1$ or $y = 1$, one can directly obtain $\hat{\Delta}|\Psi\rangle \propto |\Psi\rangle$, which is a contradiction. If $x \neq 1$ and $y \neq 1$, by subtraction $(1-y)\mathcal{K}_{\mathbb{D}}(x)|\Psi\rangle - (1-x)\mathcal{K}_{\mathbb{D}}(y)|\Psi\rangle$, one can obtain

$$(x-y)\hat{\Delta}|\Psi\rangle \propto |\Psi\rangle. \tag{93}$$

As $x - y \neq 0$, we obtain $\hat{\Delta}|\Psi\rangle \propto |\Psi\rangle$, which is still a contradiction. Therefore, the solution must be unique. $\qquad\square$

Therefore, if the reference state $|\Psi\rangle$ satisfies the stationarity condition and genericity condition, there is a unique solution for $x$ to the equation $\mathcal{K}_{\mathbb{D}}(x)|\Psi\rangle \propto |\Psi\rangle$.

## 6.1 Consistency relation of cross-ratios

We now derive the consistency relations of $\eta(\mathbb{D})$. This extends the result of emergence of conformal geometry and constant $\mathfrak{c}_{\text{tot}}$ to non-chiral cases, because the proofs of these results below will not require any assumption on chirality. In fact, the original proof of complement relations (Prop. 5.2) only requires bulk **A1** and pure state condition, so it directly applies to non-chiral cases. Therefore, to prove the consistency relations of $\eta(\mathbb{D})$, we only need to derive the decomposition relations.

**Proposition 6.3** (Decomposition relations, non-chiral)**.** *Consider a 7-partite region shown Fig. 14, which contains five conformal rulers*

$$\mathbb{D}(a,b,c), \quad \mathbb{D}(b,c,d), \quad \mathbb{D}(ab,c,d), \quad \mathbb{D}(a,b,cd), \quad \mathbb{D}(a,bc,d). \tag{94}$$

*If $|\Psi\rangle$ with bulk **A1** satisfies the stationarity condition and the genericity condition 6.1, then*

$$\eta(ab,c,d) = \frac{\eta(b,c,d)}{1-\eta(a,b,c)}, \tag{95}$$

$$\eta(a,b,cd) = \frac{\eta(a,b,c)}{1-\eta(b,c,d)}, \tag{96}$$

$$\eta(a,bc,d) = \frac{\eta(a,b,c)\,\eta(b,c,d)}{(1-\eta(a,b,c))(1-\eta(a,b,c))} = \eta(ab,c,d)\eta(a,b,cd). \tag{97}$$

We sketch the main idea behind the proof of Prop. 6.3 below, leaving the detailed proof in Appendix F. Our idea is to use the consistency of the decompositions derived from vector fixed-point equations. Vector fixed-point equations allow one to decompose a modular Hamiltonian that is anchored on a large edge interval to those on smaller edge intervals when they act on the reference state. More explicitly, on a $\mathbb{D}(a,b,c)$ shown in Fig. 5, one can write

$$\tilde{K}_{ABC}\,|\Psi\rangle = \left(\frac{\eta}{1-\eta}\tilde{\Delta}_{a,b,c} + \tilde{K}_{AB} + \tilde{K}_{BC} - \tilde{K}_B\right)|\Psi\rangle\,, \tag{98}$$

where the "$\sim$" above each operator stands for the operator with its expectation value under $|\Psi\rangle$ subtracted: $\tilde{O} = O - \langle\Psi|\,O\,|\Psi\rangle$. For $\tilde{\Delta}_{a,b,c}$, we only specify the edge intervals because it is invariant under the deformation of its support in the bulk [Appendix B]. Diagrammatically, this decomposition can be represented as

$$\text{(diagram)} \tag{99}$$

where the black line segments stand for the coarse-grained intervals, a single line that goes above or below an interval stands for $\tilde{K}_X\,|\Psi\rangle$ with $X$ being a disk anchored at the interval, and the combination of red lines stands for $\tilde{\Delta}\,|\Psi\rangle$. We shall also draw the lines below the black line for; see Eq. (101) for an example. Each diagram represents the sum of all the terms appearing in the diagram, with the coefficients specified nearby.

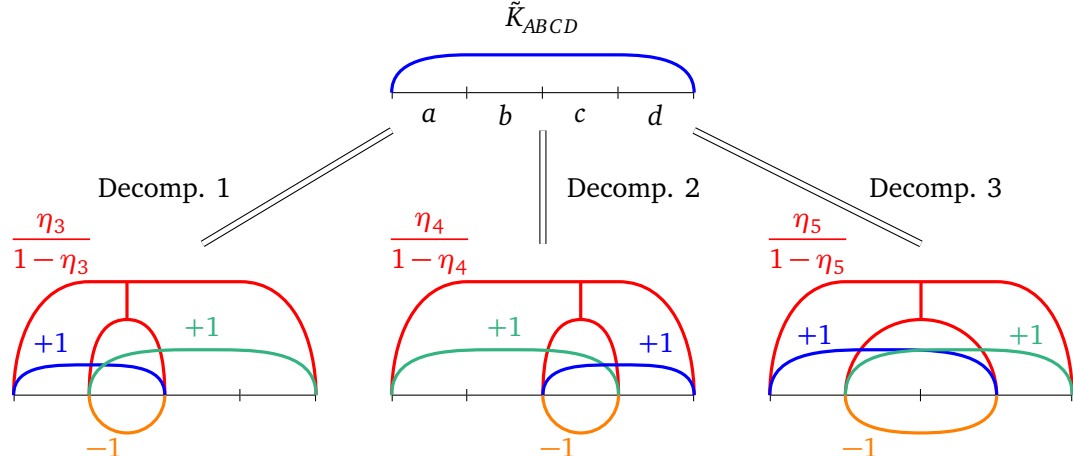

Figure 15: Decomposition of $\tilde{K}_{ABCD}|\Psi\rangle$ in three ways.

We now sketch the proof using this diagrammatic notation: On the 7-partite region shown in Fig. 14, one can write down three vector fixed-point equations on $\mathbb{D}(a,b,cd)$, $\mathbb{D}(ab,c,d)$, $\mathbb{D}(a,bc,d)$ which all allow us to decompose $\tilde{K}_{ABCD}|\Psi\rangle$ in multiple way, all of which ought to be the same. The consistency of these three results imply the cross-ratio relations in Prop. 6.3.

The computation involved can be sketched as follows. For simplicity, we index the intervals as

$$(a,b,c) \to 1, \quad (b,c,d) \to 2, \quad (a,b,cd) \to 3, \quad (ab,c,d) \to 4, \quad (a,bc,d) \to 5. \tag{100}$$

First, we use the vector fixed-point equations on $\mathbb{D}(a,b,cd)$, $\mathbb{D}(ab,c,d)$, $\mathbb{D}(a,bc,d)$ to decompose $\tilde{K}_{ABCD}$ in three ways as shown in Fig. 15. Second, we decompose $\tilde{K}_{ABC}|\Psi\rangle$ and $\tilde{K}_{BCD}|\Psi\rangle$ (denoted by the blue and green lines above the three intervals $abc$ and $bcd$) using the vector fixed-point equations on $\mathbb{D}(a,b,c)$, $\mathbb{D}(b,c,d)$ as Eq. (99). Third, we decompose $\tilde{\Delta}_{ab,c,d}|\Psi\rangle$, $\tilde{\Delta}_{ab,c,d}|\Psi\rangle$ and $\tilde{\Delta}_{ab,c,d}|\Psi\rangle$ (denoted by the combinations of red lines in Fig. 15) into $\tilde{\Delta}_{a,b,c}|\Psi\rangle$ and $\tilde{\Delta}_{b,c,d}|\Psi\rangle$ using bulk **A1** and vector fixed-point equation on $\mathbb{D}(a,b,c)$ and $\mathbb{D}(b,c,d)$. The end result of this computation is the following [Eq. (101), (102), (103)].

$$\tag{101}$$

$$\tag{102}$$

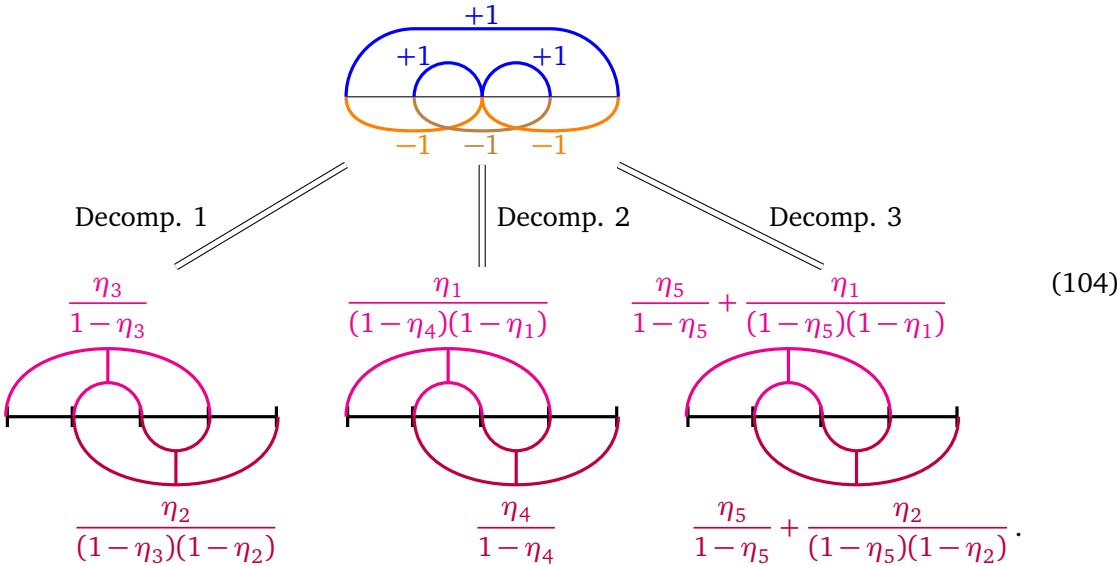

$$\tag{103}$$

Now, we successfully decompose $\tilde{K}_{ABCD}|\Psi\rangle$ into a linear combination of $\tilde{\Delta}_{a,b,c}|\Psi\rangle$, $\tilde{\Delta}_{b,c,d}|\Psi\rangle$, and some extra terms which are the same for all three decompositions. Moving these extra terms to one side of the equation, we obtain the following.

$$\tag{104}$$

The three vectors in the second row in Eq. (104) shall equal to each other, as they are all equal to the vector in the first row in Eq. (104). Since $\hat{\Delta}_{a,b,c}|\Psi\rangle$, $\hat{\Delta}_{b,c,d}|\Psi\rangle$ and $|\Psi\rangle$ are linearly independent [Assumption 6.1], the two vectors $\tilde{\Delta}_{a,b,c}|\Psi\rangle$ and $\tilde{\Delta}_{b,c,d}|\Psi\rangle$ are linearly independent, and therefore the coefficients for each of those vectors in the three decompositions should equal to each other. This leads to the following identities:

$$\frac{\eta_3}{1-\eta_3} = \frac{\eta_1}{(1-\eta_4)(1-\eta_1)} = \frac{\eta_5}{1-\eta_5} + \frac{\eta_1}{(1-\eta_5)(1-\eta_1)}, \tag{105}$$

$$\frac{\eta_2}{(1-\eta_3)(1-\eta_2)} = \frac{\eta_4}{1-\eta_4} = \frac{\eta_5}{1-\eta_5} + \frac{\eta_2}{(1-\eta_5)(1-\eta_2)}. \tag{106}$$

We can then solve for $\eta_3, \eta_4, \eta_5$:

$$\eta_3 = \frac{\eta_1}{1-\eta_2}, \qquad \eta_4 = \frac{\eta_2}{1-\eta_1}, \qquad \eta_5 = \frac{\eta_1\eta_2}{(1-\eta_1)(1-\eta_2)}. \tag{107}$$

These relations are precisely the decomposition relations in Prop. 5.3.

Thus we proved the decomposition rule of cross-ratios. Since the complement rule [Prop. 5.2] is also satisfied, we can conclude the emergence of conformal geometry, even for the non-chiral edge [Section 5.3]. In particular, the fact that $\mathfrak{c}_{\text{tot}}$ is a constant also follows.

## 6.2 Remarks on the genericity condition and nonzero $\mathfrak{c}_{\text{tot}}$

In this Section, we make several remarks on the genericity condition. Firstly, the genericity condition implies $\mathfrak{c}_{\text{tot}}(\mathbb{D}) > 0, \forall \mathbb{D}$. This is because if for a $\mathbb{D}$ that anchors at three successive

intervals $(a, b, c)$, $\mathfrak{c}_{\text{tot}}(\mathbb{D}) = 0$, then this implies $\hat{\Delta}_{a,b,c} |\Psi\rangle = 0$ or $\hat{I}_{a,b,c} |\Psi\rangle = 0$. Notice by bulk **A1** and pure state condition, $\hat{I}_{a,b,c} |\Psi\rangle = \hat{\Delta}_{b,c,d} |\Psi\rangle$, where $d$ is the complement of the interval $abc$ on the edge. Therefore, $\mathfrak{c}_{\text{tot}}(a, b, c) = 0$ implies $\hat{\Delta}_{a,b,c} |\Psi\rangle = 0$ or $\hat{\Delta}_{b,c,d} |\Psi\rangle = 0$, which violates the genericity condition.

Secondly, for a state satisfying bulk **A1** and the stationarity condition with $\mathfrak{c}_{\text{tot}}(\mathbb{D}) > 0$ for all conformal rulers $\mathbb{D}$, if $c_- \neq 0$ the genericity condition [Assumption 6.1] is satisfied. This can be proved by first computing

$$i \langle\Psi| [\hat{\Delta}_{a,b,c}, \hat{\Delta}_{b,c,d}] |\Psi\rangle = \frac{\pi c_-}{3} (1 - \eta(a, b, c) - \eta(b, c, d)), \tag{108}$$

via Prop. 5.1, where $\eta(a, b, c), \eta(b, c, d)$ are quantum cross-ratios and in the range of $(0, 1)$. Note the following fact:

$$\eta(a, b, c) + \eta(b, c, d) \neq 1. \tag{109}$$

This must be true because otherwise the following equation holds:

$$\eta(b, c, d) = 1 - \eta(a, b, c) = \eta(b, c, de) = \frac{\eta(b, c, d)}{1 - \eta(c, d, e)}, \tag{110}$$

where the second equal sign follows from the complement relation [Prop. 5.2] and the third equal sign follows from the decomposition relations [Prop. 5.3]. Eq. (110) implies $\eta(c, d, e) = 0$, which contradicts to $\mathfrak{c}_{\text{tot}}(c, d, e) > 0$. Hence Eq. (109) is proved. Therefore, if the genericity condition is violated on the region anchored at intervals $(a, b, c, d)$ in Fig. 14, the commutator in Eq. (108) should vanish, which subsequently implies that $c_- = 0$.[11] Therefore, for any state that satisfies bulk **A1**, the stationarity condition, and $\mathfrak{c}_{\text{tot}} > 0$, the genericity condition is strictly weaker than the $c_- \neq 0$ condition.

We note that the we currently do not have a simple physical motivation for the genericity condition. As such, we leave it as an open problem to provide a physical meaning to this condition. One possible alternative is the following condition. For any two thickened successive interval $(A, B)$, there exists an infinitesimal unitary transformation on $AB$, that changes $(S_A - S_B)_{|\Psi\rangle}$. This condition implies the genericity condition. (We omit the proof.) We mention this particular condition because (i) it suggests that the genericity condition is not very strong; (ii) it may be possible to relate this condition to a more physically reasonable condition.

## 7 Discussion

### 7.1 Summary

In this work, we derived the emergence of conformal geometry on gapless systems from a few locally-checkable conditions on a quantum state $|\Psi\rangle$. The physical setups include 2+1D chiral systems with an edge and also non-chiral counterparts, which can include 1+1D CFT. This work generalizes the entanglement bootstrap approach to the context of gapless systems. The bulk assumptions we took, such as bulk **A1** and nonzero bulk modular commutator, are already known [9,10]. Our main finding is an assumption about the edge from which the conformal geometry emerges: the stationarity condition ($\delta \mathfrak{c}_{\text{tot}|\Psi\rangle} = 0$). We also showed that this condition is equivalent to a locally-checkable vector fixed-point equation ($\mathcal{K}_{\mathbb{D}}(\eta)|\Psi\rangle \propto |\Psi\rangle$), similar to the one studied in [25].

What is perhaps most remarkable is that the conformal geometry emerged from these assumptions, even without putting in the distance measure. Even without making any assumptions about the distance metric or the symmetry, we obtained the set of cross-ratios, which

---

[11]We note that $c_-$ here is computed from the bulk modular commutator in Prop. 5.1.

further enable us to assign a distance measure to the physical edge up to global transformations. This was derived from our assumptions, phrased in terms of quantities that explicitly cancel out the UV contribution, leaving only the contributions from the IR.

The main workhorse behind this derivation are the quantum information-theoretic quantities we defined. We were able to define the quantum cross-ratio candidate $\eta$ and the central charge candidate $\mathfrak{c}_{\text{tot}}$ in terms of the linear combination of entanglement entropies [Eq. (1)]. Under the stationarity condition [Assumption 4.6] and the bulk assumptions [Assumption 3.1 and 3.2], we showed that three different reasonable choices of $\eta$ from the CFT point of view can be shown to be exactly the same, even without making any explicit assumption about the underlying effective field theory [Prop. 5.1]. We thus found three different ways to certify if the edge has a valid set of cross-ratios. One of these approaches, i.e., the vector fixed-point equation, is particularly useful because it implies a locality property of modular Hamiltonians. Namely, the modular Hamiltonian of a large disk that touches the edge can be decomposed into smaller pieces (at least when acting on a certain low-energy subspace).

## 7.2 Further remarks and directions

An interesting object in our work is our quantum-information theoretic definition of the "central charge" $\mathfrak{c}_{\text{tot}}$. If the wavefunction is the fixed-point with respect to the variation of this quantity ($\delta(\mathfrak{c}_{\text{tot}})_{|\Psi\rangle} = 0$), the conformal geometry emerges. Moreover, precisely under the same condition, $\mathfrak{c}_{\text{tot}}$ attains a constant value everywhere in the system. These results are evocative of the properties of the famous $c$-functions [31, 32] in 1+1D relativistic quantum field theories. On that ground, we may speculate that our $\mathfrak{c}_{\text{tot}}$ can provide further insight into states near the fixed-point.

A natural question is whether $\mathfrak{c}_{\text{tot}}$ can be a meaningful quantity in the study of RG flows. To that end, we speculate that our $\mathfrak{c}_{\text{tot}}$ is a $c$-function in a context to be made precise. If we specialize to the groundstate of a relativistic CFT in 1+1D, there is a limiting choice of intervals for which our quantity $\mathfrak{c}_{\text{tot}}$ is closely related to the RG monotone of Casini and Huerta [32] [Example 4.3]. We may ask: is $\mathfrak{c}_{\text{tot}}(a, b, c)$ monotonic under RG if the state $|\Psi\rangle$ is a relativistic QFT groundstate for every choice of $a, b, c$? How about contexts without Lorentz symmetry, including contexts considered in [56]? Can we develop a well-defined notion of RG for an edge without distance measure?

There is an intriguing analog between the equivalence between stationarity and vector fixed-point equation

$$\delta\mathfrak{c}_{\text{tot}|\Psi\rangle} = 0 \quad \Leftrightarrow \quad \mathcal{K}_{\mathbb{D}}(\eta)|\Psi\rangle \propto |\Psi\rangle, \tag{111}$$

and the well-known fact in classical mechanics: stationarity of action is equivalent to the (often vector or tensor) equation of motion associated with it. The stationarity of action was mysterious in classical mechanics (for example, it was initially motivated as "minimizing God's displeasure" [57]), but it is demystified in quantum theory: only the stationary path contributes significantly to the path integral (in the $\hbar \to 0$ limit). Can we hope for an analogous next-level understanding[12] on why $\delta\mathfrak{c}_{\text{tot}} = 0$ can be a robust phenomenon[13] for chiral edges in nature?

Furthermore, is it true that every gapped phase in 2+1D admits a conformal edge? Previously, it is believed by many [63–67] that chiral edge should be conformal, with connections

---

[12]In the context of 1+1D CFT, there are many speculations that such an equation should be the equation of motion for string theory [58–62]. That is, since 2D CFTs (with appropriate values of the central charge, including worldsheet ghosts) correspond to (perturbative) string vacua, perhaps the off-shell configuration space of (perturbative) string theory is somehow related to a 'space of quantum field theories' on the worldsheet.

[13]For chiral theory, stationarity of $\mathfrak{c}_{\text{tot}}$ does appear to be robust from our numerical study; see Appendix G. We admit that this is a puzzling surprise.

between CFT and models of chiral gapped wavefunctions in explicit models [4, 30, 68, 69].[14] Our study adds to this story by opening the door to a serious answer on how much nature likes this design and if quantum entanglement gives rise to it.

Notably, two types of "central charges" are captured by our framework, which are computable from a single wavefunction. One is the total central charge $\mathfrak{c}_{\text{tot}}$ computed from the edge, and the other is the chiral central charge $c_-$ computed from the bulk. Based on physical intuition, we would expect $\mathfrak{c}_{\text{tot}} \geq |c_-|$. A special case of this problem is to show that if $\mathfrak{c}_{\text{tot}} = 0$, then $c_- = 0$. It is desirable to prove this based on our formalism. A potential usage of $\mathfrak{c}_{\text{tot}}$ is a criterion for an ungappable edge. If at a point in the space of states where $\mathfrak{c}_{\text{tot}}$ is minimized and non-zero, this could possibly imply the edge is ungappable.

The bulk entanglement bootstrap axioms can be rephrased as the statement that the bulk reaches the global minimum of a certain entropy combination $\Delta = 0$. (Recall $\Delta \geq 0$ by strong subadditivity.) Relatedly, the stationarity condition says $\mathfrak{c}_{\text{tot}}$ defined using entropy combination near the edge is a critical point (such as a saddle point or a local extremum). This makes us wonder if entanglement bootstrap assumptions should be thought of as assumptions about critical points. Can this view suggest new generalizations to broader contexts, either gapped or gapless? Can our approach work for higher-dimensional robust gapless edges protected by a nontrivial bulk? How about finite temperature phase transitions? Generalizations to contexts with symmetries (such as charge conservation) are also foreseeable. Are there physical systems where a set of generalized cross-ratios emerges, which violate one of the rules of ordinary cross-ratio (see [34] for relaxed rules)? How about systems with a conformal boundary condition [72–74]? What happens if a chiral edge passes through a domain wall between two gapped chiral phases (such as that shown in Fig. 6 of Ref. [75])?

Deriving the emergence of conformal geometry is the first step in understanding the emergence of conformal symmetry. For future work, we would like to make progress in understanding the full dynamical structure of conformal symmetry. This includes the description of the evolution of the modular flow. The simplest context of such investigation is a "purely chiral" edge, which has only left-moving modes but not right-moving modes (phenomena related to our companion paper [35]). There we expect $|c_-| = \mathfrak{c}_{\text{tot}}$. In that context, it is possible that the edge is stationary even on coherent states obtained by applying good modular flows. We also anticipate understanding the primary states of the edge CFT. This is closely related to the consistency between the edge and the bulk. In what sense can we expect a bulk anyon to correspond to a primary field of the emergent edge CFT? Can we constrain the value of chiral central charge using bulk anyon data in our approach? We expect the other axiom **A0** and some of the machinery of bulk entanglement bootstrap to play a role in these further studies, but this problem is largely open.

## Acknowledgments

**Funding information** This work was supported in part by funds provided by the U.S. Department of Energy (D.O.E.) under cooperative research agreement DE-SC0009919, by the University of California Laboratory Fees Research Program, grant LFR-20-653926, and by the Simons Collaboration on Ultra-Quantum Matter, which is a grant from the Simons Foundation (652264, JM). IK acknowledges supports from NSF under award number PHY-2337931. JM received travel reimbursement from the Simons Foundation; the terms of this arrangement have been reviewed and approved by the University of California, San Diego, in accordance with its conflict of interest policies.

---

[14]In the context of relativistic field theory, assuming Poincaré symmetry and scale invariance in 1+1 dimensions implies conformal invariance [70] But, in $D > 1 + 1$, even in this more restricted context, the conclusion is not clear [71].

# A  Table of notations

| Notations | Meanings |
|---|---|
| $\rho_A$ | Reduced density matrix on region $A$ |
| $S(\rho_A)$ | $-\mathrm{Tr}(\rho_A \ln \rho_A)$, usually abbreviated as $S_A$ |
| $K_A$ | Modular Hamiltonian, $-\ln \rho_A$ |
| $\Delta(A,B,C)$ | $S_{AB} + S_{BC} - S_A - S_C$ |
| $\hat{\Delta}(A,B,C)$ | $K_{AB} + K_{BC} - K_A - K_C$ |
| $I(A:C|B)$ | $S_{AB} + S_{BC} - S_{ABC} - S_B$ |
| $\hat{I}(A:C|B)$ | $K_{AB} + K_{BC} - K_B - K_{ABC}$ |
| $J(A,B,C)_\rho$ | Modular commutator, defined as $i\mathrm{Tr}\big(\rho_{ABC}[K_{AB}, K_{BC}]\big)$ |
| $\mathbb{D}$ | Conformal ruler, combination of regions $(A, A', B, C, C')$ |
| $\mathbb{D}(a,b,c)$ | Conformal ruler with $A, B, C$ anchored at $a, b, c$ on the edge |
| $\Delta(\mathbb{D})$ | $\Delta(AA', B, CC')$ for $\mathbb{D} = (A, A', B, C, C')$ |
| $\hat{\Delta}(\mathbb{D})$ | $\hat{\Delta}(AA', B, CC')$ for $\mathbb{D} = (A, A', B, C, C')$ |
| $\Delta_{a,b,c}, \hat{\Delta}_{a,b,c}$ | Short-hand notation for $\Delta(\mathbb{D}(a,b,c)), \hat{\Delta}(\mathbb{D}(a,b,c))$ |
| $I(\mathbb{D})$ | $I(A:C|B)$ for $\mathbb{D} = (A, A', B, C, C')$ |
| $\hat{I}(\mathbb{D})$ | $\hat{I}(A:C|B)$ for $\mathbb{D} = (A, A', B, C, C')$ |
| $I_{a,b,c}, \hat{I}_{a,b,c}$ | Short-hand notation for $I(\mathbb{D}(a,b,c)), \hat{I}(\mathbb{D}(a,b,c))$ |
| $\mathfrak{c}_{\mathrm{tot}}(\mathbb{D})$ | Total central charge (candidate), defined in Eq. (17) |
| $\mathfrak{c}_{\mathrm{tot}}(a,b,c)$ | Short-hand notation for $\mathfrak{c}_{\mathrm{tot}}(\mathbb{D}(a,b,c))$ |
| $c_{\mathrm{tot}}$ | Total central charge of a CFT |
| $c_-$ | Chiral central charge, from the 2+1D bulk state or edge CFT |
| $\eta(\mathbb{D})$ | Quantum cross-ratio (candidate), defined in Eq. (17) |
| $\eta(a,b,c)$ | Short-hand notation for $\eta(\mathbb{D}(a,b,c))$ |
| $\eta_J$ | Cross-ratio candidate from edge modular commutators Eq. (33) |
| $\eta_K$ | Cross-ratio candidate from solving a vector equation Eq. (34) |
| $\eta_g$ | Geometric cross-ratios, see Fig. 7 |
| $\mathcal{K}_{\mathbb{D}}(x)$ | $x\hat{\Delta}(\mathbb{D}) + (1-x)\hat{I}(\mathbb{D})$ for a $\mathbb{D}$ |
| $h(x)$ | Binary entropy function: $-x\ln(x) - (1-x)\ln(1-x), x \in [0,1]$ |
| $\varphi$ | A map from the physical edge to a circle in [Prop. 5.4] |
| $\varphi_a$ | Image of an edge interval $a$ under $\varphi$ in [Prop. 5.4] |

# B  Consequences of bulk A1

In this Appendix, we first discuss invariance of the action of certain linear combinations of modular Hamiltonians acting on the reference state, under subsystem deformation in the bulk. Then we use this result to derive several vector equations which are utilized in the main text.

## B.1  Deformation invariance

Before we introduce the deformation invariance, we first derive a vector version of bulk **A1** from the usual bulk **A1** condition, namely

$$\Delta(B,C,D)_{|\Psi\rangle} = 0 \quad \Rightarrow \quad \hat{\Delta}(B,C,D)|\Psi\rangle = 0, \tag{B.1}$$

where $BCD$ is the region for bulk **A1** Fig. 3. This result directly follows from the stationarity property of $\Delta(B,C,D)_{|\Psi\rangle}$. Consider a norm-preserving perturbation $|\Psi\rangle + \epsilon|\Psi'\rangle$ for an infinitesimal $\epsilon \in \mathbb{R}$. (The norm-preserving condition implies that $\langle\Psi|\Psi'\rangle = 0$.) The linear-order variation in $\Delta(B,C,D)_{|\Psi\rangle}$ is $\epsilon\langle\Psi'|\hat{\Delta}(B,C,D)|\Psi\rangle + h.c.$, which must be zero. (Otherwise $\Delta(B,C,D)$

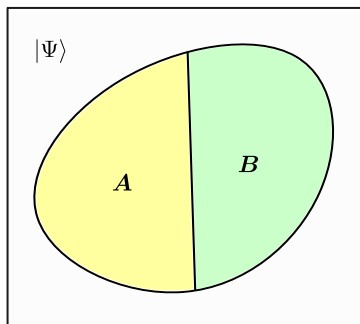
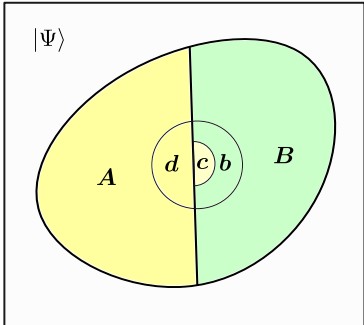

Figure 16: A bulk deformation: $A \to Ac, B \to B \setminus c$. Here we only require $c$ to be a small disk. $A$ and $B$ are not necessarily to be a disk.

may become negative, which is impossible due to SSA [36].) Choosing the perturbation as $i\epsilon|\Psi'\rangle$ instead, we get $i(\epsilon\langle\Psi'|\hat{\Delta}(B,C,D)|\Psi\rangle - h.c.) = 0$. Therefore, $\langle\Psi'|\hat{\Delta}(B,C,D)|\Psi\rangle = 0$ for any $|\Psi'\rangle$ orthogonal to $|\Psi\rangle$. We thus conclude $\hat{\Delta}(B,C,D)|\Psi\rangle \propto |\Psi\rangle$. Taking the inner product with $|\Psi\rangle$, we conclude that the constant of proportionality is zero, proving Eq. (B.1).

With this vector version bulk **A1** derived, one can derive the invariance of action of certain linear combinations of modular Hamiltonians under bulk deformations. (These are called as good modular flow generators in [35].)

We shall consider the following simplest setup to present the proof for the deformation invariance. Applying this argument to other cases is straightforward. Consider a deformation along the boundary $A \to Ac, B \to B \setminus c$. We show that $(K_A - K_B)|\Psi\rangle$ is invariant under this deformation:

$$(K_A - K_B)|\Psi\rangle = (K_{Ac} - K_{B\setminus c})|\Psi\rangle. \tag{B.2}$$

To show this, one can first partition the regions around $c$ as in Fig. 16. Note that $bcd$ is exactly the region used in the formulation of bulk **A1**, therefore

$$\Delta(b,c,d) = S_{bc} + S_{cd} - S_b - S_d = 0. \tag{B.3}$$

This implies the following Markov condition:

$$I(A \setminus d : c|d) = 0, \qquad I(B \setminus (cb) : c|b) = 0, \tag{B.4}$$

which follows from SSA [36]; see [9]. With these Markov conditions, one can write

$$K_{Ac}|\psi\rangle = (K_A + K_{dc} - K_d)|\psi\rangle, \qquad K_B|\psi\rangle = (K_{B\setminus c} + K_{cb} - K_b)|\psi\rangle. \tag{B.5}$$

Using these expressions for $K_{Ac}|\psi\rangle$ and $K_B|\psi\rangle$, one can see

$$(K_{Ac} - K_{B\setminus c})|\Psi\rangle - (K_A - K_B)|\Psi\rangle = \hat{\Delta}(b,c,d)|\Psi\rangle = 0, \tag{B.6}$$

For the last equal sign, we utilize the vector version of bulk **A1**, which, as we explained above, is a consequence of bulk **A1**.

One can apply this argument to other cases as well. For example, in the main text, we often aim to use the fact that $\hat{\Delta}(AA', B, CC')|\Psi\rangle$ and $\hat{I}(A : C|B)|\Psi\rangle$ computed from a $\mathbb{D} = (A, A', B, C, C')$ is invariant under deformations of the subsystems in the bulk [Fig. 17].

The invariance is the reason why we can use the short-hand notation

$$\hat{\Delta}_{a,b,c}|\Psi\rangle \equiv \hat{\Delta}(AA', B, CC')|\Psi\rangle, \qquad \hat{I}_{a,b,c}|\Psi\rangle \equiv \hat{I}(A : C|B)|\Psi\rangle, \tag{B.7}$$

for any conformal ruler $(A, A', B, C, C')$ anchored at three successive interval $(a, b, c)$ on the edge. The same conclusion holds true for the linear combinations of entanglement entropies $\Delta(AA', B, CC'), I(A : C|B)$ by taking an inner product with $\langle\Psi|$.

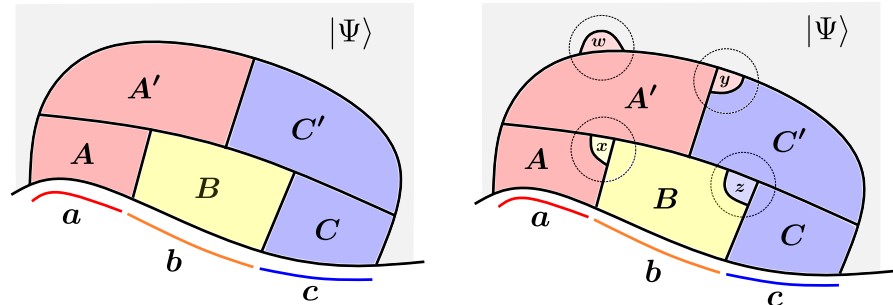

Figure 17: Four examples of bulk deformation of a $\mathbb{D}$: (i) $A \to A \setminus x, B \to Bx$; (ii) $A' \to A'w$; (iii) $A' \to A'y, C' \to C' \setminus y$; (iv) $C' \to C'z, B \to B \setminus z$.

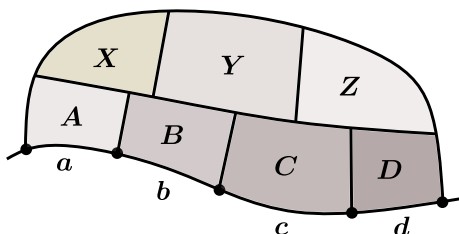

Figure 18: A combination of regions near the edge.

## B.2 Vector equations

In this Appendix, we make use of the deformation invariance property to derive several useful vector equations. These equations are used in the proof of the decomposition relation of the quantum cross-ratio [Prop. 6.3]. Their scalar versions are used in the proof of Prop. 5.6.

Consider the region shown in Fig. 18. We shall show

$$
\begin{aligned}
\hat{\Delta}_{a,b,cd} \, |\Psi\rangle &= \hat{\Delta}_{a,b,c} \, |\Psi\rangle - \hat{I}_{b,c,d} \, |\Psi\rangle \,, \\
\hat{\Delta}_{a,b,cd} \, |\Psi\rangle &= \hat{\Delta}_{a,b,c} \, |\Psi\rangle - \hat{I}_{b,c,d} \, |\Psi\rangle \,, \\
\hat{\Delta}_{a,b,cd} \, |\Psi\rangle &= \hat{\Delta}_{ab,c,d} \, |\Psi\rangle + \hat{\Delta}_{a,b,cd} \, |\Psi\rangle \,,
\end{aligned}
\tag{B.8}
$$

where the notation $\hat{\Delta}_{a,b,c}$ and $\hat{I}_{b,c,d}$ follow the convention in Eq. (B.7). This follows straight-forwardly from the bulk **A1**. We will explicitly show the first equation in Eq. (B.8). The rest follows from a similar calculation. For the first equation, we can write $\hat{\Delta}_{a,b,cd}$ explicitly from the conformal ruler $\mathbb{D} = (A, X, B, CD, YZ)$:

$$
\hat{\Delta}(AX, B, CDYZ) = \hat{\Delta}(AX, B, CY) - \hat{I}(B : ZD|CY) \,.
\tag{B.9}
$$

Note that this is an operator equation and follows from a rearrangement of the expansion of the terms on the left hand side. The $\hat{\Delta}(AX, B, CY)$ is already in the standard form of $\hat{\Delta}_{a,b,c}$ in $\mathbb{D}(A, X, B, C, Y)$. With bulk **A1**, one can deform the action of $\hat{I}(B : ZD|CY)$ into

$$
\hat{I}(B : ZD|CY) \, |\Psi\rangle = \hat{I}(B : D|C) \, |\Psi\rangle \,,
\tag{B.10}
$$

which is $\hat{I}_{b,c,d}$ for $\mathbb{D} = (B, Y, C, D, Z)$.

## C General properties of $\mathfrak{c}_{\text{tot}}$ and $\eta$

In this Appendix, we discuss some general properties about $\mathfrak{c}_{\text{tot}}$ and $\eta$, treating them as functions of two non-negative numbers $\Delta, I \geq 0$. For $\Delta, I > 0$, they are defined as the (unique)

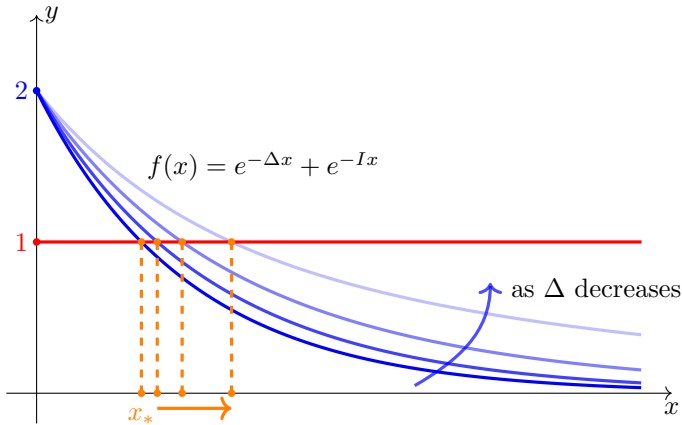

Figure 19: $c_{\text{tot}} = 6/x_*$ from the intersection point of $y = f(x)$ and $y = 1$. When $\Delta$ decreases, the curve $y = f(x)$ moves closer to the line $y = 1$, as illustrated by the group of blue curves with decreasing opacity along the blue arrow. As a result, the solution $x_*$, which is shown by the orange dot in the $x$-axis, is increasing.

solutions to the following equations

$$e^{-6\Delta/c_{\text{tot}}} + e^{-6I/c_{\text{tot}}} = 1, \tag{C.1}$$

$$\frac{\Delta}{I} = \frac{\ln \eta}{\ln(1-\eta)}. \tag{C.2}$$

We shall first discuss the properties of $c_{\text{tot}}, \eta$ for $\Delta \neq 0$ and $I \neq 0$, deferring the other cases to Appendix C.2.

## C.1   $\Delta, I > 0$ **case**

Here we study several properties of $c_{\text{tot}}$ and $\eta$, assuming $\Delta > 0$ and $I > 0$. We first study the solution $c_{\text{tot}}$ to Eq. (C.1). Define the following function.

$$f(x) = e^{-\Delta x} + e^{-Ix}. \tag{C.3}$$

Let $x_*$ be the intersection point of $y = 1$ and $y = f(x)$ [Fig. 19]. This solution must be unique because $f(x)$ decreases monotonically [Fig. 19]. Moreover, a solution exists because $f(0) = 2$ and $f(\infty) = 0$. Thus $c_{\text{tot}} = \frac{6}{x_*} > 0$ is the unique solution.

We remark that, while $I$ is fixed, $c_{\text{tot}}$ decreases as $\Delta$ increases [Fig. 19]. A similar conclusion applies when $I$ decreases while $\Delta$ is being fixed.

We now discuss an alternative definition for $c_{\text{tot}}$ and $\eta$. The definitions Eq. (C.1) and Eq. (C.2) can be restated as the solution of the following equations:

$$\Delta = \frac{c_{\text{tot}}}{6} \ln \frac{1}{\eta}, \qquad I = \frac{c_{\text{tot}}}{6} \ln \frac{1}{1-\eta}. \tag{C.4}$$

This implies

$$(\Delta - I)\eta + I = \frac{c_{\text{tot}}}{6} h(\eta), \tag{C.5}$$

where $h(\eta)$ is the binary entropy function $h(\eta) = -\eta \ln \eta - (1-\eta)\ln(1-\eta)$. We also see that

$$\Delta - I = \frac{c_{\text{tot}}}{6} [\ln \eta - \ln(1-\eta)] = \frac{c_{\text{tot}}}{6} h'(\eta), \tag{C.6}$$

where $h'(\eta)$ is the derivative of $h(\eta)$ with respect to $\eta$.

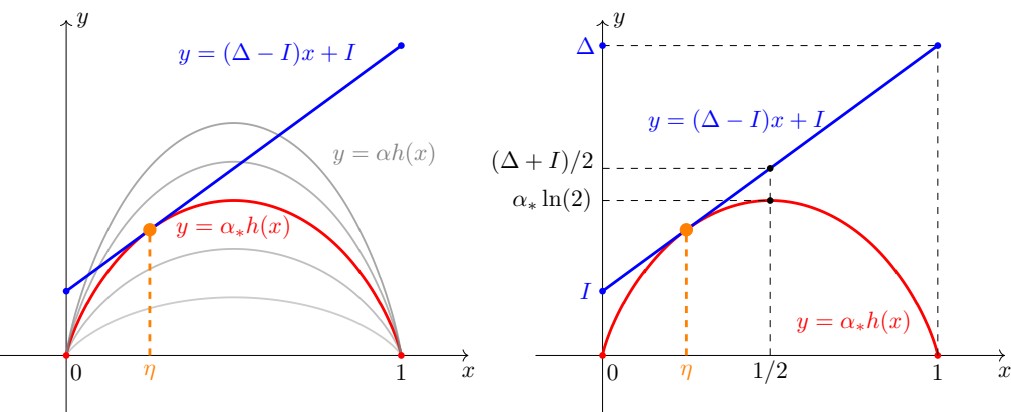

Figure 20: (Left) $\mathfrak{c}_{\text{tot}}$ and $\eta$ from the tangent point of $y = (\Delta - I)x + I$ and $y = \alpha_* h(x)$. (Right) An upper bound of $\mathfrak{c}_{\text{tot}}$ can be inferred from the plot.

Eq. (C.5) and Eq. (C.6) together imply a diagrammatic interpretation of the definition of $\mathfrak{c}_{\text{tot}}, \eta$: Given a pair $\Delta, I > 0$ consider a line $y = (\Delta - I)x + I$ and a family of curves $y = \alpha h(x)$, parameterized by $\alpha$ [Fig. 20]. One can tune $\alpha$ such that the line is tangent to the curve. When this happens, denote the parameter as $\alpha_*$. Then $\mathfrak{c}_{\text{tot}}$ is $6\alpha_*$, and $\eta$ is the $x$-coordinate of the tangent point.

As an application of this diagrammatic definition, one can see that $\mathfrak{c}_{\text{tot}}$ is upper bounded [Fig. 20]

$$\mathfrak{c}_{\text{tot}} \leq \frac{\Delta + I}{2}. \tag{C.7}$$

## C.2  $\Delta = 0$ or $I = 0$ case

We now focus on the cases where $\Delta$ or $I$ is zero. This can happen for gapped edges, such as in [76]. First, we discuss the definition of $\mathfrak{c}_{\text{tot}}$. We formally define

$$\mathfrak{c}_{\text{tot}} = 0, \quad \text{if } I = 0, \text{ or } \Delta = 0, \tag{C.8}$$

by taking the limit of the function $\mathfrak{c}_{\text{tot}}(\Delta, I)$.

More explicitly, when $\Delta = 0, I \neq 0$, $\mathfrak{c}_{\text{tot}}$ is defined as the following limit [Fig. 19]:

$$\mathfrak{c}_{\text{tot}}(0, I) \equiv \lim_{\Delta \to 0} \mathfrak{c}_{\text{tot}}(\Delta, I) = 0. \tag{C.9}$$

As $\Delta \to 0$, the solution $x_*$ to the equation $f(x) = 1$ goes to infinity and therefore $\mathfrak{c}_{\text{tot}} \to 0$. When $I = 0, \Delta \neq 0$, $\mathfrak{c}_{\text{tot}}$ is also zero under the limit

$$\mathfrak{c}_{\text{tot}}(\Delta, 0) \equiv \lim_{I \to 0} \mathfrak{c}_{\text{tot}}(\Delta, I) = 0, \tag{C.10}$$

for similar reasons. (Notice the function $\mathfrak{c}_{\text{tot}}(\Delta, I)$ is symmetric under the exchange of $\Delta$ with $I$ in the argument.) When $\Delta, I$ both are zero, the limit of $\mathfrak{c}_{\text{tot}}$ is also zero:

$$\mathfrak{c}_{\text{tot}}(0, 0) \equiv \lim_{(\Delta, I) \to (0,0)} \mathfrak{c}_{\text{tot}}(\Delta, I) = 0. \tag{C.11}$$

This limit can be proved by the upper bound Eq. (C.7).

We now discuss the values of the cross-ratio $\eta$. Here is the summary.

$$\eta(\Delta, I) = \begin{cases} 1, & \text{if } \Delta = 0, I \neq 0, \\ 0, & \text{if } \Delta \neq 0, I = 0, \\ \text{not uniquely determined}, & \text{if } \Delta = 0, I = 0. \end{cases} \tag{C.12}$$

We now discuss the limiting argument that leads to these equations.

When $\Delta = 0, I \neq 0$, $\eta$ is defined as the limit:

$$\eta(0, I) \equiv \lim_{\Delta \to 0} \eta(\Delta, I) = 1. \tag{C.13}$$

The result of the limit can be seen from the definition of $\eta$:

$$\lim_{\Delta \to 0} \frac{\Delta}{I} = \frac{\ln(\eta)}{\ln(1 - \eta)} = 0 \quad \Rightarrow \quad \eta = 1. \tag{C.14}$$

This also implies $1 - \eta = e^{-6I/\mathfrak{c}_{\text{tot}}} = 0$, from which we can see $\mathfrak{c}_{\text{tot}} = 0$. This is consistent with our conclusion about $\mathfrak{c}_{\text{tot}}$. Similarly, when $I = 0, \Delta \neq 0$, $\eta$ is defined as the limit:

$$\eta(\Delta, 0) \equiv \lim_{I \to 0} \eta(\Delta, I) = 0, \tag{C.15}$$

which is also consistent with the result $\mathfrak{c}_{\text{tot}} = 0$ because $\eta = e^{-6\Delta/\mathfrak{c}_{\text{tot}}} = 0$. When $\Delta, I$ are both zero, the limit $\lim_{(\Delta, I) \to (0,0)} \eta(\Delta, I)$ does not exist, since its value depends on the path by which $(\Delta, I)$ reaches $(0, 0)$ in the $\Delta, I$ plane. For example, when the path is along the line $(\Delta, 0)$, $\eta = 0$; while along the line $(0, I)$, $\eta = 1$. This happens if the edge is gapped, in which case the cross-ratios are undecided in the first place.

Below we relate the case of $\Delta = 0$ or $I = 0$ to physical contexts.

(i) When both $\Delta = I = 0$, the edge is gapped [22], and the cross-ratios are not uniquely decided. The fact that when $(\Delta, I) \to (0, 0)$, $\eta(\Delta, I)$ has no unique limit is consistent with this physical scenario. In fact, for any three successive intervals $(a, b, c)$, one can let $\eta(a, b, c)$ take an arbitrary value in $[0, 1]$, such that the consistency rules of cross-ratios are satisfied. This indicates that one can map the endpoints of the coarse-grained intervals to an *arbitrary set* of points on the circle that preserve the orientation. This gives a class of $\varphi$ from the edge to a circle, analogous to what we discussed in Section 5.3 and Fig. 12. The difference here is that this choice of points is arbitrary as long as the orientation is preserved.[15] This flexibility features a topological dependence rather than a geometrical dependence, and it is more flexible than the $PSL(2, R)$ in the case $\mathfrak{c}_{\text{tot}} > 0$.

(ii) One may wonder if there is a physical example for which $\Delta > 0$ but $I = 0$. One example is to consider the toric code on a square with alternating rough and smooth boundary conditions. If the conformal ruler $\mathbb{D}$ contains a point where the boundary condition changes, it gives a contribution of $\ln 2$ to $\Delta(\mathbb{D})$. This example further shows that $\mathfrak{c}_{\text{tot}} = 0$ everywhere on the edge does not imply the uniqueness of locally indistinguishable states on the physical disk.

# D  Equivalence of the stationarity condition and the vector fixed-point equation

In this Appendix, we prove that the stationarity condition and the vector fixed-point equation are equivalent [Theorem 4.8]. We begin by defining our notation and terminology. We will consider a norm-preserving perturbation of the state. We will denote objects that depend on the state $|\psi\rangle$ in the form of $\mathcal{A}(|\psi\rangle)$. The linear-order variation of this quantity is denoted as $\delta\mathcal{A}$.

We first prove the following lemma.

---

[15]Because $\eta = 0, 1$ are allowed, we may have multiple points mapped to the same point of the circle.

**Lemma D.1.** *Let $\mathcal{O}(|\psi\rangle) \equiv \langle\psi|\hat{\mathcal{O}}|\psi\rangle$ be the expectation value of a Hermitian operator $\hat{\mathcal{O}}$.*

$$\hat{\mathcal{O}}|\psi\rangle \propto |\psi\rangle\,, \tag{D.1}$$

*if and only if $\delta\mathcal{O} = 0$ for any norm-preserving perturbation of $|\psi\rangle$.*

*Proof.* We first prove the $\Leftarrow$ direction. Without loss of generality, consider a perturbation of the state $|\psi\rangle \to |\psi\rangle + \epsilon|\psi'\rangle$ for an infinitesimal $\epsilon \in \mathbb{R}$. The norm-preserving condition implies that $\langle\psi|\psi'\rangle = 0$. Note that $\delta\mathcal{O} = \epsilon\langle\psi'|\hat{\mathcal{O}}|\psi\rangle + h.c.$ We can also consider a perturbation of the form of $|\psi\rangle \to |\psi\rangle + i\epsilon|\psi'\rangle$. In this case, $\delta\mathcal{O} = i\epsilon(\langle\psi'|\hat{\mathcal{O}}|\psi\rangle - h.c.)$. The fact that both are zero implies that

$$\langle\psi'|\hat{\mathcal{O}}|\psi\rangle = 0\,, \tag{D.2}$$

for any $|\psi'\rangle$ orthogonal to $|\psi\rangle$. This immediately proves the claim.

Next, we prove the $\Rightarrow$ direction. This follows straightforwardly from the following identity:

$$\delta\mathcal{O} = \epsilon\langle\psi'|\hat{\mathcal{O}}|\psi\rangle + h.c. \tag{D.3}$$

Due to the norm-preserving condition, $|\psi'\rangle$ is orthogonal to $|\psi\rangle$. Therefore, $\delta\mathcal{O} = 0$. $\qquad\square$

The second lemma is about the linear order variation of modular Hamiltonian.

**Lemma D.2.** *Let $|\psi\rangle$ be a normalized state and $K_A$ be its modular Hamiltonian of a region $A$. For any norm-preserving perturbation of $|\psi\rangle$,*

$$\langle\psi|\,\delta K_A\,|\psi\rangle = 0\,. \tag{D.4}$$

The proof is given in Appendix A of [25], which follows from the general variation property of density matrices: $\mathrm{Tr}(\rho\,\delta(\ln\rho)) = 0$.

**Proof of Theorem 4.8:**

Now, we are in a position to prove Theorem 4.8. Consider a conformal ruler $\mathbb{D}$ and let $\Delta = \Delta(AA', B, CC')_{|\Psi\rangle}, I = I(A:C|B)_{|\Psi\rangle}$ defined in Eq. (16), and $\hat{\Delta}, \hat{I}$ be their operator version defined in Eq. (9). We remind the reader the definition of $\mathfrak{c}_{\mathrm{tot}}, \eta$ which are the solutions to the following equations.

$$e^{-6\Delta/\mathfrak{c}_{\mathrm{tot}}} + e^{-6I/\mathfrak{c}_{\mathrm{tot}}} = 1\,, \tag{D.5}$$

$$\frac{\Delta}{I} = \frac{\ln\eta}{\ln(1-\eta)}\,. \tag{D.6}$$

Let also recall that $\mathcal{K}_{\mathbb{D}}(\eta) = \eta\hat{\Delta} + (1-\eta)\hat{I}$.

We first consider the physically most interesting case of $\mathfrak{c}_{\mathrm{tot}} > 0$, which implies $\eta \in (0,1)$, we shall prove $\delta\mathfrak{c}_{\mathrm{tot}}(\mathbb{D})_{|\Psi\rangle} = 0 \Leftrightarrow \mathcal{K}_{\mathbb{D}}(\eta)|\Psi\rangle \propto |\Psi\rangle$. Under any norm-preserving perturbation $|\Psi\rangle \to |\Psi\rangle + \epsilon|\Psi'\rangle$, we can obtain

$$\delta\mathfrak{c}_{\mathrm{tot}}(\mathbb{D})_{|\Psi\rangle} = \frac{6}{h(\eta)}(\eta\delta\Delta + (1-\eta)\delta I)\,, \tag{D.7}$$

where

$$\begin{aligned}\delta\Delta &= \epsilon(\langle\Psi'|\hat{\Delta}|\Psi\rangle + \langle\Psi|\hat{\Delta}|\Psi'\rangle) + \langle\Psi|\delta\hat{\Delta}|\Psi\rangle \\ &= \epsilon(\langle\Psi'|\hat{\Delta}|\Psi\rangle + \langle\Psi|\hat{\Delta}|\Psi'\rangle)\,.\end{aligned} \tag{D.8}$$

To obtain the second line, we note that the last term $\langle\Psi|\delta\hat{\Delta}|\Psi\rangle$ in the first line vanishes [Lemma D.2]. We also note that $\delta I$ can be computed similarly. Therefore,

$$\delta\mathfrak{c}_{\mathrm{tot}} \propto \epsilon(\langle\Psi'|\mathcal{K}_{\mathbb{D}}(\eta)|\Psi\rangle + \langle\Psi|\mathcal{K}_{\mathbb{D}}(\eta)|\Psi'\rangle)\,. \tag{D.9}$$

By Lemma D.1, the right hand side of Eq. (D.9) being zero (i.e. $\delta \mathfrak{c}_{\text{tot}} = 0$) is equivalent to the condition $\mathcal{K}_{\mathbb{D}}(\eta)|\Psi\rangle \propto |\Psi\rangle$, proving the claim.

Now we discuss the $\mathfrak{c}_{\text{tot}} = 0$ case. This can be divided into three cases: (i) $\Delta = 0, I \neq 0$, (ii) $\Delta \neq 0, I = 0$, and (iii) $\Delta = I = 0$ [Appendix C]. First of all, since $\mathfrak{c}_{\text{tot}} \geq 0$, $\mathfrak{c}_{\text{tot}} = 0$ implies the first order derivative of $\mathfrak{c}_{\text{tot}}$ must vanish, therefore $\delta \mathfrak{c}_{\text{tot}} = 0$. Moreover, the vector fixed-point equation also holds: For case (i), one can obtain $\hat{\Delta}|\Psi\rangle = 0$ [Appendix B.1]. Moreover, from the definition of $\eta$ Eq. (D.6), one can see that $\eta = 1$. Therefore,

$$\mathcal{K}_{\mathbb{D}}(\eta)|\Psi\rangle = \hat{\Delta}|\Psi\rangle = 0. \tag{D.10}$$

For case (ii), it follows that $\hat{I} = 0$ [42]. Moreover, $\eta = 0$ from the definition of $\eta$. Therefore

$$\mathcal{K}_{\mathbb{D}}(\eta)|\Psi\rangle = \hat{I}|\Psi\rangle = 0. \tag{D.11}$$

For case (iii), although $\eta$ from Eq. (C.12) is not uniquely determined, both $\hat{\Delta}|\Psi\rangle$ and $\hat{I}|\Psi\rangle$ are zero. Therefore, we have a family of vector equations

$$\mathcal{K}_{\mathbb{D}}(\eta)|\Psi\rangle = (\eta\hat{\Delta} + (1-\eta)\hat{I})|\Psi\rangle = 0, \quad \forall \eta. \tag{D.12}$$

# E   Mapping the physical edge to a circle

In this Appendix, we provide a constructive proof of Prop. 5.4.

We first specify the setup and the notation: Consider a state $|\Psi\rangle$ on a disk with a set of coarse-grained intervals along the edge. Let $x_1, x_2, \ldots, x_n$ be the endpoints of these intervals, and they are labeled with a certain order. (We choose a convention where $x_1 \to x_2 \to \cdots$ is in the counterclockwise direction.) We will denote the map from the edge to the circle as $\varphi$, and the images of the endpoints $x_i$ are denoted as $\varphi(x_i)$. For the coarse-grained intervals on the edge, we will use the lower-case Roman letters to denote them, e.g., $a = (x_i, x_j)$.[16] We will sometimes use a short-hand notation of $\varphi_a$ to denote the interval $(\varphi(x_i), \varphi(x_j))$ on the circle.

In this setup, we prove the Prop. 5.4 by explicitly constructing the map $\varphi$.

*Proof.* The map can be constructed in the following steps:

- *Step 1*: On a round circle, choose three points arbitrarily, assign them as $\varphi(x_1)$, $\varphi(x_2), \varphi(x_3)$, such that $\varphi(x_1) \to \varphi(x_2) \to \varphi(x_3)$ follows counterclockwise direction.

- *Step 2*: Then, for $i = 3$ to $i = n-1$, one can find $\varphi(x_{i+1})$ on the circle one by one, by requiring that the quantum cross-ratio matches the geometric cross-ratio:

$$\eta(i-2, i-1, i) = \eta_g(\varphi_{i-2}, \varphi_{i-1}, \varphi_i). \tag{E.1}$$

Moreover, the map has the properties

1. $\varphi(x_1), \varphi(x_2), \varphi(x_3), \ldots, \varphi(x_n)$ on the circle follows the same order as $x_1, x_2, x_3, \ldots, x_n$ on the physical edge.

2. For any three successive coarse-grained intervals $(a, b, c)$, the quantum cross-ratio $\eta$ is equal to the geometric cross-ratio $\eta_g$ of three successive $(\varphi_a, \varphi_b, \varphi_c)$,

$$\eta(a, b, c) = \eta_g(\varphi_a, \varphi_b, \varphi_c). \tag{E.2}$$

---

[16]The interval starts at $x_i$, goes counterclockwise and ends at $x_j$.

We now prove both steps are doable and indeed $\varphi$ satisfies these properties:

1. Firstly, step 1 is obviously doable.

2. Then we show step 2 for $i = 3, \ldots, n-1$ is doable, which is to show one can indeed find $\varphi(x_{i+1})$ on the counterclockwise direction to $\varphi(x_i)$, such that the length of $(\varphi(x_1), \varphi(x_{i+1}))$ doesn't exceed the circumference of the circle.[17] The proof is done by induction: (i) $\varphi(x_4)$ can be found on the circle without exceeding the circle because $\eta_g(\varphi_1, \varphi_2, \varphi_3) = \eta(1, 2, 3) \in (0, 1)$. (ii) Now we show that suppose the $\varphi(x_i)$ can be put on the circle without exceeding the circle, then $\varphi(x_{i+1})$ can be put on the circle without exceeding the circle, $i = 4, \ldots, n-1$. This is because

$$\eta_g(\varphi_1, \varphi_2 \cup \cdots \cup \varphi_{i-1}, \varphi_i \cup \varphi_{i+1}) = \eta(1, 2 \cup \cdots \cup i-1, i \cup i+1) \in (0, 1). \qquad \text{(E.3)}$$

The equation above is true because both $\eta, \eta_g$ follow the same decomposition rules (Prop. 5.3), then $\eta_g(\varphi_1, \varphi_2 \cup \cdots \cup \varphi_{i-1}, \varphi_i \cup \varphi_{i+1})$ depends on $\eta_g(\varphi_j, \varphi_{j+1}, \varphi_{j+2})$ in the same way as $\eta(1, 2 \cup \cdots \cup i-1, i \cup i+1) \in (0, 1)$ depends on $\eta(j, j+1, j+2)$, $j = 1, \ldots, i-1$. Since $\eta_g(\varphi_j, \varphi_{j+1}, \varphi_{j+2}) = \eta(j, j+1, j+2)$ by construction, therefore Eq. (E.3) holds. Therefore, we finished the induction proof and show step 2 works for $i = 3, \ldots, n-1$. By now, we have shown that the construction steps are doable and have proved Property 1 by construction.

3. Now we prove the matching of the cross-ratios. As a starter, for $[1, 1, 1]$-type cross-ratio, one only need to check

$$\begin{aligned} \eta(n-2, n-1, n) &= \eta_g(\varphi_{n-2}, \varphi_{n-1}, \varphi_n), \\ \eta(n-1, n, 1) &= \eta_g(\varphi_{n-1}, \varphi_n, \varphi_1), \\ \eta(n, 1, 2) &= \eta_g(\varphi_n, \varphi_1, \varphi_2), \end{aligned} \qquad \text{(E.4)}$$

since the matching of other $[1, 1, 1]$-type cross-ratios are guaranteed by construction. To show these equations are satisfied, one can simply use the complement relation [Prop. 5.2]. We take $\eta(n-2, n-1, n) = \eta_g(\varphi_{n-2}, \varphi_{n-1}, \varphi_n)$ for an example, and the rest follows from the same idea. With the complement relation, one can first write

$$\eta(n-2, n-1, n) = 1 - \eta(1 \cup \cdots \cup n-3, n-2, n-1),$$

and

$$\eta_g(\varphi_{n-2}, \varphi_{n-1}, \varphi_n) = 1 - \eta_g(\varphi_1 \cup \cdots \cup \varphi_{n-3}, \varphi_{n-2}, \varphi_{n-1}).$$

Then

$$\eta(1 \cup \cdots \cup n-3, n-2, n-1) = \eta_g(\varphi_1 \cup \cdots \cup \varphi_{n-3}, \varphi_{n-2}, \varphi_{n-1})$$

follows from the same decomposition argument as we used for showing Eq. (E.3). By now we've shown all the $[1, 1, 1]$-type cross-ratios are matched:

$$\eta(i-2, i-1, i) = \eta_g(\varphi_{i-2}, \varphi_{i-1}, \varphi_i), \qquad \forall i = 1, \ldots, n.^{[18]} \qquad \text{(E.5)}$$

As we emphasized before, once these $[1, 1, 1]$-type cross-ratios are matched, then all the others are automatically matched, as they can be decomposed into $[1, 1, 1]$-type cross-ratios by the decomposition relations. For example, consider a $[2, 1, 1]$-type quantum

---

[17]We will abbreviate the sentence to a phrase "without exceeding the circle".

[18]The labels should be understood as values modulo $n$.

cross-ratio $\eta(ab, c, d)$:

$$
\begin{aligned}
\eta(ab, c, d) &= \frac{\eta(b, c, d)}{1 - \eta(a, b, c)} \\
&= \frac{\eta_g(\varphi_b, \varphi_c, \varphi_d)}{1 - \eta_g(\varphi_a, \varphi_b, \varphi_c)} \\
&= \eta_g(\varphi_{ab}, \varphi_c, \varphi_d).
\end{aligned}
\tag{E.6}
$$

Now we have finished the proof of property 2.

$\square$

**Remark.** Once a map $\varphi$ is obtained, it is easy to obtain a class of such maps related by $PSL(2, \mathbb{R})$ transformations of the circle. This is reflected in the first step of the construction: one can arbitrarily choose the first three points. The three real degrees of freedom specifying their three endpoints are precisely swept out by the $PSL(2, \mathbb{R})$ orbits (see *e.g.* [53] for a nice discussion).

## F  Decomposition relation of quantum cross-ratios for non-chiral states: Proof

Here we provide an proof of Prop. 6.3 which is more explicit than the diagrammatical approach in the main text. The main idea is to decompose $K_{ABCD}$ into linear combinations of modular Hamiltonians supported only on $\mathbb{D}(a, b, c)$ and $\mathbb{D}(b, c, d)$, when they are acting on $|\Psi\rangle$. There are three different decompositions and the consistency of these decompositions lead us to our results such as Eq. (95), Eq. (96) and Eq. (97).

For simplicity of the notation, we index the intervals as

$$
(a, b, c) \to 1, \quad (b, c, d) \to 2, \quad (a, b, cd) \to 3, \quad (ab, c, d) \to 4, \quad (a, bc, d) \to 5.
\tag{F.1}
$$

First, note that $K_{ABCD}$ appears in $\hat{I}_3 = \hat{I}(A : CD|B), \hat{I}_4 = \hat{I}(AB : D|C), \hat{I}_5 = \hat{I}(A : D|BC)$. Therefore, the vector fixed-point equation on $\mathbb{D}(a, b, cd), \mathbb{D}(ab, c, d), \mathbb{D}(a, bc, d)$ yields the following:

$$
K_{ABCD} |\Psi\rangle = \left[ \frac{\eta_3}{1 - \eta_3} \hat{\Delta}_3 + K_{AB} + K_{BCD} - K_B - \frac{\alpha_3}{1 - \eta_3} \right] |\Psi\rangle
\tag{F.2}
$$

$$
= \left[ \frac{\eta_4}{1 - \eta_4} \hat{\Delta}_4 + K_{ABC} + K_{CD} - K_C - \frac{\alpha_4}{1 - \eta_4} \right] |\Psi\rangle
\tag{F.3}
$$

$$
= \left[ \frac{\eta_5}{1 - \eta_5} \hat{\Delta}_5 + K_{ABC} + K_{BCD} - K_{BC} - \frac{\alpha_5}{1 - \eta_5} \right] |\Psi\rangle,
\tag{F.4}
$$

where the first, second and third line follow from $[\eta_i \hat{\Delta}_i + (1 - \eta_i) \hat{I}_i] |\Psi\rangle = \alpha_i |\Psi\rangle, i = 3, 4, 5$ respectively.

Second, using bulk **A1**, when the following operators act on $|\Psi\rangle$, they can be decomposed as [Appendix B.2]

$$
\hat{\Delta}_3 = \hat{\Delta}_1 - \hat{I}_2,
\tag{F.5}
$$

$$
\hat{\Delta}_4 = \hat{\Delta}_2 - \hat{I}_1,
\tag{F.6}
$$

$$
\hat{\Delta}_5 = \hat{\Delta}_1 + \hat{\Delta}_2 - \hat{I}_1 - \hat{I}_2.
\tag{F.7}
$$

We can then use the relation $K_{ABC} = -\hat{I}_1 + K_{AB} + K_{BC} - K_B$ and $K_{BCD} = -\hat{I}_2 + K_{CD} + K_{BC} - K_C$, which yields

$$(K_{ABCD} - K_{AB} - K_{BC} - K_{CD} + K_B + K_C)|\Psi\rangle = \left[\frac{\eta_3}{1-\eta_3}\hat{\Delta}_1 - \frac{1}{1-\eta_3}\hat{I}_2 - \frac{\alpha_3}{1-\eta_3}\right]|\Psi\rangle \tag{F.8}$$

$$= \left[\frac{\eta_4}{1-\eta_4}\hat{\Delta}_2 - \frac{1}{1-\eta_4}\hat{I}_1 - \frac{\alpha_4}{1-\eta_4}\right]|\Psi\rangle \tag{F.9}$$

$$= \left[\frac{\eta_5}{1-\eta_5}(\hat{\Delta}_1 + \hat{\Delta}_2) - \frac{1}{1-\eta_5}(\hat{I}_1 + \hat{I}_2) - \frac{\alpha_5}{1-\eta_5}\right]|\Psi\rangle. \tag{F.10}$$

Third, we can use the vector fixed-point equations on $\mathbb{D}(a,b,c)$ and $\mathbb{D}(b,c,d)$ to replace $\hat{I}_1$ and $\hat{I}_2$ by the following:

$$\hat{I}_1|\Psi\rangle = \frac{-\eta_1\hat{\Delta}_1 + \alpha_1}{1-\eta_1}|\Psi\rangle, \qquad \hat{I}_2|\Psi\rangle = \frac{-\eta_2\hat{\Delta}_2 + \alpha_2}{1-\eta_2}|\Psi\rangle. \tag{F.11}$$

Plugging these in, we get

$$(K_{ABCD} - K_{AB} - K_{BC} - K_{CD} + K_B + K_C)|\Psi\rangle$$
$$= \frac{\eta_3}{1-\eta_3}\hat{\Delta}_1|\Psi\rangle + \frac{1}{1-\eta_3}\frac{\eta_2}{1-\eta_2}\hat{\Delta}_2|\Psi\rangle - \beta_3|\Psi\rangle$$
$$= \frac{1}{1-\eta_4}\frac{\eta_1}{1-\eta_1}\hat{\Delta}_1|\Psi\rangle + \frac{\eta_4}{1-\eta_4}\hat{\Delta}_2|\Psi\rangle - \beta_4|\Psi\rangle \tag{F.12}$$
$$= \frac{1}{1-\eta_5}\left(\eta_5 + \frac{\eta_1}{1-\eta_1}\right)\hat{\Delta}_1|\Psi\rangle + \frac{1}{1-\eta_5}\left(\eta_5 + \frac{\eta_2}{1-\eta_2}\right)\hat{\Delta}_2|\Psi\rangle - \beta_5|\Psi\rangle,$$

where

$$\beta_3 = \frac{1}{1-\eta_3}\left(\alpha_3 + \frac{\alpha_2}{1-\eta_2}\right), \tag{F.13}$$

$$\beta_4 = \frac{1}{1-\eta_4}\left(\alpha_4 + \frac{\alpha_1}{1-\eta_1}\right), \tag{F.14}$$

$$\beta_5 = \frac{1}{1-\eta_5}\left(\alpha_5 + \frac{\alpha_1}{1-\eta_1} + \frac{\alpha_2}{1-\eta_2}\right). \tag{F.15}$$

Lastly, we note that $\hat{\Delta}_1|\Psi\rangle, \hat{\Delta}_2|\Psi\rangle, |\Psi\rangle$ are linearly independent [Assumption 6.1]. Therefore, if two vectors expressed as linear combination of these vectors are the same, the coefficients in front of these vectors ought to be the same. This implies

$$\frac{\eta_3}{1-\eta_3} = \frac{1}{1-\eta_4}\frac{\eta_1}{1-\eta_1} = \frac{1}{1-\eta_5}\left(\eta_5 + \frac{\eta_1}{1-\eta_1}\right), \tag{F.16}$$

$$\frac{1}{1-\eta_3}\frac{\eta_2}{1-\eta_2} = \frac{\eta_4}{1-\eta_4} = \frac{1}{1-\eta_5}\left(\eta_5 + \frac{\eta_2}{1-\eta_2}\right). \tag{F.17}$$

We can then solve for $\eta_3, \eta_4, \eta_5$:

$$\eta_3 = \frac{\eta_1}{1-\eta_2}, \qquad \eta_4 = \frac{\eta_2}{1-\eta_1}, \qquad \eta_5 = \frac{\eta_1\eta_2}{(1-\eta_1)(1-\eta_2)}. \tag{F.18}$$

We remark that the proof also implies $\beta_3 = \beta_4 = \beta_5$, which is consistent with the fact that the proportionality factor in $\mathcal{K}_{\mathbb{D}}(\eta_i)|\Psi\rangle = \alpha_i|\Psi\rangle$ is of the form $\alpha_i = \frac{c_{\text{tot}}}{6}h(\eta_i)$, where $h(\eta_i)$ being the binary entropy function. This fact might be potentially useful in studies in which some of our assumptions are modified.

# G   Numerical example: $p + ip$ superconductor with irregular edges

In this Appendix, we study the $p + ip$ superconductor (SC) on a rectangle. The translation symmetry along the edge is explicitly broken, and hence, the edge is irregular. We numerically verify the assumptions and their logical consequences, namely the emergence of the notion of cross-ratio.

## G.1   Setup

Consider the Hamiltonian for a $p + ip$ SC on a square lattice:

$$H = \sum_{\vec{r},\vec{a}} \left[ -t a_{\vec{r}}^{\dagger} a_{\vec{r}+\vec{a}} + \Delta a_{\vec{r}}^{\dagger} a_{\vec{r}+\vec{a}}^{\dagger} e^{i\vec{a}\cdot\vec{A}} + h.c. \right] - \sum_{\vec{r}} (\mu - 4t) a_{\vec{r}}^{\dagger} a_{\vec{r}}, \tag{G.1}$$

where $\vec{r} = (x, y)$ represents a site on the lattice, and $\vec{a} \in \{(1,0),(0,1)\}$ is a generator of the square lattice. We set $\vec{A} = (0, \pi/2)$, so that $e^{i\vec{a}\cdot\vec{A}}$ is either 1 or $i$. We choose $t = 1.0, \Delta = 1.0$, and $\mu = 1.3$, so that the groundstate of $H$ has a small correlation length (which is approximately 1.2 lattice spacings). We employ the open boundary condition for both $x, y$ direction, so that the system is on a rectangle and the translation symmetry along the edge is explicitly broken.

With this choice of parameters, the groundstate $|\Psi\rangle$ of $H$ satisfies our bulk assumptions [Section 3]. For one thing, it has energy gap in the bulk and the entanglement entropy of a region within the interior of the bulk satisfy an area law. Therefore, bulk **A1** is expected to be satisfied. Moreover, it has chiral central charge $c_- = 1/2$ in the bulk and a gapless chiral edge, which is robust from being gapped out by local perturbations. Therefore, the bulk modular commutator is expected to give $\frac{\pi c_-}{3}$ with $c_- = 1/2$. Indeed, we verified that these two bulk assumptions are satisfied with error decreasing with the subsystem size in our companion paper [35].[19]

At the edge, provided that it has the translational symmetry, it is natural to expect the edge to be described by a chiral CFT. However, whether the chiral CFT description remains valid in the absence of translational symmetry is less clear. We will refer to such setups as *irregular edges*. In this Appendix, we provide a strong numerical evidence that the global conformal symmetry remains intact even in such cases.

Recall that one of the main premises of our work is the stationarity condition [Assumption 4.6], which is equivalent to the vector fixed-point equation. From the vector fixed-point equation, we were able to show that the fundamental relations that define the cross-ratios emerge [Prop. 5.2 and 5.3]. Thus, we first verify that the vector fixed-point equation in Appendix G.2. In Appendix G.3, we numerically verify that the cross-ratios for irregular edges indeed satisfy the relations, as we predicted.

## G.2   Verification of the vector fixed-point equation

We first verify the edge assumption via the vector fixed-point equation

$$\mathcal{K}_{\mathbb{D}}(\eta)|\Psi\rangle \propto |\Psi\rangle, \tag{G.2}$$

on the two conformal rulers shown in Fig. 21, where $\eta$ is the quantum cross-ratio defined in Eq. (17). We also compare the $\mathfrak{c}_{\text{tot}}$s from these two conformal rulers and compare $\eta_J$ (from edge modular commutator), $\eta_K$ (from solving vector equation) with $\eta$ for each conformal ruler. These are shown to be close to each other, as expected [Prop. 5.1].

Here are more details on our numerical experiment.

---

[19]More precisely, the tests in [35] is used the cylindrical boundary condition. However, as long as the regions studied are deep inside the bulk, the results ought to be insensitive to the boundary condition.

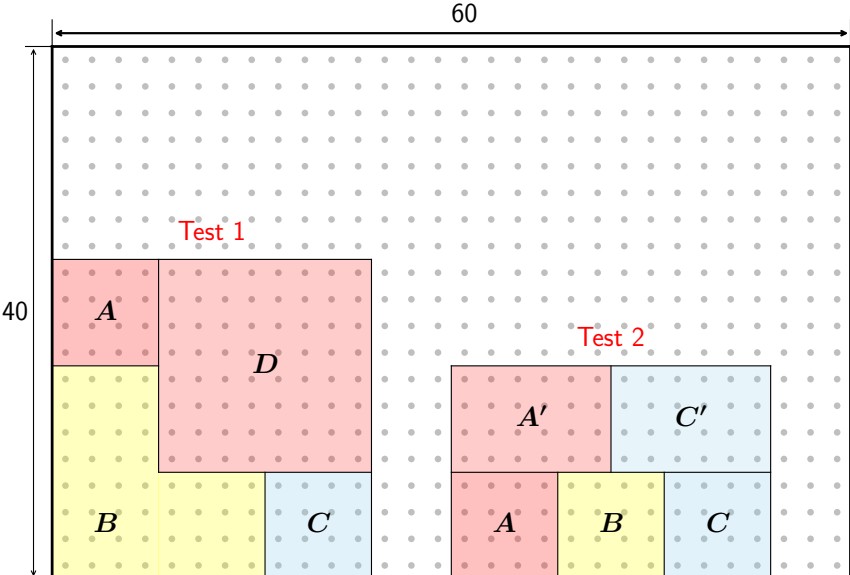

Figure 21: Two conformal rulers $\mathbb{D} = (A, D, B, C, \emptyset)$ and $\mathbb{D}(A, A', B, C, C')$ on a square lattice with open boundary condition on all four sides. Each dot in the background stands for a block of $2 \times 2$ lattice sites so that the position and size of each subsystem can be inferred.

We first test the stationarity condition utilizing the vector fixed-point equation [Definition 4.7]. On the two conformal rulers shown in Fig. 21, we computed $\mathfrak{c}_{\text{tot}}$ and $\eta$ utilizing the definition Eq. (17). To test the proportionality, we use the square root of the variance of $\mathcal{K}_{\mathbb{D}}(\eta)$ as a measure of error:

$$\sigma(\mathcal{K}_{\mathbb{D}}(\eta)) = \sqrt{\langle \Psi | \mathcal{K}_{\mathbb{D}}(\eta)^2 | \Psi \rangle - \langle \Psi | \mathcal{K}_{\mathbb{D}}(\eta) | \Psi \rangle^2} \,, \tag{G.3}$$

where

$$\mathcal{K}_{\mathbb{D}}(\eta) = \eta \hat{\Delta}(\mathbb{D}) + (1 - \eta) \hat{I}(\mathbb{D}) \,. \tag{G.4}$$

As shown in Table. 1, $\sigma(\mathcal{K}_{\mathbb{D}}) \approx 0$ in both tests, which suggests the Eq. (G.2) is approximately satisfied.

In the meantime, we can also see $\mathfrak{c}_{\text{tot}}$ computed on the two conformal rulers are approximately equal to each other, satisfying our [Prop. 5.5]. Moreover, both $\mathfrak{c}_{\text{tot}}$ are approximately $1/2$. This confirms our expectation that $\mathfrak{c}_{\text{tot}}$ shall be the total central charge of the edge CFT. The $p + ip$ SC groundstate in our test is known to have a "purely-chiral" CFT on the edge, meaning the anti-holomophic central charge $\bar{c} = 0$, and therefore the total central charge of the edge CFT is $c_{\text{tot}} = c + \bar{c} = c = 1/2$. The data in the third column of Table. 1 indeed verifies $\mathfrak{c}_{\text{tot}} \approx c_{\text{tot}}$.

The last test using this setup is the matching of $\eta, \eta_J, \eta_K$, where $\eta$ is defined in Eq. (17), $\eta_J$ is defined via edge modular commutator Eq. (33) and $\eta_K$ is defined as the solution to the vector equation Eq. (34). $\eta$ (the quantity defined in Eq. (17)) is listed in the fourth column, which has already been used to test the stationarity condition. To compute $\eta_J$, we can solve the following two systems of equations

$$\begin{aligned} \text{Test 1:} \quad & J(AD, B, C) = \frac{\pi c_-}{3}(1 - \eta_J), \quad J(A, B, C) = -\frac{\pi c_-}{3}\eta_J, \\ \text{Test 2:} \quad & J(AA', B, CC') = \frac{\pi c_-}{3}(1 - \eta_J), \quad J(A, B, C) = -\frac{\pi c_-}{3}\eta_J, \end{aligned} \tag{G.5}$$

Table 1: Results of the errors of the vector fixed-point equations $\sigma(\mathcal{K}_{\mathbb{D}}(\eta))$, total central charge $\mathfrak{c}_{tot}$ and quantum cross-ratio $\eta$ from the edge conformal rulers, $\eta_J$ from the edge modular commutators and $\eta_K$ by minimizing the errors of the vector fixed-point equations in test 1 and 2.

| Test | $\sigma(\mathcal{K}_{\mathbb{D}}(\eta))$ | $\mathfrak{c}_{tot}$ | $\eta$ | $\eta_J$ | $\eta_K$ |
|---|---|---|---|---|---|
| 1 | 0.00046 | 0.500019 | 0.046344 | 0.046346 | 0.0463 |
| 2 | 0.00475 | 0.500079 | 0.254086 | 0.254078 | 0.2541 |

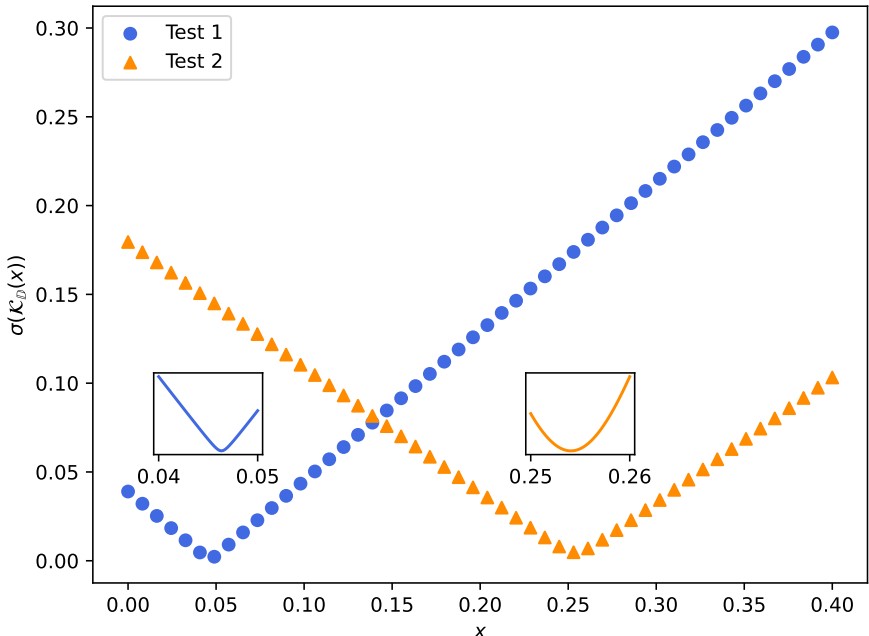

Figure 22: $\sigma(\mathcal{K}_{\mathbb{D}}(x))$ as a function of $x$. One can find an approximate solution of $\sigma(\mathcal{K}_{\mathbb{D}}(x)) = 0$ by minimizing the function. The inset is the same plot near the minima. The minima are achieved at $\eta_K = 0.0463 \pm 0.0002$ (for Test 1) and $\eta_K = 0.2541 \pm 0.0002$ (for Test 2).

for test 1 and test 2 respectively.[20] To compute $\eta_K$, we utilize the square root of the variance of $\mathcal{K}_{\mathbb{D}}(x)$ with a series of $x$, to locate the $x$ such that $\sigma(\mathcal{K}_{\mathbb{D}}(x))$ is most close to zero [Fig. 22]. That particular minimal location is the $\eta_K$. The result is listed in the last column in Table. 1. By comparing the last three columns in Table. 1, we can see the three cross-ratios are remarkably close to each other! Thus, up to a small error, $\eta = \eta_J = \eta_K$ is satisfied, in agreement with [Prop. 5.1].

## G.3 Verification of the cross-ratio relations

In this Section, we verify the consistency relation of the quantum cross-ratios. We shall focus on the decomposition relation [Prop. (5.3)].

There are two sets of subsystems we used for this purpose; see Fig. 23. The conformal ruler for the cross-ratios in each test is listed in Table. 2. In each test, we first numerically compute $\eta(a, b, c), \eta(b, c, d)$. Then, applying the decomposition relation [Prop. (5.3)], we can predict

---

[20]Another way is to use $J(A, B, C) = -\frac{\pi c_-}{3}\eta_J$ with $c_- = 1/2$. We choose not to use this way because we want to compute $\eta_J$ merely from the wavefunction, not utilizing beforehand knowledge of the state, even though the knowledge is well-known.

Table 2: Conformal rulers for quantum cross-ratio computation in the two tests.

| Test | $\mathbb{D}(a,b,c)$ | $\mathbb{D}(b,c,d)$ | $\mathbb{D}(ab,c,d)$ | $\mathbb{D}(a,b,cd)$ | $\mathbb{D}(a,bc,d)$ |
|------|---------------------|---------------------|----------------------|----------------------|----------------------|
| 1 | $(A,X,B,C,Y)$ | $(B,Y,C,D,\emptyset)$ | $(AB,X,C,D,Y)$ | $(A,X,B,CD,Y)$ | $(A,X,BC,D,Y)$ |
| 2 | $(A,X,B,C,Y)$ | $(B,Y,C,D,Z)$ | $(AB,XY,C,D,Z)$ | $(A,X,B,CD,YZ)$ | $(A,XY,BC,D,Z)$ |

Figure 23: We consider two regions for testing the consistency relations of quantum cross-ratio. Each dot in the background stands for a block of $2\times2$ lattice sites, similar to Fig. 21.

$\eta(ab,c,d)$, $\eta(a,b,cd)$, and $\eta(a,bc,d)$:

$$\eta(ab,c,d) = \frac{\eta(b,c,d)}{1-\eta(a,b,c)},$$
$$\eta(a,b,cd) = \frac{\eta(a,b,c)}{1-\eta(b,c,d)}, \tag{G.6}$$
$$\eta(a,bc,d) = \frac{\eta(a,b,c)\,\eta(b,c,d)}{(1-\eta(a,b,c))(1-\eta(a,b,c))}.$$

Then we compare these predictions and the direct numerical calculations of $\eta(ab,c,d)$, $\eta(a,b,cd)$, and $\eta(a,bc,d)$.

The results are shown in Table. 3. One can see that the predictions agree with the direct calculations with high accuracy. Note that this holds even for a highly irregular regions used in Test 1. In Test 2, one may have naively expected that $\eta(a,b,c) = \eta(b,c,d)$ using the number of lattice sites as the distance measure. However, the quantum cross-ratios have a small asymmetry [Table. 3]. Although this is small, it is still an order of magnitude larger than the discrepancy between our prediction and the direct calculation of quantum cross-ratios.

Table 3: Tests of the consistency relation of quantum cross-ratio. For each cell of $\eta(ab,c,d), \eta(a,b,cd), \eta(a,bc,d)$, the first line is directly computed from the definition of $\eta$; the second line is computed using Eq. (G.6).

| Test | $\eta(a,b,c)$ | $\eta(b,c,d)$ | $\eta(ab,c,d)$ | $\eta(a,b,cd)$ | $\eta(a,bc,d)$ |
|------|---------------|---------------|----------------|----------------|----------------|
| 1 | 0.516753 | 0.044985 | 0.093090 | 0.541094 | 0.050371 |
|   |          |          | 0.093088 | 0.541094 | 0.050369 |
| 2 | 0.232726 | 0.232739 | 0.303330 | 0.303319 | 0.092003 |
|   |          |          | 0.303332 | 0.303321 | 0.092007 |

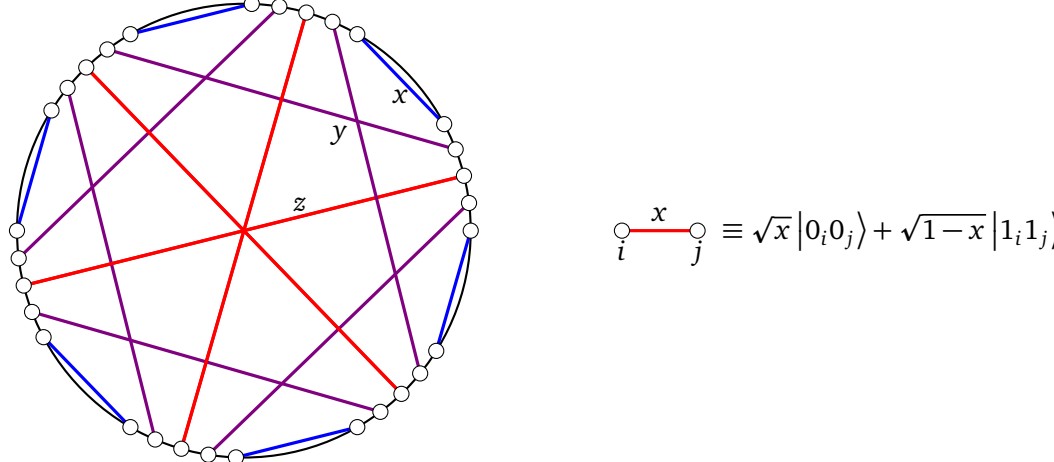

Figure 24: A state on 6 sites, each containing 5 qubits. Two qubits connected by a bond represents the specified entangled state.

This is a nontrivial evidence that supports the validity of the relations satisfied by the quantum cross-ratios at the chiral edge.

## H   Exotic examples: Non-CFT states

The main edge assumption of the main text is the stationarity condition [Assumption 4.6]. However, one may wonder if there is a weaker assumption that serves the same purpose. One example would be the assumption that $c_{\text{tot}}$ is a constant. In this Appendix, we provide counterexamples which satisfy this assumption but cannot be interpreted as CFT groundstates. In Appendix H.1, we provide such a counterexample. In Appendix H.2, we provide examples which satisfy a vector equation (weaker than the vector fixed-point equation) but yields a prediction of $\eta_K$ that cannot correspond to the geometrical cross-ratio. Therefore, certain naive attempts to formulate CFT in terms of conditions weaker than the stationarity condition do not work.

### H.1   Entropy of a state is insufficient

For a 1+1D CFT groundstate on a circle, the entanglement entropy of an interval of a chord length $\ell$ is of the form

$$S(\ell) = \frac{c_{\text{tot}}}{6} \ln\left(\frac{\ell}{\epsilon}\right), \tag{H.1}$$

where $c_{\text{tot}}$ is the total central charge and $\epsilon$ is the UV cutoff [11]. Conversely, we may ask the following question. If the entanglement entropy of any interval takes the following form

$$S(\ell) = \alpha \ln \ell + \beta, \tag{H.2}$$

where $\alpha, \beta$ are two positive constant and $\ell$ is the chord length of the interval, can we conclude the state $|\psi\rangle$ is a CFT groundstate? We show that this is not necessarily the case.

Let us start with the following simple example.

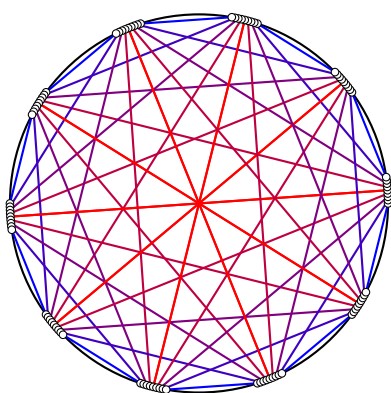

Figure 25: A state on $2N$ sites. Each site contains $2N - 1$ qubits.

**Example H.1** (six-site example)**.** We put six sites on a circle, each containing five qubits [Fig. 24]. Each white dot stands for a qubit and each colored link that connects two white dots stands for an entangled state of the form

$$\sqrt{p}\,|00\rangle + \sqrt{1-p}\,|11\rangle\,, \tag{H.3}$$

where the value of $p$ depends on the color of the edge. We call the entanglement entropy of one qubit in the entangled pair

$$\chi_p = -p\ln p - (1-p)\ln(1-p) \in [0, \ln 2]\,, \tag{H.4}$$

the *strength* of the bond. There are three types of entangled pair of strength $\chi_x, \chi_y, \chi_z$ [Fig. 24].

Due to the six-fold rotation symmetry, there are only three types of distinct intervals in this example, consisting of one, two, or three contiguous sites. The chord length of these intervals are $\ell_n = 2\sin\frac{n\pi}{6}$, where $n = 1, 2, 3$ is the number of sites in the interval. We demand the entanglement entropies of these intervals take the form of Eq. (H.2):

$$
\begin{aligned}
S(\ell_1) &= 2\chi_x + 2\chi_y + \chi_z = \alpha\ln(1) + \beta\,,\\
S(\ell_2) &= 2\chi_x + 4\chi_y + 2\chi_z = \alpha\ln(\sqrt{3}) + \beta\,,\\
S(\ell_3) &= 2\chi_x + 4\chi_y + 3\chi_z = \alpha\ln(2) + \beta\,.
\end{aligned}
\tag{H.5}
$$

The solutions to these equations are

$$
\begin{aligned}
\chi_z &= \alpha\ln\frac{2}{\sqrt{3}}\,,\\
\chi_x &= \frac{\beta - \alpha\ln\sqrt{3}}{2}\,,\\
\chi_y &= \frac{\alpha}{2}\ln\frac{3}{2}\,.
\end{aligned}
\tag{H.6}
$$

Note that $\chi_x, \chi_y, \chi_z$ must be nonnegative, which can be satisfied for some values of $\alpha$ and $\beta$. Thus we constructed a state $|\psi\rangle$ whose entanglement entropies of intervals take the same form as Eq. (H.1) in CFT groundstate, with $c_{\text{tot}} = 6\alpha$. In fact, this solution gives a tunable $c_{\text{tot}} > 0$.

Lest the reader worry that Example H.1 above is special to six sites. We note that a generalization exists; see Example H.2.

**Example H.2.** Consider the state in Fig. 25, where the circle has any even number $2N$ of sites, with $2N - 1$ qubits at each site. Each qubit is entangled with a qubit in another site, indicated by the bond in the figure. Let $\chi_k$ be the strength of the bond between sites $i$ and $i \pm k$, for $k = 1, 2, \ldots, N$, which is the same for different $i$. Therefore, the state has translation symmetry along the circle.

For an interval of $n$ sites with chord length $\ell_n = 2\sin\frac{\pi n}{N}$, we demand its entanglement entropy to be

$$S(\ell_n) = \alpha \ln\left(\frac{\ell_n}{\ell_1}\right) + \beta\,, \qquad n = 1,\dots,N\,. \tag{H.7}$$

Moreover, these entanglement entropies can be explicitly computed from the entangled pair

$$\begin{aligned}
S(\ell_1) &= \sum_{k=1}^{N-1} 2\chi_k + \chi_N\,, \\
S(\ell_n) &= nS(\ell_1) - 2\sum_{k=1}^{n-1}(n-k)\chi_k\,, \qquad n = 2,\dots,N\,.
\end{aligned} \tag{H.8}$$

Plugging Eq. (H.8) into Eq. (H.7), one obtains a system of $N$ linear equations for the $N$ variables $\chi_1,\dots,\chi_N$. The solutions are

$$\begin{aligned}
\chi_k &= \alpha \ln\left(\frac{\sin\frac{k\pi}{2N}}{\sqrt{\sin\frac{(k-1)\pi}{2N}\cdot\sin\frac{(k+1)\pi}{2N}}}\right)\,, \qquad k = 2,\dots,N\,, \\
\chi_1 &= \frac{1}{2}\left(\beta - 2\sum_{k=2}^{N-1}\chi_k - \chi_N\right)\,.
\end{aligned} \tag{H.9}$$

We see that there is a range of $\alpha, \beta > 0$ where the strength of the bonds produces a valid state, i.e., $\chi_k \in (0, \ln 2)$, $\forall k$. Again, for each given integer $N \geq 3$, we have a class of models with tunable $c_{\text{tot}} > 0$. A straightforward computation shows that the vector fixed-point equation on $[1, 1, 1]$-type intervals cannot be true because $\chi_2$ and $\chi_N$ cannot reach $\ln 2$ at the same time.

These examples cannot be CFT groundstates; in the continuum CFT, the groundstate satisfies the vector fixed-point equation [25] which these states violate. Of course, in practice, any lattice regularization will exhibit some violation of this condition. However, the important point is that we have exhibited a family of states in Example H.2 whose violation of the vector fixed point equation remains finite even in a certain thermodynamic limit. Indeed, in the limit $N \to \infty$, $\chi_k = \frac{\alpha}{2}\ln\frac{k^2}{k^2-1}, k \geq 2$, therefore the vector fixed-point equation is still violated. One could be concerned that our thermodynamic limit does not fix the size of the local Hilbert space, and that including the assumption of fixed local Hilbert space, perhaps the form of the entropy could be enough to guarantee a CFT groundstate. However, we observe that the states constructed above also exhibit a continuously tunable value of the central charge, which may be taken to be less than $1/2$ (i.e., $c_{\text{tot}}/2 < 1/2$). Thus, they cannot represent unitary CFT groundstates.

Note that the states in Example H.1 and H.2 have zero modular commutator $J(A, B, C)_{|\psi\rangle} = 0$. Thus, supplementing the correct CFT entropy with the requirement that the modular commutator for any three contiguous intervals vanishes is also insufficient.

## H.2 States with a vector equation such that $\eta_K \neq \eta$

An important equation of this work is the *vector fixed-point equation* $\mathcal{K}_{\mathbb{D}}(\eta)|\psi\rangle \propto |\psi\rangle$, where $\eta$ is the quantum cross-ratio computed from entropy combinations $I(\mathbb{D})$ and $\Delta(\mathbb{D})$ associated with a conformal ruler $\mathbb{D}$. Its importance can be seen from its equivalence to the stationarity condition (Theorem 4.8). In contexts related to chiral edges [Section 5.1] and non-chiral setups [Section 6], we found a unique solution $\eta = \eta_K$ of

$$\mathcal{K}_{\mathbb{D}}(\eta_K)|\psi\rangle \equiv \eta_K \hat{\Delta} + (1 - \eta_K)\hat{I}|\psi\rangle \propto |\psi\rangle\,. \tag{H.10}$$

In other words, the vector fixed-point equation is the unique "vector equation" of this form.

In this Appendix, we ask (1) if there are states with a vector equation Eq. (H.10), where the value $\eta_K \neq \eta$, and (2) what such state means physically. We provide two examples. Example H.3 has a family of vector equations, while Example H.4 has a vector equation with $\eta_K \neq \eta$. We argue that they are not physical CFT groundstates.

**Example H.3** (Flat entanglement spectrum states)**.** Any state with a flat entanglement spectrum has

$$K_A|\lambda\rangle \propto |\lambda\rangle, \tag{H.11}$$

for any subsystem $A$.

Such states include "absolutely maximally entangled states", which are states constructed from perfect tensor [77, 78], as well as stabilizer states [79]. Note that a state with a flat entanglement spectrum has a vanishing modular commutator for any three regions that is, $J(X, Y, Z)_{|\lambda\rangle} = 0$ identically. Suppose we arrange the qubits of such a state $|\lambda\rangle$ on a spin chain and pick three contiguous intervals $A, B, C$ from the chain. Then, the state satisfies the vector equation

$$\mathcal{K}_{\mathbb{D}}(\eta_K)|\lambda\rangle \propto |\lambda\rangle, \qquad \forall \eta_K. \tag{H.12}$$

Namely, there is a family of vector equations. Thus, one of the solutions is $\eta_K = \eta$, and thus the stationarity condition holds. Nonetheless, the genericity condition [Assumption 6.1] breaks. We argue, based on this, that such state $|\lambda\rangle$ are unrelated to CFT groundstates.

The next example is to attach a flat entanglement spectrum state to a CFT groundstate.

**Example H.4** (States with $\eta_K \neq \eta$)**.** Suppose we have a state $|\lambda\rangle$, which has a flat entanglement spectrum arranged on a spin chain. and let $|\psi\rangle$ be a CFT state on a circle. Consider the tensering state state $|\psi'\rangle = |\psi\rangle \otimes |\lambda\rangle$, such that the qubits in the state $|\lambda\rangle$ are added on top of the CFT circle. One immediately verifies that it satisfies

$$\mathcal{K}_{\mathbb{D}}(\eta_{|\psi\rangle})|\psi'\rangle \propto |\psi'\rangle, \tag{H.13}$$

where $\eta_{|\psi\rangle}$ is the quantum cross-ratio computed from the state $|\psi\rangle$. This follows from the general properties of modular Hamiltonian on tensor states, $\mathcal{K}_{\mathbb{D}}(\eta_{|\psi\rangle})|\psi\rangle \propto |\psi\rangle$ as well as Eq. (H.12). In fact, $\eta_{|\psi\rangle}$ is the only solution for vector equation. Generally, such a state $|\psi'\rangle$ does not satisfy the stationary condition. This is because $\eta_{|\psi\rangle} \neq \eta_{|\psi'\rangle}$ in general.

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
