# Peer review of "Conformal geometry from entanglement"

_SciPost Physics, doi:SciPost Phys. 18, 102 (2025)_

## Round 1 · Referee Report · Anonymous (Referee 1) · 2024-9-12

Report

I liked this paper very much. It is very original, and despite that, it is very well thought and reasoned. It is about understanding the emergence of conformal symmetry in the boundary of a gapped bulk. Building on previous results in the literature the authors make some natural conjectures about the quantum state in the gapped bulk and a new condition is imposed that has the interpretation of stationarity of the central charge. They derive from these assumptions two quantities with the interpretation of central charge and cross ratio of the emerging CFT. It is interesting that the work only uses very general tools in quantum information to derive the conformal geometry, what makes the scheme suitable to general systems and particularly suitable to computations in concrete lattice models. The work makes patent that there should be a complete and relatively simple understanding of the phenomenon in quantum information theory terms, and I think it is also interesting for developing ideas in quantum information/quantum computation itself. I recommend publication in its preset form.

Recommendation

Publish (surpasses expectations and criteria for this Journal; among top 10%)

---

## Round 1 · Referee Report · Anonymous (Referee 2) · 2024-10-31

Report

This is an interesting paper that attempts to reformulate the emergence of concepts in chiral 2d CFTs on the edge of a gapped 2+1d system purely in information-theoretic terms. The main results are a definition of the boundary central charge and the geometric cross-ratio $\frac{\ell_a \cdot \ell_b}{\ell_{ab} \cdot \ell_{bc}}$ for neighboring boundary intervals $(a,b,c)$, which are shown to satisfy consistency conditions for different ways of grouping a larger interval into three sub-intervals. They hope that this framework may be the starting point for understanding the emergence of conformal properties of chiral edge states more generally.

This is probably an exciting paper for the community of those trying to understand dynamic properties of QFTs using information theory methods. I was confused though how conformal theories could appear so generally in their analysis without having to assume Lorentz invariance from the outset - in the absence of such an assumption, couldn't one easily construct counter-examples with multiple decoupled chiral edge modes that all have different speeds of light, and therefore would not be conformal? The authors talk about potentially deriving all the Virasoro generators of the boundary, but this kind of theory would not be conformal - which of their assumptions eliminates it?

One minor weakness of the work is that their main assumption A1 is a condition that only holds in the infinite IR limit, and they do not have a quantitative description of the corrections to this expression in a low-energy expansion. Since such corrections are generically present in lattice models, comparisons with computations in such models would benefit from a principled way of parameterizing them. Can they comment on how their formalism might also include such corrections?

Recommendation

Publish (easily meets expectations and criteria for this Journal; among top 50%)

---

## Round 1 · Referee Report · Anonymous (Referee 3) · 2024-11-4

Strengths

The logic are well-explained and easy follow. This work brings a number of ideas from CFT and entanglement together in a novel way.

Weaknesses

Some of the critical assumptions in this work may seem opaque and confusing if the reader is not familiar with some of the background and motivations relying on previous (recent) work.

Report

In this paper, the authors argues that several aspects of conformal field theory and geometry can be extracted from a single wavefunction on the disk. This work is part of the so-called "entanglement bootstrap" program that studies states satisfying a number of locally checkable conditions, extracting universal properties of such states.

As I understand it, the motivation for the work is as follows: For three contaguous partitions A,B,C in a CFT, using the standard entropy formula: I = (ctot/6) ln(1/(1-eta)), Delta = (ctot/6) ln(eta), where eta = |A| |C| / |AB| |BC| is the cross-ratio. Here the authors inverts this idea. From I and Delta, the authors define ctot and eta, which effective defines the edge geometry up to a mobius transformation. Then they show consistency of the definitions, such as cross-ratio via different definitions (5.1), and different rulers (5.2).

I have a number of questions/issues with the manuscript.

  • Ref [25] argues that |psi> is stationary when K[D] is applied. The discussion on the stationary condition is motivated/relies heavily on previous work. This manuscript can be substantially improved by providing CFT motivations for the assumption.

  • It is possible for ctot to change along the boundary by adding domain walls or modifying boundary Hamiltonian. Trivially, any pair of opposite chiral theories can be gapped out within a region. I would guess that the staionary assumption is broken on the edge. Is this true and how does such assumption fail?

  • What happens to with a non-conformal boundary to a chiral state? For instance, the p+ip superconductor can be modified such that its edge has a p^3 dispersion. Again, what assumptions are violated with such edge?

Overall the paper appears technically sound and advances the entanglement bootstrap program to include critical states. The presentation is good overall, but some of the technical formula / assumptions can use more explanations to motivate their form. Perhaps the authors can provide a few more counterexamples of how the assumptions can fail for physical bulk/edges.

Requested changes

See report

Recommendation

Ask for minor revision

---

## Round 2 · Author Response

Response to referee 1:

We are grateful to the referee for the encouraging remarks.

Response to referee 2:

We are grateful to the referee for the encouraging remarks. We also thank the referee for raising some interesting questions. Below we list the questions and our replies:

1- " I was confused though how conformal theories could appear so generally in their analysis without having to assume Lorentz invariance from the outset - in the absence of such an assumption, couldn't one easily construct counter-examples with multiple decoupled chiral edge modes that all have different speeds of light, and therefore would not be conformal? The authors talk about potentially deriving all the Virasoro generators of the boundary, but this kind of theory would not be conformal - which of their assumptions eliminates it?"

We share the same surprise that the conformal geometry just arises from such simple and general assumptions, and in particular that these assumptions do not involve any symmetries. This is one of the reasons that gets us excited about the results. The assumptions we posit only concern \emph{universal properties} of phases of matter or critical theories \emph{that are reflected on the corresponding representative states}. For the possible counter-example the referee mentioned, it is true that the Hamiltonian for such a system isn't conformally invariant. However, we anticipate its groundstate could still be a representative state with a conformal field theory on the edge, and our assumptions do not rule out such a state. For example, consider two decoupled chiral systems with dispersion relation $\epsilon(p_1) = v_1 p_1$, $\epsilon(p_2) = v_2 p_2$, with $v_1\neq v_2$; the groundstate of such total system is a stacking of the groundstates of the two individual systems. However, these two groundstates are the same state, and hence their stacking still has conformal symmetry. In summary: The speed of light, which depends on the Hamiltonian, is not a universal property and is not encoded in its groundstate. Therefore, stacking of such states won't cause violation of our assumptions.

2- "One minor weakness of the work is that their main assumption A1 is a condition that only holds in the infinite IR limit, and they do not have a quantitative description of the corrections to this expression in a low-energy expansion. Since such corrections are generically present in lattice models, comparisons with computations in such models would benefit from a principled way of parameterizing them. Can they comment on how their formalism might also include such corrections?"

This is an important question that is currently under investigation by us and other groups (for example, this is the motivation for Kitaev's recent paper arXiv:2405.02434). Indeed, we do not have a general quantitative description of the violation of A1. However, for some specific lattice models (the p+ip superconductor), in our recent work (arXiv:2403.18410), we've found that the violation of A1 decays either exponentially or algebraically as a function of the subsystem size in different parts of the phase diagram. To complete our formalism, we wish to show a general result, whose physical statement can be roughly formulated as follows: if the quantum state is ``close enough'' to a zero-correlation-length RG fixed-point, then the violation of A1 will decrease to zero as the state approaches the fixed point. We also remark that there is a similar to-be-shown statement for the stationarity assumption. In the revision, we have tried to make it clearer that this is an important open problem.

Response to referee 3:

We thank the referee for bringing up these questions. Our point-by-point responses are the following: 1- " Ref [25] argues that $|\psi\rangle$ is stationary when $K[D]$ is applied. The discussion on the stationary condition is motivated/relies heavily on previous work. This manuscript can be substantially improved by providing CFT motivations for the assumption."

We thank the referee for such a suggestion. In fact, in our precedent work (arXiv:2403.18410), we have shown in detail how to derive the vector fixed-point equation, which is equivalent to the stationarity condition, based on the CFT assumption of the edge. We included the discussion as Example 4.11. We remark that one ``high level'' physical motivation behind the assumptions (both A1 and stationarity condition) is as some conditions for an RG fixed-point. This is what we focus on explaining in the paper.

2- " It is possible for $c_{tot}$ to change along the boundary by adding domain walls or modifying boundary Hamiltonian. Trivially, any pair of opposite chiral theories can be gapped out within a region. I would guess that the stationary assumption is broken on the edge. Is this true, and how does such an assumption fail?"

The referee is correct in stating that the stationarity condition will fail when $c_{tot}$ varies with the position on the edge. As we proved in the paper, $c_{tot}$ being a constant for any regions along the edge is a logical consequence of the stationarity condition for a generic state with a bulk energy gap. Therefore, if one adds some perturbation near the edge such that $c_{tot}$ at some region is changed and doesn't take a constant value along the edge, then the stationarity condition must fail. The exact way in which the assumption fails depends on the details of the perturbation.

A general and intuitive way to see the violation of the stationarity assumption in the scenario the referee mentioned is the following: Consider a two-dimensional gapped system $H$ on a disk, obtained by stacking a chiral system and its anti-chiral system. As a result, the edge is non-chiral and let us assume it is described by a CFT. The groundstate $|\Psi \rangle$ of $H$ shall satisfy the stationarity condition everywhere along the edge. If one add local edge perturbation $gV$ to $H$, such that for some region $c_{tot}(g)$ from the groundstate $|\Psi(g)\rangle$ of $H(g) = H + gV$ is no longer equal to $c_{tot}(g = 0)$, then by simple facts from calculus, for generic $g$, $\delta c_{tot}(g)/\delta g$ is non-zero and hence $c_{tot}(|\Psi(g)\rangle)$ is not stationary. Here by generic $g$, we mean the parameter $g$ is not at some fine-tuned or critical value and it changes the state in a way that $\delta c/\delta g$ is well-defined and $d |\Psi(g)\rangle/dg$ does not blow up.

3- " What happens with a non-conformal boundary to a chiral state? For instance, the $p+ip$ superconductor can be modified such that its edge has a $p^3$ dispersion. Again, what assumptions are violated with such edge?"

The stationarity condition/vector fixed-point equation will be violated if the state is not at an RG fixed-point representative. The general proof of such a statement is currently under investigation. For the specific case of free fermions with $p^3$ dispersion, we did indeed check this: (1) If the dispersion is precisely $\epsilon(p) \sim p^3$, then the groundstate of such a Hamiltonian would be the same as the one obtained from some Hamiltonian of dispersion $\epsilon(p) \sim p$. (At least, we confirm this in some examples.) Such a state is a representative state with edge conformal symmetry and our edge assumption is satisfied. (2) A more interesting and less fine-tuned case is the edge dispersion $\epsilon(p) \sim p - \lambda^2 p^3$, where $\lambda$ is some dimensionful parameter that depends on a Hamiltonian parameter $g$ as $\lambda(g)$. Then, the stationarity condition for the corresponding groundstate will be violated as we tune $g$, as we discussed above. Aside from the violation of stationarity, let us comment that the RG behavior of $c_{tot}$ can be compared to that in the exotic non-relativistic theory [arXiv:1307.8117].

4- "Overall the paper appears technically sound and advances the entanglement bootstrap program to include critical states. The presentation is good overall, but some of the technical formula / assumptions can use more explanations to motivate their form. Perhaps the authors can provide a few more counterexamples of how the assumptions can fail for physical bulk/edges."

We thank the referee for the positive comments. As we mentioned before, we have modified the paragraph that explains the motivation of the edge assumption, and referred to a reference that explains in detail the edge vector fixed-point equation from the assumption that the edge is described by a conformal field theory. In both original and current drafts, we also show that if the edge is at a zero-correlation length fixed-point, then the edge assumption is satisfied. Motivated by these facts, as we've mentioned in the manuscript, we anticipate that the edge assumption shall constrain the edge to be at an RG fixed point.

As for adding more counterexamples: We appreciate the referee's suggestion. We believe a generic state that is not an RG fixed-point will violate the assumption. However, currently we would only be able to explain violation of the assumption via calculations in specific models, but not in a universal way that reflects the insight of RG fixed point of the systems. Therefore, we think having more counterexamples might not be so helpful. We hope in the future we would be able to prove the relation between the RG fixed-point and the edge assumption, and then we would explicitly explain any counterexample from the angle that it is not at an RG fixed point.

---

## Round 2 · List of Changes

1- In the revised manuscript, in response to comments of Referee 2, we briefly mentioned the possible formulation of a robust version of the axioms mentioned in the comments above (page 12 last paragraph).

2- In the revised manuscript, in response to comments of Referee 3, we added some more explanation in the paragraph (second paragraph in page 20) where we discussed the motivation of the vector fixed-point equation assumption, and reminded the reader that the full detailed derivation can be found in arXiv:2403.18410.

3- We fixed some typos, in particular in the titles of appendices C.2 and G.

---

## Editorial Decision

published